# Asymptotics of representation learning in finite Bayesian neural networks

**Jacob A. Zavatone-Veth**[1,2], **Abdulkadir Canatar**[1,2], **Benjamin S. Ruben**[3],
**Cengiz Pehlevan**[2,4]
[1]Department of Physics, [2]Center for Brain Science, [3]Biophysics Graduate Program,
[4]John A. Paulson School of Engineering and Applied Sciences
Harvard University
Cambridge, MA 02138
`{jzavatoneveth,canatara,benruben}@g.harvard.edu`
`cpehlevan@seas.harvard.edu`

## Abstract

Recent works have suggested that finite Bayesian neural networks may sometimes outperform their infinite cousins because finite networks can flexibly adapt their internal representations. However, our theoretical understanding of how the learned hidden layer representations of finite networks differ from the fixed representations of infinite networks remains incomplete. Perturbative finite-width corrections to the network prior and posterior have been studied, but the asymptotics of learned features have not been fully characterized. Here, we argue that the leading finite-width corrections to the average feature kernels for any Bayesian network with linear readout and Gaussian likelihood have a largely universal form. We illustrate this explicitly for three tractable network architectures: deep linear fully-connected and convolutional networks, and networks with a single nonlinear hidden layer. Our results begin to elucidate how task-relevant learning signals shape the hidden layer representations of wide Bayesian neural networks.

## 1 Introduction

The expressive power of deep neural networks critically depends on their ability to learn to represent the features of data [1–24]. However, the structure of their hidden layer representations is only theoretically well-understood in certain infinite-width limits, in which these representations cannot flexibly adapt to learn data-dependent features [3–11, 24]. In the Bayesian setting, these representations are described by fixed, deterministic kernels [3–11]. As a result of this inflexibility, recent works have suggested that finite Bayesian neural networks (henceforth BNNs) may generalize better than their infinite counterparts because of their ability to learn representations [10].

Theoretical exploration of how finite and infinite BNNs differ has largely focused on the properties of the prior and posterior distributions over network outputs [12–17]. In particular, several works have studied the leading perturbative finite-width corrections to these distributions [12–16]. Yet, the corresponding asymptotic corrections to the feature kernels, which measure how representations evolve from layer to layer, have only been studied in a few special cases [16]. Therefore, the structure of these corrections, as well as their dependence on network architecture, remain poorly understood. In this paper, we make the following contributions towards the goal of a complete understanding of feature learning at asymptotically large but finite widths:

- We argue that the leading finite-width corrections to the posterior statistics of the hidden layer kernels of any BNN with a linear readout layer and Gaussian likelihood have a largely

35th Conference on Neural Information Processing Systems (NeurIPS 2021).

prescribed form (Conjecture 1). In particular, we argue that the posterior cumulants of the kernels have well-defined asymptotic series in terms of their prior cumulants, with coefficients that have fixed dependence on the target outputs.

- We explicitly compute the leading finite-width corrections for deep linear fully-connected networks (§4.1), deep linear convolutional networks (§4.2), and networks with a single nonlinear hidden layer (§4.3). We show that our theory yields quantitatively accurate predictions for the result of numerical experiment for tractable linear network architectures, and qualitatively accurate predictions for deep nonlinear networks, where quantitative analytical predictions are intractable.

Our results begin to elucidate the structure of learned representations in wide BNNs. The assumptions of our general argument are satisfied in many regression settings, hence our qualitative conclusions should be broadly applicable.

## 2 Preliminaries

We begin by defining our notation, setup, and assumptions. We will index training and test examples by Greek subscripts $\mu, \nu, \ldots$, and layer dimensions (that is, neurons) by Latin subscripts $j, l, \ldots$. Layers will be indexed by the script Latin letter $\ell$. Matrix- or vector-valued quantities corresponding to a given layer will be indexed with a parenthesized superscript, while scalar quantities that depend only on the layer will be indexed with a subscript. Depending on context, $\| \cdot \|$ will denote the $\ell_2$ norm on vectors or the Frobenius norm on matrices. We denote the standard Euclidean inner product of two vectors $\mathbf{a}, \mathbf{b} \in \mathbb{R}^n$ by $\mathbf{a} \cdot \mathbf{b}$.

### 2.1 Bayesian neural networks with linear readout

Throughout this paper, we consider deep Bayesian neural networks with fully connected linear readout. Such a network $\mathbf{f} : \mathbb{R}^{n_0} \to \mathbb{R}^{n_d}$ with $d$ layers can be written as

$$\mathbf{f}(\mathbf{x}; W^d, \mathcal{W}) = \frac{1}{\sqrt{n_{d-1}}} W^{(d)} \boldsymbol{\psi}(\mathbf{x}; \mathcal{W}), \tag{1}$$

where the feature map $\boldsymbol{\psi}(\cdot; \mathcal{W}) : \mathbb{R}^{n_0} \to \mathbb{R}^{n_{d-1}}$ includes all $d - 1$ hidden layers, collectively parameterized by $\mathcal{W}$. Here, $\boldsymbol{\psi}$ can be some combination of fully-connected feedforward networks, convolutional networks, recurrent networks, et cetera; we assume only that it has a well-defined infinite-width limit in the sense of §2.2. We let the widths of the hidden layers be $n_1, n_2, \ldots, n_{d-1}$; we define the width of a convolutional layer to be its channel count [7]. We assume isotropic Gaussian priors over the trainable parameters [1–23], with $W_{ij}^{(d)} \sim_{\text{i.i.d}} \mathcal{N}(0, \sigma_d^2)$ in particular.

In our analysis, we fix an arbitrary training dataset $\mathcal{D} = \{(\mathbf{x}_\mu, \mathbf{y}_\mu)\}_{\mu=1}^p$ of $p$ examples. We define the input and output Gram matrices of this dataset as $[G_{xx}]_{\mu\nu} \equiv n_0^{-1} \mathbf{x}_\mu \cdot \mathbf{x}_\nu$ and $[G_{yy}]_{\mu\nu} \equiv n_d^{-1} \mathbf{y}_\mu \cdot \mathbf{y}_\nu$, respectively. For analytical tractability, we consider a Gaussian likelihood $p(\mathcal{D} \,|\, \Theta) \propto \exp(-\beta E)$ for

$$E(\Theta; \mathcal{D}) = \frac{1}{2} \sum_{\mu=1}^p \|\mathbf{f}(\mathbf{x}_\mu; \Theta) - \mathbf{y}_\mu\|^2, \tag{2}$$

where $\beta \geq 0$ is an inverse temperature parameter that sets the variance of the likelihood and $\Theta = \{W^{(d)}, \mathcal{W}\}$ [23]. We then introduce the Bayes posterior over parameters given these data:

$$p(\Theta \,|\, \mathcal{D}) = \frac{p(\mathcal{D} \,|\, \Theta) p(\Theta)}{p(\mathcal{D})}; \tag{3}$$

we denote averages with respect to this distribution by $\langle \cdot \rangle$. By tuning $\beta$, one can then adjust whether the posterior is dominated by the prior ($\beta \ll 1$) or the likelihood ($\beta \gg 1$). We will mostly focus on the case in which the input dimension is large and the training dataset can be linearly interpolated; the low-temperature limit $\beta \to \infty$ then enforces the interpolation constraint.

### 2.2 The Gaussian process limit

We consider the limit of large hidden layer widths $n_1, n_2, \ldots, n_{d-1} \to \infty$ with $n_0, n_d, p$, and $d$ fixed. More precisely, we consider a limit in which $n_\ell = \alpha_\ell n$ for $\ell = 1, \ldots, d - 1$, where $\alpha_\ell \in (0, \infty)$ and

$n \to \infty$, as studied by [3–15, 17–19, 24] and others. Importantly, we note that size of $n_0$ relative to $n$ is unimportant for our results, whereas $n_d/n$ and $d/n$ must be small [10, 12, 17].

In this limit, for $\psi$ built out of compositions of most standard neural network architectures, the prior over function values $\mathbf{f}$ tends to a Gaussian process (GP) [3–8]. Moreover, with our choice of a Gaussian likelihood, the posterior over function values also tends weakly to the posterior induced by the limiting GP prior [25]. The kernel of the limiting GP prior is given by the deterministic limit $K_\infty^{(d-1)}$ of the inner product kernel of the postactivations of the final hidden layer,

$$K^{(d-1)}(\mathbf{x}, \mathbf{x}') \equiv n_{d-1}^{-1} \psi(\mathbf{x}, \mathcal{W}) \cdot \psi(\mathbf{x}', \mathcal{W}), \tag{4}$$

multiplied by the prior variance $\sigma_d^2$ [3–8]. For a broad range of network architectures, $K_\infty^{(d-1)}$ can be computed recursively [5–8]. For brevity, we define the kernel matrix evaluated on the training data: $[K^{(d-1)}]_{\mu\nu} \equiv K^{(d-1)}(\mathbf{x}_\mu, \mathbf{x}_\nu)$.

## 3 Elementary perturbation theory for finite Bayesian neural networks

We first present our main result, which shows that the form of the leading perturbative correction to the average hidden layer kernels of a BNN is tightly constrained by the assumptions that the readout is linear, that the cost is quadratic, and that the GP limit is well-defined.

### 3.1 Finite-width corrections to the posterior cumulants of hidden layer observables

Our main result is as follows:

**Conjecture 1** *Consider a BNN of the form* (1)*, with posterior* (3)*. Assume that this network admits a well-defined GP limit as discussed in §2.2. Let $O$ be a* hidden layer observable*, that is, a function of the hidden layer activations that is not a function of the readout weights $W_d$. Assume that $O$ tends in probability to a finite, deterministic limit $O_\infty$ under the posterior in the GP limit.*

*Then, the posterior cumulants of this observable admit well-behaved asymptotic series at large widths in terms of its joint prior cumulants with the postactivation kernel $K^{(d-1)}$. In particular, the asymptotic expansion of the posterior mean $\langle O \rangle$ has leading terms*

$$\langle O \rangle = \mathbb{E}_\mathcal{W} O + \frac{1}{2} n_d \sum_{\rho, \lambda=1}^p [\sigma_d^{-2} \Gamma^{-1} G_{yy} \Gamma^{-1} - \Gamma^{-1}]_{\rho\lambda} \operatorname{cov}_\mathcal{W}(O, K_{\rho\lambda}^{(d-1)}) + \dots, \tag{5}$$

*where $\Gamma \equiv K_\infty^{(d-1)} + \beta^{-1}\sigma_d^{-2} I_p$. Here, the cumulants of the kernels are computed with respect to the* prior*, and are themselves given by asymptotic series at large widths. The ellipsis denotes terms that are of subleading order in the inverse hidden layer widths.*

In Appendix B, we derive this result perturbatively by expanding the posterior cumulant generating function of $O$ in powers of the deviations of $O$ and $K^{(d-1)}$ from their deterministic infinite-width values. There, we also give an asymptotic formula for the posterior covariance of two observables. However, the resulting perturbation series may not rigorously be an asymptotic series, and this method does not yield quantitative bounds for the width-dependence of the terms. We therefore frame it as a conjecture. We note that similar methods can be applied to compute asymptotic corrections to the posterior predictive statistics; we comment on this possibility in Appendix G.

Though this conjecture applies to a broad class of hidden layer observables, the observables of greatest interest are the preactivation or postactivation kernels of the hidden layers within the feature map $\psi$. We will focus on the postactivation kernels $K^{(\ell)}$, which measure how the similarities between inputs evolve as they are propagated through the network [5–10].

Conjecture 1 posits that there are two possible types of leading finite-width corrections to the average kernels. The first class of corrections are deviations of $\mathbb{E}_\mathcal{W} K^{(\ell)}$ from $K_\infty^{(\ell)}$. These terms reflect corrections to the prior, and do not reflect non-trivial representation learning as they are independent of the outputs. For fully-connected networks, also known as multilayer perceptrons (MLPs), work by Yaida [12] and by Gur-Ari and colleagues [18, 19] shows that $\mathbb{E}_\mathcal{W} K^{(\ell)} = K_\infty^{(\ell)} + \mathcal{O}(n^{-1})$. The

second type of correction is the output-dependent term that depends on $\mathrm{cov}_{\mathcal{W}}(K_{\mu\nu}^{(\ell)}, K_{\rho\lambda}^{(d-1)})$. For deep linear MLPs or MLPs with a single hidden layer, $\mathbb{E}_{\mathcal{W}} K^{(\ell)}$ is exactly equal to $K_{\infty}^{(\ell)}$ at any width (see Appendix C) [3, 12, 18], and only the covariance term contributes. More broadly, these prior works show that $\mathrm{cov}_{\mathcal{W}}(K_{\mu\nu}^{(\ell)}, K_{\rho\lambda}^{(d-1)}) = \mathcal{O}(n^{-1})$ for MLPs, and that higher cumulants are of $\mathcal{O}(n^{-2})$ [12, 18, 19]. Some of these results have recently been extended to convolutional networks by Andreassen and Dyer [26]. Thus, the finite-width correction to the prior mean should not dominate the feature-learning covariance term, and the terms hidden in the ellipsis should indeed be suppressed.

The leading output-dependent correction has several interesting features. First, it includes a factor of $n_d$, reflecting the fact that inference in wide Bayesian networks with many outputs is qualitatively different from that in networks with few outputs relative to their hidden layer width [10]. If $n_d/n$ does not tend to zero with increasing $n$, the infinite-width behavior is not described by a standard GP [8, 10]. Moreover, we note that the matrix $\Gamma$ is invertible at any finite temperature, even when $K_{\infty}^{(d-1)}$ is singular. Therefore, provided that one can extend the GP kernel by continuity to non-invertible $G_{xx}$, Conjecture 1 can be applied in the data-dense regime $n_0 < p$ as well as the data-sparse regime $n_0 > p$. Furthermore, we observe that the correction depends on the outputs only through their Gram matrix $G_{yy}$. This result is intuitively sensible, since with our choice of likelihood and prior the function-space posterior is invariant under simultaneous rotation of the output activations and targets. Finally, $G_{yy}$ is transformed by factors of the matrix $\Gamma^{-1}$, hence the correction depends on certain interactions between the output similarities and the GP kernel $K_{\infty}^{(d-1)}$.

## 3.2 High- and low-temperature limits of the leading correction

To gain some intuition for the properties of the leading finite-width corrections, we consider their high- and low-temperature limits. These limits correspond to tuning the posterior (3) to be dominated by the prior or the likelihood, respectively. At high temperatures ($\beta \ll 1$), expanding $\Gamma^{-1}$ as a Neumann series (see Appendix A and [27]) yields

$$\sigma_d^{-2}\Gamma^{-1}G_{yy}\Gamma^{-1} - \Gamma^{-1} = -\beta\sigma_d^2 I_p + (\beta\sigma_d^2)^2(\sigma_d^{-2}G_{yy} + K_{\infty}^{(d-1)}) + \mathcal{O}[(\beta\sigma_d^2)^3]. \tag{6}$$

Thus, at high temperatures, the outputs only influence the average kernels of Conjecture 1 to subleading order in both width and $\beta$, which reflects the fact that the likelihood is discounted relative to the prior in this regime. Moreover, the leading output-dependent contribution averages together $G_{yy}$ and $K_{\infty}^{(d-1)}$, hence, intuitively, there is no way to 'cancel' the GP contributions to the average kernels. We note that, at infinite temperature ($\beta = 0$), the posterior reduces to the prior, and all finite-width corrections to the average kernels arise from the discrepancy between $\mathbb{E}_{\mathcal{W}} K^{(\ell)}$ and $K_{\infty}^{(\ell)}$.

At low temperatures ($\beta \gg 1$), the behavior of $\Gamma^{-1}$ differs depending on whether or not $K_{\infty}^{(d-1)}$ is of full rank. Assuming for simplicity that it is invertible, we have

$$\sigma_d^{-2}\Gamma^{-1}G_{yy}\Gamma^{-1} - \Gamma^{-1} = [K_{\infty}^{(d-1)}]^{-1}(\sigma_d^{-2}G_{yy} - K_{\infty}^{(d-1)})[K_{\infty}^{(d-1)}]^{-1} + \mathcal{O}[(\beta\sigma_d^2)^{-1}]; \tag{7}$$

in the non-invertible case there are additional contributions involving projectors onto the null space of $K_{\infty}^{(d-1)}$. Therefore, the leading-order low temperature correction depends on the difference between the target and GP kernels, while the leading non-trivial high temperature correction depends on their sum.

# 4 Learned representations in tractable network architectures

Having derived the general form of the leading perturbative finite-width correction to the average feature kernels, we now consider several example network architectures. For these tractable examples, we provide explicit formulas for the feature-learning corrections to the hidden layer kernels, and test the accuracy of our theory with numerical experiments.

## 4.1 Deep linear fully-connected networks

We first consider deep linear fully-connected networks with no bias terms. Concretely, we consider a network with activations $\mathbf{h}^{(\ell)} \in \mathbb{R}^{n_\ell}$ recursively defined via $\mathbf{h}^{(\ell)} = n_{\ell-1}^{-1/2}W^{(\ell)}\mathbf{h}^{(\ell-1)}$ with base case

$\mathbf{h}^{(0)} = \mathbf{x}$, where the prior distribution of weights is $[W^{(\ell)}]_{ij} \sim_{\text{i.i.d.}} \mathcal{N}(0, \sigma_\ell^2)$. For such a network, the hidden layer kernels $[K^{(\ell)}]_{\mu\nu} \equiv n_\ell^{-1} \mathbf{h}_\mu^{(\ell)} \cdot \mathbf{h}_\nu^{(\ell)}$ have deterministic limits $K_\infty^{(\ell)} = m_\ell^2 G_{xx}$, where $m_\ell^2 \equiv \sigma_\ell^2 \sigma_{\ell-1}^2 \cdots \sigma_1^2$ is the product of prior variances up to layer $\ell$. Higher prior cumulants of the kernels are easy to compute with the aid of Isserlis' theorem for Gaussian moments (see Appendix C) [28, 29], yielding

$$\frac{\langle K^{(\ell)} \rangle}{m_\ell^2} = G_{xx} + \left( \sum_{\ell'=1}^{\ell} \frac{n_d}{n_{\ell'}} \right) G_{xx} \Gamma^{-1} \left( m_d^{-2} G_{yy} - \Gamma \right) \Gamma^{-1} G_{xx} + \mathcal{O}(n^{-2}), \tag{8}$$

where $\Gamma \equiv G_{xx} + I_p/(\beta m_d^2)$ and $\ell = 1, \ldots, d-1$. In Appendix D, we show that this result can be derived directly through an *ab initio* perturbative calculation of the cumulant generating function of the kernels, without relying on our heuristic argument for the general version of Conjecture 1. Moreover, in Appendix E, we show that the form of the correction remains the same even if one allows arbitrary forward skip connections, though the dependence on width and depth is given by a more complex recurrence relation.

Thus, the leading corrections to the normalized average kernels $\langle K^{(\ell)} \rangle / m_\ell^2$ are identical across all hidden layers up to a scalar factor that encodes the width-dependence of the correction. This sum-of-inverse-widths dependence was previously noted by Yaida [12] in his study of the corrections to the prior of a deep linear network. For a network with hidden layers of equal width $n$, we have the simple linear dependence $\sum_{\ell'=1}^{\ell}(n_d/n_{\ell'}) = n_d\ell/n$. If one instead includes a narrow bottleneck in an otherwise wide network, this dependence predicts that the kernels before the bottleneck should be close to their GP values, while those after the bottleneck should deviate strongly.

This result simplifies further at low temperatures, where, by the result of §3.2, we have

$$\frac{\langle K^{(\ell)} \rangle}{m_\ell^2} = G_{xx} + \left( \sum_{\ell'=1}^{\ell} \frac{n_d}{n_{\ell'}} \right) \left( m_d^{-2} G_{yy} - G_{xx} \right) + \mathcal{O}(n^{-2}, \beta^{-1}) \tag{9}$$

in the regime in which $G_{xx}$ is invertible. We thus obtain the simple qualitative picture that the low-temperature average kernels linearly interpolate between the input and output Gram matrices. In Appendix F, we show that this limiting result can be recovered from the recurrence relation derived through other methods by Aitchison [10], who did not use it to compute finite-width corrections. We note that the low-temperature limit is peculiar in that the mean predictor reduces to the least-norm pseudoinverse solution to the underlying underdetermined linear system $XW = Y$; we comment on this property in Appendix G.

We can gain some additional understanding of the structure of the correction by using the eigendecomposition of $G_{xx}$. As $G_{xx}$ is by definition a real positive semidefinite matrix, it admits a unitary eigendecomposition $G_{xx} = U\Lambda U^\dagger$ with non-negative eigenvalues $\Lambda_{\mu\mu}$. In this basis, the average kernel is

$$\frac{1}{m_\ell^2} U^\dagger \langle K^{(\ell)} \rangle U = \Lambda + \left( \sum_{\ell'=1}^{\ell} \frac{n_d}{n_{\ell'}} \right) \left( m_d^{-2} \tilde{\Lambda} U^\dagger G_{yy} U \tilde{\Lambda} - \tilde{\Lambda}\Lambda \right) + \mathcal{O}(n^{-2}), \tag{10}$$

where we have defined the diagonal matrix $\tilde{\Lambda} \equiv \beta m_d^2 \Lambda (I_p + \beta m_d^2 \Lambda)^{-1}$. As $\beta m_d^2 \Lambda \geq 0$, the diagonal elements of $\tilde{\Lambda}$ are bounded as $0 \leq \tilde{\Lambda}_{\mu\mu} \leq 1$. Thus, the factors of $\Gamma^{-1} G_{xx}$ by which $G_{yy}$ is conjugated have the effect of suppressing directions in the projection of $G_{yy}$ onto the eigenspace of $G_{xx}$ with small eigenvalues. We can see that this effect will be enhanced at high temperatures ($\beta \ll 1$) and small scalings ($m_d^2 \ll 1$), and suppressed at low temperatures and large scalings. For this linear network, similarities are not enhanced, only suppressed. Moreover, if $G_{xx}$ is diagonal, then a given element of the average kernel will depend only on the corresponding element of $G_{yy}$.

We now seek to numerically probe how accurately these asymptotic corrections predict learned representations in deep fully-connected linear BNNs. Using Langevin sampling [30, 31], we trained deep linear networks of varying widths, and compared the difference between the empirical and GP kernels with theory predictions. We provide a detailed discussion of our numerical methods in Appendix I. In Figure 1, we present an experiment with a 2-layer linear neural network trained on the MNIST dataset of handwritten digit images [32] using the Neural Tangents library [33]. We find an excellent agreement with our theory, confirming the inverse scaling with width and linear scaling with depth for the deviations from GP kernel.

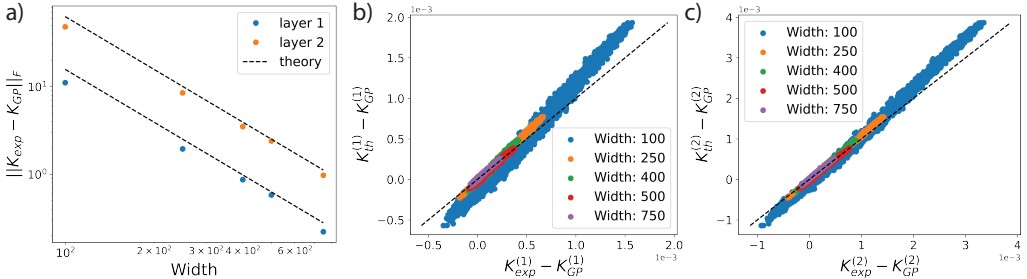

Figure 1: Learned representations in two-hidden-layer linear fully-connected neural networks with varying widths trained via Langevin sampling on 5000 MNIST images (see Appendix I for more details). **(a)** The Frobenius norm of the deviation of the empirical average kernel of each layer from its GP value (in this case, simply $G_{xx}$) for varying widths. We see perfect match with theoretical predictions, which are shown as dashed lines. We obtain the predicted $1/n$ decay with increasing width and the linear scaling with the depth where the deviations for first and second layers differ by a factor of 2. **(b-c)** Scatter plot of individual elements of the experimental (ordinate) and theoretical (abscissa) kernels for both layers. For low widths a slight deviation is visible between experiment and theory, while for larger widths the agreement is better.

## 4.2 Deep linear convolutional networks

To demonstrate the applicability of Conjecture 1 to non-fully-connected BNNs, we consider deep convolutional linear networks with no bias terms. Here, the appropriate notion of width is the number of channels in each hidden layer [7]. Following the setup of Novak et al. [7] and Xiao et al. [34], we consider a network consisting of $d - 1$ linear convolutional layers followed by a fully-connected linear readout layer. For simplicity, we restrict our attention to convolutions with periodic boundary conditions, and do not include internal pooling layers (see Appendix C for more details). Concretely, we consider a network with hidden layer activations $h_{i,\mathfrak{a}}^{(\ell)}$, where $i$ indexes the $n_\ell$ channels of the layer and $\mathfrak{a}$ is a spatial multi-index. The hidden layer activations are then defined through the recurrence

$$h_{i,\mathfrak{a}}^{(\ell)}(x) = \frac{1}{\sqrt{n_{\ell-1}}} \sum_{j=1}^{n_{\ell-1}} \sum_{\mathfrak{b}} w_{ij,\mathfrak{b}}^{(\ell)} h_{j,\mathfrak{a}+\mathfrak{b}}^{(\ell-1)}(x) \tag{11}$$

with base case $h_{i,\mathfrak{a}}^{(0)}(x) = x_{i,\mathfrak{a}}$, where $i$ indexes the input channels (e.g., image color channels). The feature map is then formed by flattening the output of the last hidden layer into an $n_{d-1}s$-dimensional vector, where $s$ is the total dimensionality of the inputs (see Appendix C for details). We fix the prior distribution of the filter elements to be $w_{ij,\mathfrak{a}}^{(\ell)} \underset{\text{i.i.d.}}{\sim} \mathcal{N}(0, \sigma_\ell^2 v_{\mathfrak{a}})$, where $v_{\mathfrak{a}} > 0$ is a weighting factor that sets the fraction of receptive field variance at location $\mathfrak{a}$ (and is thus subject to the constraint $\sum_{\mathfrak{a}} v_{\mathfrak{a}} = 1$). For inputs $[x_\mu]_{i,\mathfrak{a}}$ and $[x_\nu]_{i,\mathfrak{a}}$, we introduce the four-index hidden layer kernels

$$K_{\mu\nu,\mathfrak{a}\mathfrak{b}}^{(\ell)} \equiv \frac{1}{n_\ell} \sum_{i=1}^{n_\ell} h_{i,\mathfrak{a}}^{(\ell)}(x_\mu) h_{i,\mathfrak{b}}^{(\ell)}(x_\nu). \tag{12}$$

With the given readout strategy, the two-index feature map kernel appearing in Conjecture 1 is related to the four-index kernel of the last hidden layer by $K_{\mu\nu}^{(d-1)} = \frac{1}{s} \sum_{\mathfrak{a}} K_{\mu\nu,\mathfrak{a}\mathfrak{a}}^{(d-1)}$. We discuss other readout strategies in Appendix C, but use this vectorization strategy in our numerical experiments.

As shown by Xiao et al. [34], the infinite-width four-index kernel obeys the recurrence

$$[K_\infty^{(\ell)}]_{\mu\nu,\mathfrak{a}\mathfrak{b}} = \sigma_\ell^2 \sum_{\mathfrak{c}} v_{\mathfrak{c}} [K_\infty^{(\ell-1)}]_{\mu\nu,(\mathfrak{a}+\mathfrak{c})(\mathfrak{b}+\mathfrak{c})} \tag{13}$$

with base case $[K_\infty^0]_{\mu\nu,\mathfrak{a}\mathfrak{b}} = [G_{xx}]_{\mu\nu,\mathfrak{a}\mathfrak{b}} \equiv \frac{1}{n_0} \sum_{i=1}^{n_0} [x_\mu]_{i,\mathfrak{a}} [x_\nu]_{i,\mathfrak{b}}$. This gives convolutional linear networks a sense of spatial hierarchy that is not present in the fully-connected case: even at infinite width, the kernels include iterative spatial averaging.

In Appendix C, we derive the kernel covariances appearing in Conjecture 1. As in the fully-connected case, this computation is easy to perform with the aid of Isserlis' theorem. The general result is

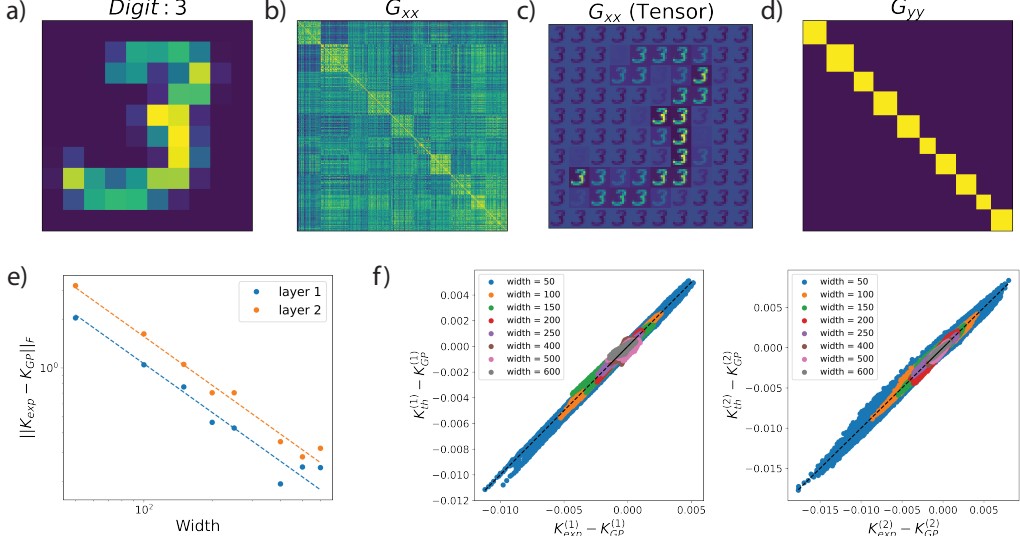

Figure 2: The MNIST image dataset and experiments for neural networks with two 1D convolutional layers. **(a)** A $10 \times 10$ MNIST image downsized from $28 \times 28$ pixels. **(b)** Input Gram matrix for 300 MNIST images. **(c)** A single $(\mu, \nu)$ component of the input tensor $[G_{xx}]_{\mu\nu,\mathfrak{ab}}$ obtained using Eq. (13). **(d)** The output Gram matrix. **(e)** The Frobenius norm of the correction to the 1D convolutional GP kernel is inversely proportional to the width. Here, the dashed lines are the theoretical predictions. **(f)** Scatter plots of individual elements of the empirical corrections to the GP kernels against the theoretical predictions for both layers show excellent agreement.

somewhat complicated, but things simplify under the assumption that readout is performed using vectorization. Then, one finds that

$$\langle K_{\mu\nu,\mathfrak{ab}}^{(\ell)} \rangle = [K_\infty^{(\ell)}]_{\mu\nu,\mathfrak{ab}} + \left( \prod_{\ell'=\ell}^{d-1} \sigma_\ell^2 \right) \left( \sum_{\ell'=1}^{\ell} \frac{n_d}{n_{\ell'}} \right) \frac{1}{s} \sum_{\mathfrak{c}=1}^{s} \sum_{\rho,\lambda=1}^{p} [K_\infty^{(\ell)}]_{\mu\rho,\mathfrak{ac}} \Phi_{\rho\lambda} [K_\infty^{(\ell)}]_{\lambda\nu,\mathfrak{cb}} + \mathcal{O}(n^{-2}),$$

(14)

where we have defined $\Phi_{\rho\lambda} \equiv [\sigma_d^{-2} \Gamma^{-1} G_{yy} \Gamma^{-1} - \Gamma^{-1}]_{\rho\lambda}$ for brevity. Thus, the correction to the convolutional kernel is quite similar to that obtained in the fully-connected case. To this order, the difference between these network architectures manifests itself largely through the difference in the infinite-width kernels. In Appendix C, we show that a similar simplification holds if readout is performed using global average pooling over space.

As we did for fully-connected networks, we test whether our theory accurately predicts the results of numerical experiment, using the MNIST digit images illustrated in 2(a-d). We consider a network with one-dimensional (Figure 2e and f) and two-dimensional (Figure 3) convolutional hidden layers, trained to classify 50 MNIST images (see Appendix I for details of our numerical methods). As shown in Figure 2(e, f) (Figure 3(a,b) for 2D convolutions), we again obtain good quantitative agreement between the predictions of our asymptotic theory and the results of numerical experiment. In Figure 3c, we directly visualize the learned feature kernels for 2D convolutional layers, illustrating the good agreement between theory and experiment. Therefore, our asymptotic theory can be applied to accurately predict learned representations in deep convolutional linear networks.

## 4.3 Networks with a single nonlinear hidden layer

Finally, we would like to gain some understanding of how including nonlinearity affects the structure of learned representations. However, for a nonlinear MLP, it is usually not possible to analytically compute $\mathrm{cov}_{\mathcal{W}}(K_{\mu\nu}^{(\ell)}, K_{\rho\lambda}^{(d-1)})$ to the required order [9, 12, 18, 19]. Here, we consider the case of a network with a single nonlinear layer and no bias terms, in which we can both summarize the key obstacles to studying deep nonlinear networks and gain some intuitions about how they might differ from linear BNNs. Concretely, we consider a network with feature map $\psi(\mathbf{x}; W^{(1)}) =$

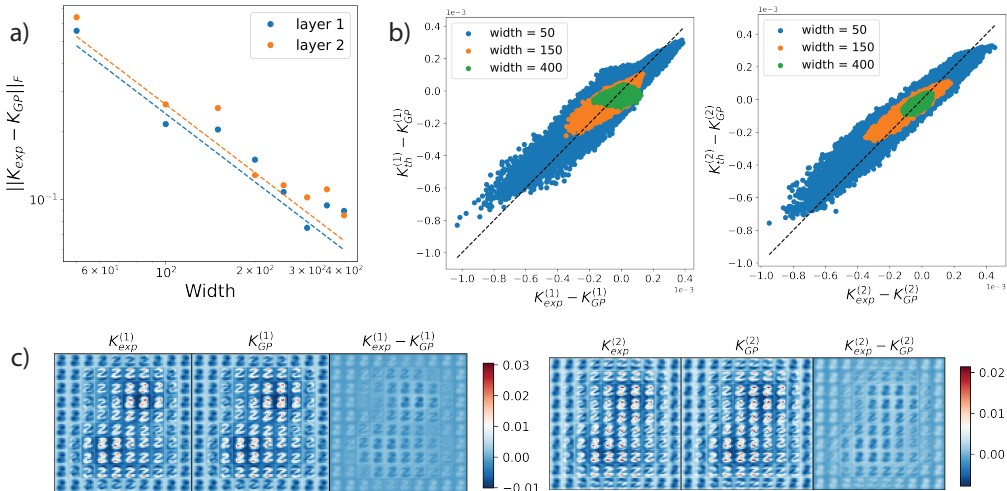

Figure 3: Learned representations in two-hidden-layer linear 2-D convolutional networks of varying channel widths. **(a)** The Frobenius norm of the correction to the GP kernel is inversely proportional to the width. Here, the dashed lines represent theory predictions. **(b)** Scatter plots of individual elements of the empirical corrections to the GP kernels against the theoretical predictions for both layers show good agreement. **(c)** A single component $(\mu, \nu)$ of the learned feature kernels in 2-layer CNN experiments for both convolutional layers. While the experimental kernel looks quite similar the GP (first and second columns), their difference shows the finite width corrections to the GP (last column).

$\phi(n_0^{-1/2} W^{(1)} \mathbf{x})$ for an elementwise activation function $\phi$, where the weight matrix $W^{(1)}$ has prior distribution $[W^{(1)}]_{ij} \sim_{\text{i.i.d.}} \mathcal{N}(0, \sigma_1^2)$. The only hidden layer kernel of this network is the feature map postactivation kernel $K_{\mu\nu}$ defined in (4), where we drop the layer index for brevity. As detailed in Appendix H, for such a network we have the exact expressions

$$[K_\infty]_{\mu\nu} = \mathbb{E}_{\mathcal{W}} K_{\mu\nu} = \mathbb{E}[\phi(h_\mu)\phi(h_\nu)], \tag{15}$$

$$n_1 \operatorname{cov}_{\mathcal{W}}(K_{\mu\nu}, K_{\rho\lambda}) = \mathbb{E}[\phi(h_\mu)\phi(h_\nu)\phi(h_\rho)\phi(h_\lambda)] - [K_\infty]_{\mu\nu}[K_\infty]_{\rho\lambda}, \tag{16}$$

where expectations are taken over the $p$-dimensional Gaussian random vector $h_\mu$, which has mean zero and covariance $\operatorname{cov}(h_\mu, h_\nu) = \sigma_1^2 [G_{xx}]_{\mu\nu}$. Unlike for deeper nonlinear networks, here there are no finite-width corrections to the prior expectations [3, 12, 18].

Though these expressions are easy to define, it is not possible to evaluate the four-point expectation in closed form for general Gram matrices $G_{xx}$ and activation functions $\phi$, including ReLU and erf. This obstacle has been noted in previous studies [9, 12, 15], and makes it challenging to extend approaches similar to those used here to deeper nonlinear networks. For polynomial activation functions, the required expectations can be evaluated using Isserlis' theorem (see Appendix A). However, even for a quadratic activation function $\phi(x) = x^2$, the resulting formula for the kernel will involve many elementwise matrix products, and cannot be simplified into an intuitively comprehensible form.

If the input Gram matrix $G_{xx}$ is diagonal, the four-point expectation becomes tractable because the required expectations factor across sample indices. In this simple case, there is an interesting distinction between the behavior of activation functions that yield $\mathbb{E}\phi(h) = 0$ and those that yield $\mathbb{E}\phi(h) \neq 0$. As detailed in §4.1, if $\mathbb{E}\phi(h) = 0$, $K_\infty$ is diagonal, and a given element of the leading finite-width correction to $\langle K \rangle$ depends only on the corresponding element of $G_{yy}$. However, if $\mathbb{E}\phi(h) \neq 0$, then $K_\infty$ includes a rank-1 component, and each element of the correction depends on all elements of $G_{yy}$. This means that the case in which $G_{xx}$ is diagonal is qualitatively distinct from the case in which there is only a single training input for such activation functions.

## 5 Learned representations in deep nonlinear networks

In the preceding section, we noted that analytical study of learned representations in deep nonlinear BNNs is generally quite challenging. Here, we use numerical experiments to explore whether any

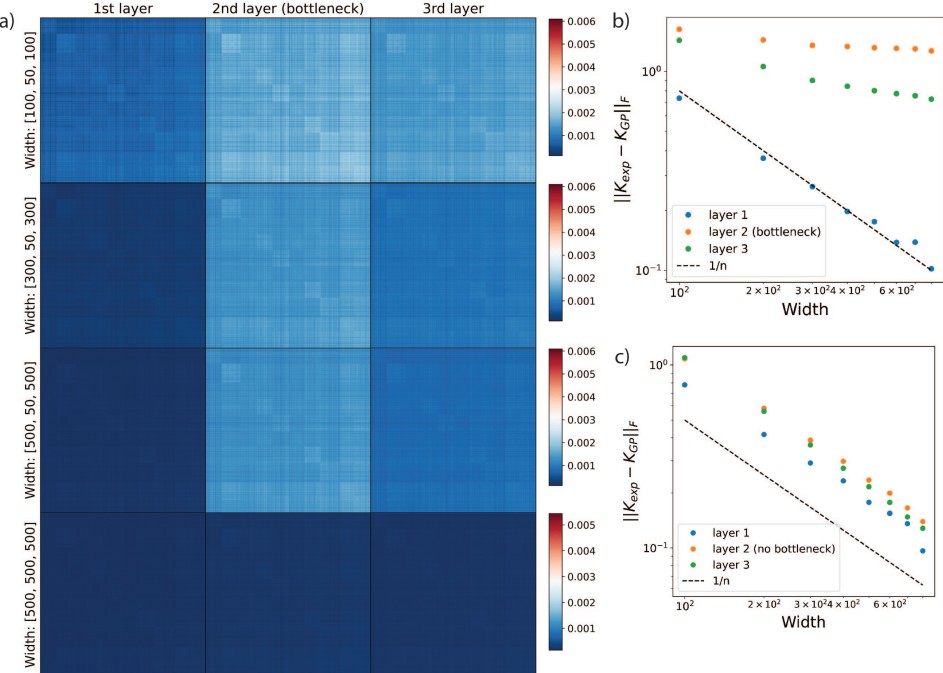

Figure 4: 3-hidden layer neural network with ReLU activations trained via Langevin sampling on 1000 MNIST images (see Appendix I). **(a)** The empirical average kernels subtracted from their corresponding GP kernels for all layers with varying widths. Labels on the y-axes indicate the widths of each layer. We observe that for networks with bottleneck layers, the deviation from $K_\infty^{(\ell)}$ is largest at the bottleneck indicating representation learning; without a bottleneck deviations are considerably less (the last row). **(b)** Hidden layer kernel deviation from GP kernels as a function of width for bottleneck networks. While the first layer shows $1/n$ scaling, the bottleneck layer and the 3rd layer deviations stay almost constant. This behavior is predicted analytically for linear networks. **(c)** As in (b) for networks without a bottleneck. Consistent with our theory, all layers display $1/n$ decay.

of the intuitions gained in the linear setting carry over to nonlinear networks. Concretely, we study how narrow bottlenecks affect representation learning in a more realistic nonlinear network. We train a network with three hidden layers and ReLU activations on a subset of the MNIST dataset [32]. Despite its analytical simplicity, ReLU is among the activation functions for which the covariance term in Conjecture 1 cannot be evaluated in closed form (see §4.3). However, it is straightforward to simulate numerically. Consistent with the predictions of our theory for linear networks, we find that introducing a narrow bottleneck leads to more representation learning in subsequent hidden layers, even if those layers are quite wide (Figure 4). Quantitatively, if one increases the width of the hidden layers between which the fixed-width bottleneck is sandwiched, the deviation of the first layer's kernel from its GP value decays roughly as $1/n$ with increasing width, while the deviations for the bottleneck and subsequent layers remain roughly constant. In contrast, the kernel deviations throughout a network with equal-width hidden layers decay roughly as $1/n$ (Figure 4). These observations are qualitatively consistent with the width-dependence of the linear network kernel (8), as well as with previous studies of networks with infinitely-wide layers separated by a finite bottleneck [35]. Keeping in mind the obstacles noted in §4.3, precise characterization of nonlinear networks will be an interesting objective for future work.

## 6 Related work

Our work is closely related to several recent analytical studies of finite-width BNNs. First, Aitchison [10] argued that the flexibility afforded by finite-width BNNs can be advantageous. He derived a

recurrence relation for the learned feature kernels in deep linear networks, which he solved in the limits of infinite width and few outputs, narrow width and many outputs, and infinite width and many outputs. As discussed in §4.1 and in Appendix F, our results on deep linear networks extend those of his work. Furthermore, our numerical results support his suggestion that networks with narrow bottlenecks may learn interesting features.

Moreover, our analytical approach and the asymptotic regime we consider mirror recent perturbative studies of finite-width BNNs. As noted in §3 and Appendix B, we make use of the results of Yaida [12], who derived recurrence relations for the perturbative corrections to the cumulants of the finite-width prior for an MLP. However, Yaida did not attempt to study the statistics of learned features; the goal of his work was to establish a general framework for the study of finite-width corrections. Bounds on the prior cumulants of a broader class of observables have been studied by Gur-Ari and colleagues [18, 19, 26]; these results could allow for the identification of observables to which Conjecture 1 should apply. Finally, perturbative corrections to the network prior and posterior have been studied by Halverson et al. [13] and Naveh et al. [15], respectively. Our work builds upon these studies by perturbatively characterizing the internal representations that are learned upon inference.

Following the appearance of our work in preprint form, Roberts et al. [36] announced an alternative derivation of the zero-temperature limit of Conjecture 1 for MLPs; we have adopted their terminology of hidden layer observables. As in Yaida [12]'s earlier work, they rely on sequential perturbative approximation of the prior over preactivations as the hidden layers are marginalized out in order from the first to the last. While our elementary perturbative argument for Conjecture 1 does not require assuming a particular network architecture for the hidden layers, it takes as input information regarding the prior cumulants that would have to be approximated using such methods. Moreover, the approach of layer-by-layer approximation to the prior could enable a fully rigorous version of Conjecture 1 to be proved on an architecture-by-architecture basis [37].

Our work, like most studies of wide BNNs [3–15, 17–19, 24], focuses on the regime in which the sample size $p$ is held fixed while the hidden layer width scale $n$ tends to infinity, i.e., $p \ll n$. One can instead consider regimes in which $p$ is not negligible relative to $n$, in which the posterior would be expected to concentrate. The behavior of deep linear BNNs in this regime was recently studied by Li and Sompolinsky [16], who computed asymptotic approximations for the predictor statistics and hidden layer kernels. In Appendix F, we show that our result (9) for the zero-temperature kernel can be recovered as the $p/n \downarrow 0$ limit of their result. As the dataset size $p$ appears only implicitly in our approach, we leave the incorporation of large-$p$ corrections as an interesting objective for future work. We note, however, that alternative methods developed to study the large-$p$ regime [16, 38] cannot overcome the obstacles to analytical study of deep nonlinear networks encountered here.

# 7   Conclusions

In this paper, we have shown that the leading perturbative feature learning corrections to the infinite-width kernels of wide BNNs with linear readout and least-squares cost should be of a tightly constrained form. We demonstrate analytically and with numerical experiments that these results hold for certain tractable network architectures, and conjecture that they should extend to more general network architectures that admit a well-defined GP limit.

*Limitations.* We emphasize that our perturbative argument for Conjecture 1 is not rigorous, and that we have not obtained quantitative bounds on the remainder for general network architectures. It is possible that there are non-perturbative contributions to the posterior statistics that are not captured by Conjecture 1; non-perturbative investigation of feature learning in finite BNNs will be an interesting objective for future work [17, 39]. More broadly, we leave rigorous proofs of the applicability of our results to more general architectures and of the smallness of the remainder as objective for future work. As mentioned above, one could attempt such a proof on an architecture-by-architecture basis [12, 36, 37]. Alternatively, one could attempt to treat all sufficiently sensible architectures uniformly [8, 9]. Furthermore, we have considered only one possible asymptotic regime: that in which the width is taken to infinity with a finite training dataset and small output dimensionality. As discussed above in reference to the work of Aitchison [10] and Li and Sompolinsky [16], investigation of alternative limits in which output dimension, dataset size, depth, and hidden layer width are all taken to infinity with fixed ratios may be an interesting subject for future work.

## Acknowledgments and Disclosure of Funding

We thank B. Bordelon for helpful comments on our manuscript. JAZ-V acknowledges partial support from the NSF-Simons Center for Mathematical and Statistical Analysis of Biology at Harvard and the Harvard Quantitative Biology Initiative. This work was further supported by the Harvard Data Science Initiative Competitive Research Fund, the Harvard Dean's Competitive Fund for Promising Scholarship, and a Google Faculty Research Award. The authors declare no conflict of interest.

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
