# Supplemental Material for:
# "Asymptotics of representation learning in finite Bayesian neural networks"

**Jacob A. Zavatone-Veth**[1,2]**, Abdulkadir Canatar**[1,2]**, Benjamin S. Ruben**[3]**,
Cengiz Pehlevan**[2,4]

[1]Department of Physics, [2]Center for Brain Science, [3]Biophysics Graduate Program,
[4]John A. Paulson School of Engineering and Applied Sciences
Harvard University
Cambridge, MA 02138
{jzavatoneveth,canatara,benruben}@g.harvard.edu
cpehlevan@seas.harvard.edu

## Contents

35th Conference on Neural Information Processing Systems (NeurIPS 2021).

## A   Preliminary technical results

In this appendix, we review useful technical results upon which our calculations rely.

### A.1  Isserlis' theorem for Gaussian moments

Let $(x_1, x_2, \ldots, x_n)$ be a zero-mean Gaussian random vector. Then, Isserlis' theorem [1] states that

$$\mathbb{E}[x_1 x_2 \cdots x_n] = \begin{cases} \sum_{p \in P_n^2} \prod_{(i,j) \in p} \operatorname{cov}(x_i, x_j) & n \text{ even} \\ 0 & n \text{ odd}, \end{cases} \tag{A.1}$$

where the sum is over all pairings $p$ of $\{1, 2, \ldots, n\}$ and the product is over all pairs contained in $p$. In particular, for $n = 4$, we have

$$\mathbb{E}[x_1 x_2 x_3 x_4] = \operatorname{cov}(x_1, x_2) \operatorname{cov}(x_3, x_4) + \operatorname{cov}(x_1, x_3) \operatorname{cov}(x_2, x_4) + \operatorname{cov}(x_1, x_4) \operatorname{cov}(x_2, x_3). \tag{A.2}$$

In physics, Isserlis' theorem is often known as Wick's probability theorem [2].

### A.2  Neumann series for matrix inverses near the identity

The Neumann series is the generalization of the geometric series to bounded linear operators, including square matrices. In particular, let $A$ be a $p \times p$ square matrix. Then, we have

$$(I_p - A)^{-1} = \sum_{k=0}^{\infty} A^k \tag{A.3}$$

provided that the series converges in the operator norm [3]. We will use this result without concern for rigorous convergence conditions, as we are interested only in asymptotic expansions.

### A.3  Series expansion of the log-determinant near the identity

Let $A$ be a $p \times p$ square matrix, and let $t$ be a small parameter. Then, we have

$$\log \det(I_p + tA) = \sum_{k=1}^{\infty} \frac{(-1)^{k+1}}{k} \operatorname{tr}(A^k) t^k \tag{A.4}$$

assuming that the series converges. We will not concern ourselves with rigorous convergence conditions, as we will use this expansion formally.

This result follows from the fact that

$$\frac{\partial^k}{\partial t^k} \log \det(I_p + tA) = (-1)^{k+1}(k-1)! \operatorname{tr}((I_p + tA)^{-k} A^k) \quad (k = 1, 2, \ldots). \tag{A.5}$$

The base case $k = 1$ is given by Jacobi's formula [3]:

$$\frac{\partial}{\partial t} \log \det(I_p + tA) = \text{tr}((I_p + tA)^{-1}A). \tag{A.6}$$

Then, using the identity

$$\frac{\partial}{\partial t}(I_p + tA)^{-1} = -(I_p + tA)^{-1}A(I_p + tA)^{-1} \tag{A.7}$$

and the fact that $A$ commutes with $(I_p + tA)^{-1}$, we find that the claim holds by induction. As $\log \det(I_p + tA)|_{t=0} = 0$, this implies the desired Maclaurin series.

# B  Perturbation theory for wide Bayesian neural networks with linear readout

In this appendix, we derive Conjecture 1. As outlined in the main text, we consider a depth-$d$ neural network $\mathbf{f} : \mathbb{R}^{n_0} \to \mathbb{R}^{n_d}$ with linear readout, written as

$$\mathbf{f}(\mathbf{x}; W^d, \mathcal{W}) = \frac{1}{\sqrt{n_{d-1}}} W^{(d)} \boldsymbol{\psi}(\mathbf{x}; \mathcal{W}) \tag{B.1}$$

in terms of the hidden layer feature map $\boldsymbol{\psi}(\cdot; \mathcal{W}) : \mathbb{R}^{n_0} \to \mathbb{R}^{n_{d-1}}$. The full set of trainable parameters is then $\Theta = \{W^{(d)}, \mathcal{W}\}$, where $\mathcal{W}$ is the set of feature map parameters. We assume isotropic Gaussian priors over these parameters, with, for instance,

$$W_{ij}^{(d)} \underset{\text{i.i.d.}}{\sim} \mathcal{N}(0, \sigma_d^2). \tag{B.2}$$

We fix an arbitrary training dataset $\mathcal{D} = \{(\mathbf{x}_\mu, \mathbf{y}_\mu)\}_{\mu=1}^p$ of $p$ examples, and use a Gaussian likelihood $p(\mathcal{D} \,|\, \Theta) \propto \exp(-\beta E)$, where

$$E(\Theta; \mathcal{D}) = \frac{1}{2} \sum_{\mu=1}^p \|\mathbf{f}(\mathbf{x}_\mu) - \mathbf{y}_\mu\|^2 \tag{B.3}$$

is a quadratic cost. We then introduce the Bayes posterior

$$p(\Theta \,|\, \mathcal{D}) = \frac{p(\mathcal{D} \,|\, \Theta)p(\Theta)}{p(\mathcal{D})}; \tag{B.4}$$

averages with respect to this distribution will be denoted by $\langle \cdot \rangle$.

We define the postactivation feature map kernel

$$K^{(d-1)}(\mathbf{x}, \mathbf{x}') \equiv n_{d-1}^{-1} \boldsymbol{\psi}(\mathbf{x}, \mathcal{W}) \cdot \boldsymbol{\psi}(\mathbf{x}', \mathcal{W}), \tag{B.5}$$

and write $[K^{(d-1)}]_{\mu\nu} \equiv K^{(d-1)}(\mathbf{x}_\mu, \mathbf{x}_\nu)$ for the kernel evaluated on the training set. For brevity, we will frequently abbreviate $K \equiv K^{(d-1)}$ throughout this appendix.

We denote expectation by $\mathbb{E}$, and prior expectation by $\mathbb{E}_\mathcal{W}$. We also introduce the joint cumulant operator $\mathbb{K}$ and its prior counterpart $\mathbb{K}_\mathcal{W}$. We will only require the second and third joint cumulants, which, for random variables $A$, $B$, and $C$, are given as

$$\mathbb{K}(A, B) = \mathbb{E}[(A - \mathbb{E}A)(B - \mathbb{E}B)] \tag{B.6}$$

and

$$\mathbb{K}(A, B, C) = \mathbb{E}[(A - \mathbb{E}A)(B - \mathbb{E}B)(C - \mathbb{E}C)], \tag{B.7}$$

respectively.

Our starting point is the partition function $Z$ of the Bayes posterior (B.4) for the network (B.1), including a source term for the (generically matrix-valued) observable $O$:

$$Z(J) = \mathbb{E}_{W^{(d)}} \mathbb{E}_\mathcal{W} \exp\left(-\beta E + \text{tr}(J^\top O)\right), \tag{B.8}$$

where $\mathcal{W}$ denotes all of the parameters except for the readout weight matrix $W^{(d)}$ and expectation is taken with respect to the Gaussian prior. The logarithm of the partition function is the posterior cumulant generating function of the observable $O$, with

$$\langle O \rangle = \frac{\delta \log Z}{\delta J}\bigg|_{J=0} \tag{B.9}$$

and covariance

$$\mathrm{cov}(O_{\rho\gamma}, O_{\omega\chi}) = \frac{\partial^2 \log Z}{\partial J_{\rho\gamma} \partial J_{\omega\chi}}\bigg|_{J=0}. \tag{B.10}$$

## B.1 Integrating out the readout layer

We first show that the readout layer can be integrated out exactly. As the source term is independent of $W^{(d)}$, Fubini's theorem yields

$$Z = \mathbb{E}_{\mathcal{W}}\left[\exp(\mathrm{tr}(J^\top O))\, \mathbb{E}_{W^{(d)}} \exp(-\beta E)\right]. \tag{B.11}$$

The expectation over $W^d$ is a Gaussian integral, hence it is easy to evaluate exactly:

$$\mathbb{E}_{W^{(d)}} \exp(-\beta E) \tag{B.12}$$

$$= \mathbb{E}_{W^{(d)}} \exp\left(-\frac{1}{2}\beta \sum_{\mu=1}^p \left\|\frac{1}{\sqrt{n_{d-1}}} W^{(d)} \boldsymbol{\psi}_\mu - \mathbf{y}_\mu\right\|^2\right) \tag{B.13}$$

$$= \exp\left(-\frac{1}{2}\beta\, \mathrm{tr}(Y^\top Y)\right)$$

$$\times \prod_{j=1}^{n_d}\left[\int \frac{d\mathbf{w}_j}{(2\pi\sigma_d^2)^{n_{d-1}/2}} \exp\left(-\frac{1}{2}\mathbf{w}_j^\top \left(\sigma_d^{-2} I_n + \frac{\beta}{n_{d-1}}\Psi^\top\Psi\right)\mathbf{w}_j + \frac{\beta}{\sqrt{n_{d-1}}}(Y^\top\Psi)_{j\cdot}\mathbf{w}_j\right)\right] \tag{B.14}$$

$$= \det\left(I_n + \frac{\beta\sigma_d^2}{n_{d-1}}\Psi^\top\Psi\right)^{-n_d/2}$$

$$\times \exp\left(\frac{1}{2}\frac{\beta^2\sigma_d^2}{n_{d-1}}\mathrm{tr}\left[Y^\top\Psi\left(I_n + \frac{\beta\sigma_d^2}{n_{d-1}}\Psi^\top\Psi\right)^{-1}\Psi^\top Y\right] - \frac{1}{2}\beta\,\mathrm{tr}(Y^\top Y)\right), \tag{B.15}$$

where we abbreviate $\boldsymbol{\psi}_\mu \equiv \boldsymbol{\psi}(\mathbf{x}_\mu; \mathcal{W})$ and introduce the matrices $\Psi_{\mu j} \equiv \psi_{\mu,j}$ and $Y_{\mu j} \equiv y_{\mu,j}$. Here, we have used the fact that the matrix $I_n + (\beta\sigma_d^2/n_{d-1})\Psi^\top\Psi$ is invertible at any finite temperature. By the Weinstein–Aronszajn identity [3],

$$\det\left(I_n + \frac{\beta\sigma_d^2}{n_{d-1}}\Psi^\top\Psi\right) = \det\left(I_p + \frac{\beta\sigma_d^2}{n_{d-1}}\Psi\Psi^\top\right) = \det(I_p + \beta\sigma_d^2 K), \tag{B.16}$$

where we introduce the (non-constant) kernel matrix

$$K = K^{(d-1)} \equiv \frac{1}{n_{d-1}}\Psi\Psi^\top; \tag{B.17}$$

as mentioned above, we abbreviate $K \equiv K^{(d-1)}$ for brevity. By the push-through identity [3],

$$\frac{1}{n_{d-1}}\Psi\left(I_n + \frac{\beta\sigma_d^2}{n_{d-1}}\Psi^\top\Psi\right)^{-1}\Psi^\top = \left(I_p + \frac{\beta\sigma_d^2}{n_{d-1}}\Psi\Psi^\top\right)^{-1}\frac{1}{n_{d-1}}\Psi\Psi^\top = (I_p + \beta\sigma_d^2 K)^{-1}K, \tag{B.18}$$

hence, using the cyclic property of the trace,

$$\frac{1}{2}\frac{\beta^2\sigma_d^2}{n_{d-1}}\mathrm{tr}\left[Y^\top\Psi\left(I_n + \frac{\beta\sigma_d^2}{n_{d-1}}\Psi^\top\Psi\right)^{-1}\Psi^\top Y\right] - \frac{1}{2}\beta\,\mathrm{tr}(Y^\top Y)$$

$$= \frac{1}{2}\beta n_d\,\mathrm{tr}\left[\left(\beta\sigma_d^2(I_p + \beta\sigma_d^2 K)^{-1}K - I_p\right)G_{yy}\right] \tag{B.19}$$

$$= -\frac{1}{2}\beta n_d\,\mathrm{tr}[(I_p + \beta\sigma_d^2 K)^{-1}G_{yy}], \tag{B.20}$$

S4

where we have defined the normalized Gram matrix of the outputs

$$G_{yy} \equiv \frac{1}{n_d} Y Y^\top \tag{B.21}$$

and noticed that

$$I_p - \beta \sigma_d^2 (I_p + \beta \sigma_d^2 K)^{-1} K = (I_p + \beta \sigma_d^2 K)^{-1}. \tag{B.22}$$

Therefore, we conclude that

$$Z = \mathbb{E}_{\mathcal{W}} \exp \left[ \mathrm{tr}(J^\top O) - \frac{n_d}{2} \left( \beta \, \mathrm{tr}[(I_p + \beta \sigma_d^2 K)^{-1} G_{yy}] + \log \det(I_p + \beta \sigma_d^2 K) \right) \right] \tag{B.23}$$

at any width.

## B.2 Perturbative expansion

We now consider how this expression behaves in the large-width limit. We assume that this limit is well-defined in the sense that the readout kernel $K$ tends in probability to the constant GP kernel $K_\infty$ [4–7], and that the observable $O$ similarly tends to a deterministic limit $O_\infty$. Then, we formally write $K$ and $O$ as their infinite-width limits plus corrections which are small at large hidden layer widths:

$$K = K_\infty + \lambda \, \delta K, \tag{B.24}$$
$$O = O_\infty + \lambda \, \delta O, \tag{B.25}$$

where the parameter $\lambda$ is used to track powers of the small deviations.

We first expand the term resulting from integrating out the readout layer into its infinite-width limit and a finite-width correction. We define the constant matrix

$$\Gamma \equiv K_\infty + \frac{1}{\beta \sigma_d^2} I_p, \tag{B.26}$$

which is invertible at any finite temperature. Then, by the Woodbury identity [3], we have,

$$\beta \sigma_d^2 (I_p + \beta \sigma_d^2 K)^{-1} = (\Gamma + \lambda \delta K)^{-1} = \Gamma^{-1} - \lambda \Gamma^{-1} \delta K (\Gamma + \lambda \delta K)^{-1} \tag{B.27}$$

and, similarly,

$$\log \det(I_p + \beta \sigma_d^2 K) = \log \det(\beta \sigma_d^2 \Gamma) + \log \det(I_p + \lambda \Gamma^{-1} \delta K). \tag{B.28}$$

Noting that that both $\lambda \Gamma^{-1} \delta K (\Gamma + \lambda \delta K)^{-1}$ and $\log \det(I_p + \lambda \Gamma^{-1} \delta K)$ are $\mathcal{O}(\lambda)$, we expand the logarithm of the partition function as

$$\log Z = \log Z_\infty + \mathrm{tr}(J^\top O_\infty) + \log \mathbb{E}_{\mathcal{W}} \exp[\lambda \, \mathrm{tr}(J^\top \delta O) + \lambda \Omega], \tag{B.29}$$

where

$$Z_\infty \equiv \det(\beta \sigma_d^2 \Gamma)^{-n_d/2} \exp \left( -\frac{1}{2} n_d \sigma_d^{-2} \, \mathrm{tr}(\Gamma^{-1} G_{yy}) \right) \tag{B.30}$$

is the GP partition function and

$$\Omega \equiv \frac{1}{2} n_d \, \mathrm{tr}[\sigma_d^{-2} \Gamma^{-1} \delta K (\Gamma + \lambda \delta K)^{-1} G_{yy}] - \frac{1}{2} n_d \lambda^{-1} \log \det(I_p + \lambda \Gamma^{-1} \delta K) \tag{B.31}$$

is the remainder. $\log \mathbb{E}_{\mathcal{W}} \exp[\lambda \, \mathrm{tr}(J^\top \delta O) + \lambda \Omega]$ has the form of a cumulant generating function, hence it has a formal series expansion in $\lambda$ given by

$$\begin{aligned}
\log \mathbb{E}_{\mathcal{W}} \exp[\lambda \, \mathrm{tr}(J^\top \delta O) + \lambda \Omega] &= \lambda \mathbb{E}_{\mathcal{W}}[\mathrm{tr}(J^\top \delta O) + \Omega] \\
&+ \frac{1}{2} \lambda^2 \, \mathbb{E}_{\mathcal{W}} \{ \mathrm{tr}[J^\top (\delta O - \mathbb{E}_{\mathcal{W}} \delta O)] + \Omega - \mathbb{E}_{\mathcal{W}} \Omega \}^2 \\
&+ \frac{1}{6} \lambda^3 \mathbb{E}_{\mathcal{W}} \{ \mathrm{tr}[J^\top (\delta O - \mathbb{E}_{\mathcal{W}} \delta O)] + \Omega - \mathbb{E}_{\mathcal{W}} \Omega \}^3 \\
&+ \mathcal{O}(\lambda^4).
\end{aligned} \tag{B.32}$$

We can then see that the $k$-th cumulant is $\mathcal{O}(J^k)$, hence the $k$-th posterior cumulant of $O$ will be $\mathcal{O}(\lambda^k)$. Specifically, we can read off the posterior mean

$$\langle O \rangle = O_\infty + \lambda \mathbb{E}_{\mathcal{W}} \delta O + \lambda^2 \mathbb{K}_{\mathcal{W}}(\delta O, \Omega) + \frac{1}{2} \lambda^3 \mathbb{K}_{\mathcal{W}}(\delta O, \Omega, \Omega) + \mathcal{O}(\lambda^4). \tag{B.33}$$

and covariance

$$\mathrm{cov}(O_{\rho\gamma}, O_{\omega\chi}) = \lambda^2 \mathbb{K}_{\mathcal{W}}(\delta O_{\rho\gamma}, \delta O_{\omega\chi}) + \lambda^3 \mathbb{K}_{\mathcal{W}}(\delta O_{\rho\gamma}, \delta O_{\omega\chi}, \Omega) + \mathcal{O}(\lambda^4). \tag{B.34}$$

To make further progress, we expand $\Omega$ in powers of $\lambda$. Using the Neumann series for the matrix inverse (see Appendix A), we have

$$(\Gamma + \lambda \delta K)^{-1} = \Gamma^{-1} - \lambda \Gamma^{-1} \delta K \Gamma^{-1} + \mathcal{O}(\lambda^2), \tag{B.35}$$

and, using the series expansion of the log-determinant near the identity (see Appendix A), we have

$$\lambda^{-1} \log \det(I_p + \lambda \Gamma^{-1} \delta K) = \mathrm{tr}(\Gamma^{-1} \delta K) - \frac{1}{2} \lambda \, \mathrm{tr}(\Gamma^{-1} \delta K \Gamma^{-1} \delta K) + \mathcal{O}(\lambda^2). \tag{B.36}$$

This yields

$$\begin{aligned}
\Omega = {}& \frac{n_d}{2} \mathrm{tr}[(\sigma_d^{-2} \Gamma^{-1} G_{yy} \Gamma^{-1} - \Gamma^{-1}) \delta K] \\
& - \frac{n_d}{2} \lambda \, \mathrm{tr}\left[ \left( \sigma_d^{-2} \Gamma^{-1} G_{yy} \Gamma^{-1} - \frac{1}{2} \Gamma^{-1} \right) \delta K \Gamma^{-1} \delta K \right] \\
& + \mathcal{O}(\lambda^2).
\end{aligned} \tag{B.37}$$

The leading term is simple because it is linear in $\delta K$. Then, keeping only the leading non-trivial corrections and recognizing that

$$O_\infty + \lambda \mathbb{E}_{\mathcal{W}} \delta O = \mathbb{E}_{\mathcal{W}} O, \tag{B.38}$$

$$\lambda^2 \mathbb{K}_{\mathcal{W}}(\delta O, \delta K_{\mu\nu}) = \mathbb{K}_{\mathcal{W}}(O, K_{\mu\nu}), \tag{B.39}$$

$$\lambda^2 \mathbb{K}_{\mathcal{W}}(\delta O_{\rho\gamma}, \delta O_{\omega\chi}) = \mathbb{K}_{\mathcal{W}}(O_{\rho\gamma}, O_{\omega\chi}), \quad \text{and} \tag{B.40}$$

$$\lambda^3 \mathbb{K}_{\mathcal{W}}(\delta O_{\rho\gamma}, \delta O_{\omega\chi}, \delta K_{\mu\nu}) = \mathbb{K}_{\mathcal{W}}(O_{\rho\gamma}, O_{\omega\chi}, K_{\mu\nu}), \tag{B.41}$$

we have

$$\langle O \rangle = \mathbb{E}_{\mathcal{W}} O + \frac{1}{2} n_d \sum_{\mu,\nu=1}^{p} (\sigma_d^{-2} \Gamma^{-1} G_{yy} \Gamma^{-1} - \Gamma^{-1})_{\mu\nu} \mathbb{K}_{\mathcal{W}}(O, K_{\mu\nu}) + \mathcal{O}(\lambda^3) \tag{B.42}$$

and

$$\begin{aligned}
\mathrm{cov}(O_{\rho\gamma}, O_{\omega\chi}) = {}& \mathbb{K}_{\mathcal{W}}(O_{\rho\gamma}, O_{\omega\chi}) \\
& + \frac{1}{2} n_d \sum_{\mu,\nu=1}^{p} (\sigma_d^{-2} \Gamma^{-1} G_{yy} \Gamma^{-1} - \Gamma^{-1})_{\mu\nu} \mathbb{K}_{\mathcal{W}}(O_{\rho\gamma}, O_{\omega\chi}, K_{\mu\nu}) \\
& + \mathcal{O}(\lambda^4).
\end{aligned} \tag{B.43}$$

Restoring the layer indices to $K = K^{(d-1)}$, the above result for $\langle O \rangle$ yields the expression given in the main text. From the structure of these expressions, we can see that higher-order terms (in $\lambda$) will involve higher joint cumulants of the kernel deviations $\delta K^{(\ell)}$, which can in turn be converted into joint cumulants of the kernels $K^{(\ell)}$. Therefore, to show that the perturbative expansion yields a valid asymptotic series , one would need to show that these joint cumulants themselves have asymptotic series expansions at large width, with leading terms that are successively suppressed by powers of $n^{-1}$.

## C  Explicit covariance computations in deep linear networks

In this appendix, we detail how to compute the prior covariances appearing in (B.42) for the hidden layer kernels of deep linear fully-connected and convolutional networks.

## C.1 Fully-connected linear networks

In this brief subsection, we provide a self-contained derivation of the behavior of the prior cumulants of the kernels of a deep fully-connected linear network with no bias terms. This is a special case of Yaida [8]'s results, and provides some intuition for his results on general MLPs. As in the main text, we consider a network with activations $\mathbf{h}^{(\ell)} \in \mathbb{R}^{n_\ell}$ recursively defined as

$$\mathbf{h}^{(\ell)} = n_{\ell-1}^{-1/2} W^{(\ell)} \mathbf{h}^{(\ell-1)} \qquad (\ell = 1, \ldots, d) \tag{C.1}$$

with base case $\mathbf{h}^{(0)} = \mathbf{x}$. We take the prior distribution over weights to be $[W^{(\ell)}]_{ij} \sim_{\text{i.i.d.}} \mathcal{N}(0, \sigma_\ell^{(2)})$, and define the hidden layer kernels $[K^{(\ell)}]_{\mu\nu} \equiv n_\ell^{-1} \mathbf{h}_\mu^{(\ell)} \cdot \mathbf{h}_\nu^{(\ell)}$ for $\ell = 1, \ldots, d-1$. Then, we have

$$\mathbb{E}_{\mathcal{W}} K_{\mu\nu}^{(\ell)} = \frac{1}{n_\ell \cdots n_0} \mathbb{E}_{\mathcal{W}} \mathbf{x}_\mu^\top (W^{(1)})^\top \cdots (W^{(\ell)})^\top W^{(\ell)} \cdots W^{(1)} \mathbf{x}_\nu \tag{C.2}$$

$$= \sigma_1^2 \cdots \sigma_\ell^2 \frac{\mathbf{x}_\mu \cdot \mathbf{x}_\nu}{n_0} \tag{C.3}$$

$$= [K_\infty^{(\ell)}]_{\mu\nu} \tag{C.4}$$

at any width, as $\mathbb{E}_{W^{(\ell)}} (W^{(\ell)})^\top W^{(\ell)} / n_\ell = \sigma_\ell^2 I_{n_{\ell-1}}$. We now consider the second moments of the kernels. We first note that

$$\mathbb{E}_{\mathcal{W}} K_{\mu\nu}^{(\ell)} K_{\rho\lambda}^{(\ell+\tau)} = \sigma_{\ell+\tau}^2 \cdots \sigma_{\ell+1}^2 \mathbb{E}_{\mathcal{W}} K_{\mu\nu}^{(\ell)} K_{\rho\lambda}^{(\ell)} \tag{C.5}$$

for any $\tau \geq 1$. By Isserlis' theorem (see Appendix A), we have

$$\mathbb{E}_{W^{(\ell)}} W_{ik}^{(\ell)} W_{il}^{(\ell)} W_{jm}^{(\ell)} W_{jr}^{(\ell)} = \sigma_\ell^4 \delta_{ij} (\delta_{km}\delta_{lr} + \delta_{kr}\delta_{lm}) + \sigma_\ell^4 \delta_{kl}\delta_{mr}, \tag{C.6}$$

hence we have the exact recursion

$$\begin{aligned}
\mathbb{E}_{\mathcal{W}} K_{\mu\nu}^{(\ell)} K_{\rho\lambda}^{(\ell)} = \frac{1}{(n_\ell \cdots n_0)^2} \mathbb{E}_{\mathcal{W}} \sum_{i,j=1}^{n_\ell} \sum_{k,l,m,r=1}^{n_{\ell-1}} & W_{ik}^{(\ell)} W_{il}^{(\ell)} W_{jm}^{(\ell)} W_{jr}^{(\ell)} \\
& \times [W^{(\ell-1)} \cdots W^{(1)} \mathbf{x}_\mu]_k [W^{(\ell-1)} \cdots W^{(1)} \mathbf{x}_\nu]_l \\
& \times [W^{(\ell-1)} \cdots W^{(1)} \mathbf{x}_\rho]_m [W^{(\ell-1)} \cdots W^{(1)} \mathbf{x}_\lambda]_r
\end{aligned} \tag{C.7}$$

$$= \sigma_\ell^4 \mathbb{E}_{\mathcal{W}} K_{\mu\nu}^{(\ell-1)} K_{\rho\lambda}^{(\ell-1)} + \frac{1}{n_\ell} \sigma_\ell^4 \left( \mathbb{E}_{\mathcal{W}} K_{\mu\rho}^{(\ell-1)} K_{\nu\lambda}^{(\ell-1)} + \mathbb{E}_{\mathcal{W}} K_{\mu\lambda}^{(\ell-1)} K_{\nu\rho}^{(\ell-1)} \right) \tag{C.8}$$

with base case

$$\mathbb{E}_{\mathcal{W}} K_{\mu\nu}^{(1)} K_{\rho\lambda}^{(1)} = \frac{1}{(n_1 n_0)^2} \sum_{i,j=1}^{n_1} \sum_{k,l,m,r=1}^{n_0} \mathbb{E}_{\mathcal{W}} W_{ik}^{(1)} W_{il}^{(1)} W_{jm}^{(1)} W_{jr}^{(1)} x_{\mu,k} x_{\nu,l} x_{\rho,m} x_{\lambda,r} \tag{C.9}$$

$$= \sigma_1^4 \frac{\mathbf{x}_\mu \cdot \mathbf{x}_\nu}{n_0} \frac{\mathbf{x}_\rho \cdot \mathbf{x}_\lambda}{n_0} + \frac{1}{n_1} \sigma_1^4 \left( \frac{\mathbf{x}_\mu \cdot \mathbf{x}_\rho}{n_0} \frac{\mathbf{x}_\nu \cdot \mathbf{x}_\lambda}{n_0} + \frac{\mathbf{x}_\mu \cdot \mathbf{x}_\lambda}{n_0} \frac{\mathbf{x}_\nu \cdot \mathbf{x}_\rho}{n_0} \right) \tag{C.10}$$

$$= [K_\infty^{(1)}]_{\mu\nu} [K_\infty^{(1)}]_{\rho\lambda} + \frac{1}{n_1} \left( [K_\infty^{(1)}]_{\mu\rho} [K_\infty^{(1)}]_{\nu\lambda} + [K_\infty^{(1)}]_{\mu\lambda} [K_\infty^{(1)}]_{\nu\rho} \right) \tag{C.11}$$

for the second moments of the kernels at each layer. This recurrence relation is in principle exactly solvable for any finite width, but we are interested only in its leading-order behavior at large widths. In particular, we can read off that

$$\begin{aligned}
\text{cov}_{\mathcal{W}}(K_{\mu\nu}^{(\ell)}, K_{\rho\lambda}^{(\ell+\tau)}) = \sigma_{\ell+\tau}^2 \cdots \sigma_{\ell+1}^2 & \left( \sum_{\ell'=1}^\ell \frac{1}{n_{\ell'}} \right) \left( [K_\infty^{(\ell)}]_{\mu\rho} [K_\infty^{(\ell)}]_{\nu\lambda} + [K_\infty^{(\ell)}]_{\mu\lambda} [K_\infty^{(\ell)}]_{\nu\rho} \right) \\
& + \mathcal{O}(n^{-2}).
\end{aligned} \tag{C.12}$$

Moreover, one can see by Isserlis' theorem that the third and higher cumulants will be $\mathcal{O}(n^{-2})$. Substituting this result into (B.42) with the hidden layer kernel as the observable of interest, we obtain the expression (8) given in the main text.

## C.2 Convolutional linear networks

In this subsection, we derive the prior cumulants required to compute corrections to the average feature kernels of deep convolutional linear networks. As described in the main text, following the setup of Novak et al. [6] and Xiao et al. [9], we consider a network consisting of $d-1$ linear convolutional layers followed by a fully-connected linear readout layer. For simplicity, we assume circular padding and no internal pooling. As discussed in Novak et al. [6], this setup could be easily extended to other padding strategies, strided convolutions, and average pooling in intermediate layers.

We write the activations at the $\ell$-th hidden layer as $h_{i,\mathfrak{a}}^{(\ell)}$, where $i$ indexes the $n_\ell$ channels of the layer and $\mathfrak{a}$ is a $q$-dimensional spatial multi-index. We take the filters to be of size $(2k+1) \times \cdots \times (2k+1)$ in all convolutional layers; the extension to differently-sized filters would be straightforward but notationally cumbersome. The ranges of all spatial summations will be implied.

The hidden layer activations are then defined through the recurrence

$$h_{i,\mathfrak{a}}^{(\ell)}(x) = \frac{1}{\sqrt{n_{\ell-1}}} \sum_{j=1}^{n_{\ell-1}} \sum_{\mathfrak{b}} w_{ij,\mathfrak{b}}^{(\ell)} h_{j,\mathfrak{a}+\mathfrak{b}}^{(\ell-1)}(x) \tag{C.13}$$

with base case $h_{i,\mathfrak{a}}^{(0)}(x) = x_{i,\mathfrak{a}}$. We fix the prior distribution of the filter elements to be

$$w_{ij,\mathfrak{a}}^{(\ell)} \underset{\text{i.i.d.}}{\sim} \mathcal{N}(0, \sigma_\ell^2 v_\mathfrak{a}), \tag{C.14}$$

where $v_\mathfrak{a} > 0$ is a weighting factor that sets the fraction of receptive field variance at location $\mathfrak{a}$ (and is thus subject to the constraint $\sum_\mathfrak{a} v_\mathfrak{a} = 1$). For inputs $[x_\mu]_{i,\mathfrak{a}}$ and $[x_\nu]_{i,\mathfrak{a}}$, we introduce the hidden layer kernels

$$K_{\mu\nu,\mathfrak{a}\mathfrak{b}}^{(\ell)} \equiv \frac{1}{n_\ell} \sum_{i=1}^{n_\ell} h_{i,\mathfrak{a}}^{(\ell)}(x_\mu) h_{i,\mathfrak{b}}^{(\ell)}(x_\nu). \tag{C.15}$$

We will first compute the prior mean and covariance of these four-indexed kernels, and then address how to handle readout across space.

As shown by Xiao et al. [9], the prior mean obeys the recurrence

$$\mathbb{E}_\mathcal{W} K_{\mu\nu,\mathfrak{a}\mathfrak{b}}^{(\ell)}$$

$$= \mathbb{E}_{W^{(1)}\dots W^{(\ell-1)}} \frac{1}{n_\ell n_{\ell-1}} \sum_{i=1}^{n_\ell} \sum_{j,j'=1}^{n_{\ell-1}} \sum_{\mathfrak{c},\mathfrak{d}} h_{j,\mathfrak{a}+\mathfrak{c}}^{(\ell-1)}(x_\mu) h_{j',\mathfrak{b}+\mathfrak{d}}^{(\ell-1)}(x_\nu) \mathbb{E}_{W^{(\ell)}} w_{ij,\mathfrak{c}}^{(\ell)} w_{ij',\mathfrak{d}}^{(\ell)} \tag{C.16}$$

$$= \sigma_\ell^2 \mathbb{E}_{W^{(1)}\dots W^{(\ell-1)}} \sum_\mathfrak{c} v_\mathfrak{c} \frac{1}{n_{\ell-1}} \sum_{j=1}^{n_{\ell-1}} h_{j,\mathfrak{a}+\mathfrak{c}}^{(\ell-1)}(x_\mu) h_{j,\mathfrak{b}+\mathfrak{c}}^{(\ell-1)}(x_\nu) \tag{C.17}$$

$$= \sigma_\ell^2 \sum_\mathfrak{c} v_\mathfrak{c} \mathbb{E}_\mathcal{W} K_{\mu\nu,(\mathfrak{a}+\mathfrak{c})(\mathfrak{b}+\mathfrak{c})}^{(\ell-1)} \tag{C.18}$$

with base case

$$\mathbb{E}_\mathcal{W} K_{\mu\nu,\mathfrak{a}\mathfrak{b}}^{(1)} = \sigma_1^2 \sum_\mathfrak{c} v_\mathfrak{c} [G_{xx}]_{\mu\nu,(\mathfrak{a}+\mathfrak{c})(\mathfrak{b}+\mathfrak{c})} \tag{C.19}$$

for

$$[G_{xx}]_{\mu\nu,\mathfrak{a}\mathfrak{b}} \equiv \frac{1}{n_0} \sum_{i=1}^{n_0} [x_\mu]_{i,\mathfrak{a}} [x_\nu]_{i,\mathfrak{b}}. \tag{C.20}$$

This recurrence yields

$$\mathbb{E}_\mathcal{W} K_{\mu\nu,\mathfrak{a}\mathfrak{b}}^{(\ell)} = \sigma_1^2 \cdots \sigma_\ell^2 \sum_{\mathfrak{c}_1,\dots,\mathfrak{c}_\ell} v_{\mathfrak{c}_1} \cdots v_{\mathfrak{c}_\ell} [G_{xx}]_{\mu\nu,(\mathfrak{a}+\mathfrak{c}_1+\cdots+\mathfrak{c}_\ell)(\mathfrak{b}+\mathfrak{c}_1+\cdots+\mathfrak{c}_\ell)}. \tag{C.21}$$

Moreover, as in the fully-connected case considered in the preceding section, we have

$$[K_\infty^{(\ell)}]_{\mu\nu,\mathfrak{a}\mathfrak{b}} = \mathbb{E}_\mathcal{W} K_{\mu\nu,\mathfrak{a}\mathfrak{b}}^{(\ell)} \tag{C.22}$$

at any width.

We now consider the prior covariance of the kernels of two different hidden layers $\ell$ and $\ell + \tau$. As the weight prior factors across layers, we have

$$\mathbb{E}_{\mathcal{W}} K^{(\ell)}_{\mu\nu,\mathfrak{ab}} K^{(\ell+\tau)}_{\rho\lambda,\mathfrak{cd}} = \sigma^2_{\ell+1} \cdots \sigma^2_{\ell+\tau} \sum_{\mathfrak{e}_1,\ldots,\mathfrak{e}_\tau} v_{\mathfrak{e}_1} \cdots v_{\mathfrak{e}_\tau}$$
$$\times \mathbb{E}_{\mathcal{W}} K^{(\ell)}_{\mu\nu,\mathfrak{ab}} K^{(\ell)}_{\rho\lambda,(\mathfrak{c}+\mathfrak{e}_1+\cdots+\mathfrak{e}_\tau)(\mathfrak{d}+\mathfrak{e}_1+\cdots+\mathfrak{e}_\tau)}. \quad \text{(C.23)}$$

By Isserlis' theorem (see Appendix A),

$$\mathbb{E}_{W^{(\ell)}} w^{(\ell)}_{ij,\mathfrak{e}} w^{(\ell)}_{ij',\mathfrak{f}} w^{(\ell)}_{i'j'',\mathfrak{g}} w^{(\ell)}_{i'j''',\mathfrak{h}} = \sigma^4_\ell v_{\mathfrak{e}} v_{\mathfrak{g}} \delta_{jj'} \delta_{j''j'''} \delta_{\mathfrak{ef}} \delta_{\mathfrak{gh}}$$
$$+ \sigma^4_\ell v_{\mathfrak{e}} v_{\mathfrak{f}} \delta_{ii'} \delta_{jj''} \delta_{j'j'''} \delta_{\mathfrak{eg}} \delta_{\mathfrak{fh}}$$
$$+ \sigma^4_\ell v_{\mathfrak{e}} v_{\mathfrak{f}} \delta_{ii'} \delta_{jj'''} \delta_{j'j''} \delta_{\mathfrak{eh}} \delta_{\mathfrak{fg}}, \quad \text{(C.24)}$$

hence we have the recurrence

$$\mathbb{E}_{\mathcal{W}} K^{(\ell)}_{\mu\nu,\mathfrak{ab}} K^{(\ell)}_{\rho\lambda,\mathfrak{cd}}$$

$$= \mathbb{E}_{W^{(1)}\ldots W^{(\ell-1)}} \frac{1}{n^2_\ell n^2_{\ell-1}} \sum_{i,i'=1}^{n_\ell} \sum_{j,j',j'',j'''=1}^{n_{\ell-1}} \sum_{\mathfrak{e},\mathfrak{f},\mathfrak{g},\mathfrak{h}}$$
$$\times h^{(\ell-1)}_{j,\mathfrak{a}+\mathfrak{e}}(x_\mu) h^{(\ell-1)}_{j',\mathfrak{b}+\mathfrak{f}}(x_\nu) h^{(\ell-1)}_{j'',\mathfrak{c}+\mathfrak{g}}(x_\rho) h^{(\ell-1)}_{j''',\mathfrak{d}+\mathfrak{h}}(x_\lambda)$$
$$\times \mathbb{E}_{W'^{(\ell)}} w^{(\ell)}_{ij,\mathfrak{e}} w^{(\ell)}_{ij',\mathfrak{f}} w^{(\ell)}_{i'j'',\mathfrak{g}} w^{(\ell)}_{i'j''',\mathfrak{h}} \quad \text{(C.25)}$$

$$= \sigma^4_\ell \sum_{\mathfrak{e},\mathfrak{f}} v_{\mathfrak{e}} v_{\mathfrak{f}} \left[ \mathbb{E}_{\mathcal{W}} K^{(\ell-1)}_{\mu\nu,(\mathfrak{a}+\mathfrak{e})(\mathfrak{b}+\mathfrak{e})} K^{(\ell-1)}_{\rho\lambda,(\mathfrak{c}+\mathfrak{f})(\mathfrak{d}+\mathfrak{f})} \right.$$

$$+ \frac{1}{n_\ell} \mathbb{E}_{\mathcal{W}} K^{(\ell-1)}_{\mu\rho,(\mathfrak{a}+\mathfrak{e})(\mathfrak{c}+\mathfrak{e})} K^{(\ell-1)}_{\nu\lambda,(\mathfrak{b}+\mathfrak{f})(\mathfrak{d}+\mathfrak{f})}$$

$$\left. + \frac{1}{n_\ell} \mathbb{E}_{\mathcal{W}} K^{(\ell-1)}_{\mu\lambda,(\mathfrak{a}+\mathfrak{e})(\mathfrak{d}+\mathfrak{e})} K^{(\ell-1)}_{\nu\rho,(\mathfrak{a}+\mathfrak{f})(\mathfrak{c}+\mathfrak{f})} \right] \quad \text{(C.26)}$$

with base case

$$\mathbb{E}_{\mathcal{W}} K^{(1)}_{\mu\nu,\mathfrak{ab}} K^{(1)}_{\rho\lambda,\mathfrak{cd}} = \sigma^4_1 \sum_{\mathfrak{e},\mathfrak{f}} v_{\mathfrak{e}} v_{\mathfrak{f}} \left[ [G_{xx}]_{\mu\nu,(\mathfrak{a}+\mathfrak{e})(\mathfrak{b}+\mathfrak{e})} [G_{xx}]_{\rho\lambda,(\mathfrak{c}+\mathfrak{f})(\mathfrak{d}+\mathfrak{f})} \right.$$

$$+ \frac{1}{n_\ell} [G_{xx}]_{\mu\rho,(\mathfrak{a}+\mathfrak{e})(\mathfrak{c}+\mathfrak{e})} [G_{xx}]_{\nu\lambda,(\mathfrak{b}+\mathfrak{f})(\mathfrak{d}+\mathfrak{f})}$$

$$\left. + \frac{1}{n_\ell} [G_{xx}]_{\mu\lambda,(\mathfrak{a}+\mathfrak{e})(\mathfrak{d}+\mathfrak{e})} [G_{xx}]_{\nu\rho,(\mathfrak{a}+\mathfrak{f})(\mathfrak{c}+\mathfrak{f})} \right] \quad \text{(C.27)}$$

$$= [K^{(1)}_\infty]_{\mu\nu,\mathfrak{ab}} [K^{(1)}_\infty]_{\rho\lambda,\mathfrak{cd}}$$
$$+ \frac{1}{n_\ell} \left[ [K^{(1)}_\infty]_{\mu\rho,\mathfrak{ac}} [K^{(1)}_\infty]_{\nu\lambda,\mathfrak{bd}} + [K^{(1)}_\infty]_{\mu\lambda,\mathfrak{ad}} [K^{(1)}_\infty]_{\nu\rho,\mathfrak{bc}} \right] \quad \text{(C.28)}$$

for the second prior moments of the kernels. As in the fully-connected case, these recurrence relations could in principle be solved exactly, but we are only interested in their large-width behavior. Using the forward recurrence for the GP kernels, we can easily read off that

$$\mathrm{cov}_{\mathcal{W}}(K^{(\ell)}_{\mu\nu,\mathfrak{ab}}, K^{(\ell)}_{\rho\lambda,\mathfrak{cd}}) = \left( \sum_{\ell'=1}^{\ell} \frac{1}{n_{\ell'}} \right) \left( [K^{(\ell)}_\infty]_{\mu\rho,\mathfrak{ac}} [K^{(\ell)}_\infty]_{\nu\lambda,\mathfrak{bd}} + [K^{(\ell)}_\infty]_{\mu\lambda,\mathfrak{ad}} [K^{(\ell)}_\infty]_{\nu\rho,\mathfrak{bc}} \right)$$
$$+ \mathcal{O}(n^{-2}), \quad \text{(C.29)}$$

which can then be substituted into the desired cross-layer covariance:

$$\mathrm{cov}_{\mathcal{W}}(K^{(\ell)}_{\mu\nu,\mathfrak{ab}}, K^{(\ell+\tau)}_{\rho\lambda,\mathfrak{cd}}) = \sigma^2_{\ell+1} \cdots \sigma^2_{\ell+\tau} \sum_{\mathfrak{e}_1,\ldots,\mathfrak{e}_\tau} v_{\mathfrak{e}_1} \cdots v_{\mathfrak{e}_\tau}$$
$$\times \mathrm{cov}_{\mathcal{W}}(K^{(\ell)}_{\mu\nu,\mathfrak{ab}}, K^{(\ell)}_{\rho\lambda,(\mathfrak{c}+\mathfrak{e}_1+\cdots+\mathfrak{e}_\tau)(\mathfrak{d}+\mathfrak{e}_1+\cdots+\mathfrak{e}_\tau)}). \quad \text{(C.30)}$$

We now address the question of how to read out the convolutional layer activities across space. Following Novak et al. [6], we consider two strategies: vectorization and projection. With vectorization, the output of the final convolutional layer is flattened into a $n_{d-1}s$-dimensional vector before readout, i.e., $\psi_{i+s(\mathfrak{a}-1)}(x) = h_{i,\mathfrak{a}}^{(d-1)}(x)$ or $\psi_{n_d(i-1)+\mathfrak{a}}(x) = h_{i,\mathfrak{a}}^{(d-1)}(x)$. The two-index feature map kernel appearing in Conjecture 1 is then related to the four-index convolutional hidden layer kernel analyzed above via

$$K_{\mu\nu}^{(d-1)} = \frac{1}{s} \sum_{\mathfrak{a}} K_{\mu\nu,\mathfrak{a}\mathfrak{a}}^{(d-1)}. \tag{C.31}$$

With projection, the feature map is formed by contracting the final convolutional layer with a fixed vector $\mathbf{u}$, i.e.,

$$\psi_i(x) = \sum_{\mathfrak{a}} u_{\mathfrak{a}} h_{i,\mathfrak{a}}^{(d-1)}(x). \tag{C.32}$$

The feature map kernel is then given as

$$K_{\mu\nu}^{(d-1)} = \sum_{\mathfrak{a},\mathfrak{b}} u_{\mathfrak{a}} u_{\mathfrak{b}} K_{\mu\nu,\mathfrak{a}\mathfrak{b}}^{(d-1)}. \tag{C.33}$$

Examples of common projection readout strategies include global average pooling ($u_{\mathfrak{a}} = 1/s$) and single-pixel subsampling ($u_{\mathfrak{a}} = \delta_{\mathfrak{a}\mathfrak{c}}$ for some desired location $\mathfrak{c}$). These readout approaches endow the network with differing properties under spatial transformations; global average pooling has the particular property of making the output translation-invariant.

We now seek to simplify the resulting expression for the leading-order correction to the posterior mean of some four-index feature kernel $K_{\mu\nu,\mathfrak{a}\mathfrak{b}}^{(\ell)}$. Per Conjecture 1, the general form of this correction is

$$\frac{1}{2} n_d \sum_{\rho,\lambda=1}^{p} \Phi_{\rho\lambda} \operatorname{cov}_{\mathcal{W}}(K_{\mu\nu,\mathfrak{a}\mathfrak{b}}^{(\ell)}, K_{\rho\lambda}^{(d-1)}), \tag{C.34}$$

where we have defined $\Phi_{\rho\lambda} = [\sigma_d^{-2}\Gamma^{-1}G_{yy}\Gamma^{-1} - \Gamma^{-1}]_{\rho\lambda}$ for notational convenience. As elsewhere, $\Gamma \equiv K_\infty^{(d-1)} + \beta^{-1}\sigma_d^{-2}I_p$ for $K_\infty^{(d-1)}$ the two-index kernel determined by the chosen readout strategy. Depending on the chosen readout strategy, this general expression can be simplified dramatically. In particular, for vectorization or global average pooling, the correction does not depend on the particular form of $v_{\mathfrak{a}}$.

To show this for vectorization (the strategy used in our experiments), we substitute the definition of $K_\infty^{(d-1)}$ from (C.31) and the expression for the cross-layer kernel covariance from (C.30) into the general expression for the correction to obtain

$$\frac{n_d}{2s}\sigma_{\ell+1}^2 \cdots \sigma_{d-1}^2$$
$$\times \sum_{\rho,\lambda} \Phi_{\rho\lambda} \sum_{\mathfrak{e}_1,\cdots,\mathfrak{e}_{d-\ell-1}} v_{\mathfrak{e}_1} \cdots v_{\mathfrak{e}_{d-\ell-1}} \sum_{\mathfrak{c}} \operatorname{cov}_{\mathcal{W}}(K_{\mu\nu,\mathfrak{a}\mathfrak{b}}^{(\ell)}, K_{\rho\lambda,(\mathfrak{c}+\mathfrak{e}_1+\cdots+\mathfrak{e}_{d-\ell-1})(\mathfrak{c}+\mathfrak{e}_1+\cdots+\mathfrak{e}_{d-\ell-1})}^{(\ell)}). \tag{C.35}$$

Thanks to the periodic boundary conditions, the summation over $\mathfrak{c}$ is independent of the index shift $\mathfrak{e}_1 + \cdots + \mathfrak{e}_{d-\ell-1}$. Then, the sums over $\mathfrak{e}_1, \cdots, \mathfrak{e}_{d-\ell-1}$ factor, yielding

$$\frac{n_d}{2s}\sigma_{\ell+1}^2 \cdots \sigma_{d-1}^2 \sum_{\rho,\lambda=1}^{p} \Phi_{\rho\lambda} \sum_{\mathfrak{c}} \operatorname{cov}_{\mathcal{W}}(K_{\mu\nu,\mathfrak{a}\mathfrak{b}}^{(\ell)}, K_{\rho\lambda,\mathfrak{c}\mathfrak{c}}^{(\ell)}) \tag{C.36}$$

thanks to the normalization constraint $\sum_{\mathfrak{e}} v_{\mathfrak{e}} = 1$. We now notice that $\Phi_{\rho\lambda}$ is a symmetric matrix, and that the kernel remains invariant under the simultaneous exchange of indices $\rho \leftrightarrow \lambda$ and $\mathfrak{c} \leftrightarrow \mathfrak{d}$. Then, substituting in the expression for the same-layer kernel covariance (C.29), it is easy to show that the correction reduces to

$$\sigma_{\ell+1}^2 \cdots \sigma_{d-1}^2 \left( \sum_{\ell'=1}^{\ell} \frac{n_d}{n_{\ell'}} \right) \frac{1}{s} \sum_{\mathfrak{c}} \sum_{\rho,\lambda=1}^{p} [K_\infty^{(\ell)}]_{\mu\rho,\mathfrak{a}\mathfrak{c}} \Phi_{\rho\lambda} [K_\infty^{(\ell)}]_{\lambda\nu,\mathfrak{c}\mathfrak{b}}. \tag{C.37}$$

This yields the expression given in the main text.

For projection, an analogous simplification is possible in the case of global average pooling ($u_{\mathfrak{a}} = 1/s$). Substituting the definition of $K_\infty^{(d-1)}$ from (C.33) and expression for the cross-layer kernel covariance (C.30) into the correction, we have

$$\frac{n_d}{2s^2}\sigma_{\ell+1}^2 \cdots \sigma_{d-1}^2 \sum_{\rho,\lambda=1}^{p} \Phi_{\rho\lambda} \sum_{\mathfrak{c},\mathfrak{d}} \mathrm{cov}_{\mathcal{W}}(K_{\mu\nu,\mathfrak{a}\mathfrak{b}}^{(\ell)}, K_{\rho\lambda,\mathfrak{c}\mathfrak{d}}^{(\ell)}). \tag{C.38}$$

Substituting in the expression for the same-layer kernel covariance (C.29), it is again easy to show that the correction reduces to

$$\sigma_{\ell+1}^2 \cdots \sigma_{d-1}^2 \left(\sum_{\ell'=1}^{\ell} \frac{n_d}{n_{\ell'}}\right) \frac{1}{s^2} \sum_{\mathfrak{c},\mathfrak{d}} \sum_{\rho,\lambda=1}^{p} [K_\infty^{(\ell)}]_{\mu\rho,\mathfrak{a}\mathfrak{c}} \Phi_{\rho\lambda} [K_\infty^{(\ell)}]_{\lambda\nu,\mathfrak{d}\mathfrak{b}}. \tag{C.39}$$

For projection strategies other than global average pooling (more precisely, for strategies for which $u_{\mathfrak{a}}$ is not constant), the sum over indices in the cross-layer covariance is not independent of the shift, hence we cannot simplify the correction in a similar fashion. This can be seen explicitly when treating the case of single-pixel subsampling ($u_{\mathfrak{a}} = \delta_{\mathfrak{a}\mathfrak{c}}$ for some desired location $\mathfrak{c}$). In this case, the correction reduces to

$$\frac{n_d}{2}\sigma_{\ell+1}^2 \cdots \sigma_{d-1}^2$$
$$\times \sum_{\rho,\lambda} \Phi_{\rho\lambda} \sum_{\mathfrak{e}_1,\cdots,\mathfrak{e}_{d-\ell-1}} v_{\mathfrak{e}_1} \cdots v_{\mathfrak{e}_{d-\ell-1}} \, \mathrm{cov}_{\mathcal{W}}(K_{\mu\nu,\mathfrak{a}\mathfrak{b}}^{(\ell)}, K_{\rho\lambda,(\mathfrak{c}+\mathfrak{e}_1+\cdots+\mathfrak{e}_{d-\ell-1})(\mathfrak{c}+\mathfrak{e}_1+\cdots+\mathfrak{e}_{d-\ell-1})}^{(\ell)}).$$
$$\tag{C.40}$$

Unlike for vectorization or for projection using global average pooling, this expression is manifestly dependent on the form of $v_{\mathfrak{a}}$.

Naïvely, the computation of the corrections to the linear convolutional kernels requires the computation of $\mathrm{cov}_{\mathcal{W}}(K_{\mu\nu,\mathfrak{a}\mathfrak{b}}^{(\ell)}, K_{\rho\lambda,(\mathfrak{c}+\mathfrak{e}_1+\cdots+\mathfrak{e}_{d-\ell-1})(\mathfrak{d}+\mathfrak{e}_1+\cdots+\mathfrak{e}_{d-\ell-1})}^{(\ell)})$ for each index, which takes impractical amounts of compute time and storage. We only found it practical to compute the theoretical kernels in the special cases presented above.

## D  Direct computation of the average hidden layer kernels of a deep linear MLP

In this appendix, we provide a self-contained derivation of the average hidden layer kernels of a deep linear fully-connected network (MLP). This derivation relies upon neither the results of Appendices B and C nor those of Yaida [8].

### D.1  The cumulant generating function of learned features for a MLP

In this section, we briefly describe the full partition function of the Bayes posterior for a general fully connected network, or multi-layer perceptron (MLP), with no bias terms. An MLP $\mathbf{f} : \mathbb{R}^{n_0} \to \mathbb{R}^{n_d}$ with $d$ layers, no biases, and parameters $\Theta = \{W^{(\ell)}\}_{\ell=1}^{d}$ can be defined recursively in terms of its layer-wise preactivations $\mathbf{h}^{(\ell)} \in \mathbb{R}^{n_\ell}$ as

$$\mathbf{h}^{(0)} = \mathbf{x}, \tag{D.1}$$

$$\mathbf{h}^{(\ell)} = \frac{1}{\sqrt{n_{\ell-1}}} W^{(\ell)} \phi_{\ell-1}(\mathbf{h}^{(\ell-1)}) \quad (\ell = 1, \ldots, d), \tag{D.2}$$

$$\mathbf{f} = \phi_d(\mathbf{h}^{(d)}), \tag{D.3}$$

where the activation functions $\phi_\ell$ act elementwise. As always, we focus on networks with linear readout, i.e., $\phi_d(x) = x$, and assume Gaussian priors over the weights:

$$W_{ij}^{(\ell)} \underset{\text{i.i.d.}}{\sim} \mathcal{N}(0, \sigma_\ell^2). \tag{D.4}$$

S11

We enforce the definition of the network architecture via Fourier representations of the Dirac distribution, with $\mathbf{q}_\mu^{(\ell)}$ being the Lagrange multiplier that enforces the definition of the preactivation $\mathbf{h}_\mu^{(\ell)}$. Then, after integrating out the weights using the fact that the relevant integrals are Gaussian, this allows us to write the partition function as

$$Z = \int \prod_{\mu=1}^{p} \prod_{\ell=1}^{d} \frac{d\mathbf{h}_\mu^{(\ell)} d\mathbf{q}_\mu^{(\ell)}}{(2\pi)^{n_\ell}} \exp\left[ S(\{\mathbf{h}_\mu^{(\ell)}\}, \{\mathbf{q}_\mu^{(\ell)}\}) \right], \tag{D.5}$$

where the "effective action" for the preactivations and Lagrange multipliers is

$$S = -\frac{1}{2}\beta \sum_{\mu=1}^{p} \|\mathbf{h}_\mu^{(d)} - \mathbf{y}_\mu\|^2 + \sum_{\ell=1}^{d} \sum_{\mu=1}^{p} i\mathbf{q}_\mu^{(\ell)} \cdot \mathbf{h}_\mu^{(\ell)}$$

$$-\frac{1}{2} \sum_{\ell=1}^{d} \frac{\sigma_\ell^2}{n_{\ell-1}} \sum_{\mu,\nu=1}^{p} \mathbf{q}_\mu^{(\ell)} \cdot \mathbf{q}_\nu^{(\ell)} \phi_{\ell-1}(\mathbf{h}_\mu^{(\ell-1)}) \cdot \phi_{\ell-1}(\mathbf{h}_\nu^{(\ell-1)}). \tag{D.6}$$

As described in Appendix B, source terms can be added to the effective action to allow computation of various averages. For deep linear networks, it is convenient to scale the source terms by an overall factor of $-1/2$, for which we must correct when computing the averages:

$$S_{\mathrm{J}} = -\frac{1}{2} \sum_{\ell=1}^{d-1} \sum_{\mu,\nu=1}^{p} J_{\mu\nu}^{(\ell)} \phi_\ell(\mathbf{h}_\mu^{(\ell)}) \cdot \phi_\ell(\mathbf{h}_\nu^{(\ell)}). \tag{D.7}$$

For an MLP, our task is therefore to integrate out the preactivations and corresponding Lagrange multipliers. We will do so sequentially from the first layer to the last, keeping terms up to the desired order at each step, akin to the approach of Yaida [8]. So long as $n_d$ and $d$ are fixed and small relative to the width of the hidden layers, this is a consistent perturbative approach, as noted by Yaida [8].

## D.2 General form of the perturbative layer integrals for a deep linear network

In this section, we evaluate the general form of the integrals required to perturbatively marginalize out a given layer of a deep linear network to $\mathcal{O}(n^{-1})$. These integrals are generically of the form

$$I = \int \prod_{\mu=1}^{p} \frac{d\mathbf{h}_\mu d\mathbf{q}_\mu}{(2\pi)^{n_2}} \exp\left( \sum_{\mu=1}^{p} i\mathbf{q}_\mu \cdot \mathbf{h}_\mu - \frac{1}{2} \sum_{\mu,\nu=1}^{p} G_{\mu\nu}(\mathbf{q}_\mu \cdot \mathbf{q}_\nu) + \sum_{\mu=1}^{p} \mathbf{j}_\mu \cdot \mathbf{h}_\mu \right.$$

$$-\frac{1}{2}\frac{1}{n_2} \sum_{\mu,\nu=1}^{p} A_{\mu\nu}(\mathbf{h}_\mu \cdot \mathbf{h}_\nu)$$

$$+\frac{1}{4}\frac{g}{n_1} \sum_{\mu,\nu,\rho,\lambda=1}^{p} G_{\mu\nu}(\mathbf{q}_\nu \cdot \mathbf{q}_\rho) G_{\rho\lambda}(\mathbf{q}_\lambda \cdot \mathbf{q}_\mu)$$

$$\left. +\frac{1}{2}\frac{1}{n_1} \sum_{\mu,\nu=1}^{p} B_{\mu\nu}(\mathbf{q}_\mu \cdot \mathbf{q}_\nu) \right), \tag{D.8}$$

where $\mathbf{h}_\mu, \mathbf{q}_\mu \in \mathbb{R}^{n_2}$. Here, $G$ is a positive semidefinite matrix, while $A$ and $B$ are symmetric matrices that need not be positive semidefinite. Furthermore, $\mathbf{j}_\mu$ is some source, while $g$ is a coupling constant. We will first evaluate this integral up to terms of $\mathcal{O}(n_1^{-1})$ for $n_1 \gg 1$, assuming that $G$, $A$, $B$, $\mathbf{j}_\mu$, and $g$ are $\mathcal{O}(1)$ functions of $n_1$, and then evaluate it up to terms of $\mathcal{O}(n_1^{-1}, n_2^{-1})$ for $n_1, n_2 \gg 1$, assuming that $G$, $A$, $B$, $\mathbf{j}_\mu$, and $g$ are also $\mathcal{O}(1)$ functions of $n_2$.

We will proceed by evaluating the integrals for $G$ invertible, and then infer the general case by a continuity argument. We treat the quartic term perturbatively, and all other terms directly. Writing

$$C \equiv G - \frac{1}{n_1}B, \tag{D.9}$$

the leading term in the integral over $\mathbf{q}_\mu$ is

$$\frac{1}{(2\pi)^{n_2 p/2} \det(C)^{n_2/2}} \exp\left( -\frac{1}{2} \sum_{\mu,\nu=1}^{p} C_{\mu\nu}^{-1}(\mathbf{h}_\mu \cdot \mathbf{h}_\nu) \right). \tag{D.10}$$

Multiplying and dividing by this quantity, we can compute the perturbative correction from the quartic term using the fact that $\mathbf{q}_\mu$ then behaves as a Gaussian random vector of mean $\bar{\mathbf{q}}_\mu = i \sum_{\nu=1}^{p} C_{\mu\nu}^{-1} \mathbf{h}_\nu$ and covariance $C_{\mu\nu}^{-1} I_{n_2}$. Denoting expectation with respect to this distribution as $\langle\!\langle \cdot \rangle\!\rangle_q$ and writing $\tilde{\mathbf{q}}_\mu \equiv \mathbf{q}_\mu - \bar{\mathbf{q}}_\mu$, Isserlis' theorem yields

$$\langle\!\langle (\mathbf{q}_\nu \cdot \mathbf{q}_\rho)(\mathbf{q}_\lambda \cdot \mathbf{q}_\mu) \rangle\!\rangle_q = \langle\!\langle ([\tilde{\mathbf{q}}_\nu + \bar{\mathbf{q}}_\nu] \cdot [\tilde{\mathbf{q}}_\rho + \bar{\mathbf{q}}_\rho])([\tilde{\mathbf{q}}_\lambda + \bar{\mathbf{q}}_\lambda] \cdot [\tilde{\mathbf{q}}_\mu + \bar{\mathbf{q}}_\mu]) \rangle\!\rangle_q \tag{D.11}$$

$$= \langle\!\langle (\tilde{\mathbf{q}}_\nu \cdot \tilde{\mathbf{q}}_\rho + \tilde{\mathbf{q}}_\nu \cdot \bar{\mathbf{q}}_\rho + \bar{\mathbf{q}}_\nu \cdot \tilde{\mathbf{q}}_\rho + \bar{\mathbf{q}}_\nu \cdot \bar{\mathbf{q}}_\rho)$$
$$\times (\tilde{\mathbf{q}}_\lambda \cdot \tilde{\mathbf{q}}_\mu + \tilde{\mathbf{q}}_\lambda \cdot \bar{\mathbf{q}}_\mu + \bar{\mathbf{q}}_\lambda \cdot \tilde{\mathbf{q}}_\mu + \bar{\mathbf{q}}_\lambda \cdot \bar{\mathbf{q}}_\mu) \rangle\!\rangle_q \tag{D.12}$$

$$= \langle\!\langle (\tilde{\mathbf{q}}_\nu \cdot \tilde{\mathbf{q}}_\rho)(\tilde{\mathbf{q}}_\lambda \cdot \tilde{\mathbf{q}}_\mu) \rangle\!\rangle_q + \langle\!\langle (\tilde{\mathbf{q}}_\nu \cdot \tilde{\mathbf{q}}_\rho) \rangle\!\rangle_q (\bar{\mathbf{q}}_\lambda \cdot \bar{\mathbf{q}}_\mu)$$
$$+ \langle\!\langle (\tilde{\mathbf{q}}_\nu \cdot \bar{\mathbf{q}}_\rho)(\tilde{\mathbf{q}}_\lambda \cdot \bar{\mathbf{q}}_\mu) \rangle\!\rangle_q + \langle\!\langle (\tilde{\mathbf{q}}_\nu \cdot \bar{\mathbf{q}}_\rho)(\bar{\mathbf{q}}_\lambda \cdot \tilde{\mathbf{q}}_\mu) \rangle\!\rangle_q$$
$$+ \langle\!\langle (\bar{\mathbf{q}}_\nu \cdot \tilde{\mathbf{q}}_\rho)(\tilde{\mathbf{q}}_\lambda \cdot \bar{\mathbf{q}}_\mu) \rangle\!\rangle_q + \langle\!\langle (\bar{\mathbf{q}}_\nu \cdot \tilde{\mathbf{q}}_\rho)(\bar{\mathbf{q}}_\lambda \cdot \tilde{\mathbf{q}}_\mu) \rangle\!\rangle_q$$
$$+ (\bar{\mathbf{q}}_\nu \cdot \bar{\mathbf{q}}_\rho) \langle\!\langle (\tilde{\mathbf{q}}_\lambda \cdot \tilde{\mathbf{q}}_\mu) \rangle\!\rangle_q + (\bar{\mathbf{q}}_\nu \cdot \bar{\mathbf{q}}_\rho)(\bar{\mathbf{q}}_\lambda \cdot \bar{\mathbf{q}}_\mu) \tag{D.13}$$

$$= n_2^2 C_{\nu\rho}^{-1} C_{\lambda\mu}^{-1} + n_2 C_{\nu\lambda}^{-1} C_{\rho\mu}^{-1} + n_2 C_{\nu\mu}^{-1} C_{\rho\lambda}^{-1} + n_2 C_{\nu\rho}^{-1} (\bar{\mathbf{q}}_\lambda \cdot \bar{\mathbf{q}}_\mu)$$
$$+ C_{\nu\lambda}^{-1} (\bar{\mathbf{q}}_\rho \cdot \bar{\mathbf{q}}_\mu) + C_{\nu\mu}^{-1} (\bar{\mathbf{q}}_\rho \cdot \bar{\mathbf{q}}_\lambda)$$
$$+ C_{\rho\lambda}^{-1} (\bar{\mathbf{q}}_\nu \cdot \bar{\mathbf{q}}_\mu) + C_{\rho\mu}^{-1} (\bar{\mathbf{q}}_\nu \cdot \bar{\mathbf{q}}_\lambda)$$
$$+ n_2 (\bar{\mathbf{q}}_\nu \cdot \bar{\mathbf{q}}_\rho) C_{\mu\lambda}^{-1} + (\bar{\mathbf{q}}_\nu \cdot \bar{\mathbf{q}}_\rho)(\bar{\mathbf{q}}_\lambda \cdot \bar{\mathbf{q}}_\mu). \tag{D.14}$$

Then, the quartic correction to the integral over $\mathbf{q}_\mu$ is proportional to

$$\sum_{\mu,\nu,\rho,\lambda=1}^{p} G_{\mu\nu} G_{\rho\lambda} \langle\!\langle (\mathbf{q}_\nu \cdot \mathbf{q}_\rho)(\mathbf{q}_\lambda \cdot \mathbf{q}_\mu) \rangle\!\rangle_q = n_2(n_2+1) \operatorname{tr}(GC^{-1}GC^{-1}) + n_2 \operatorname{tr}(GC^{-1})^2$$

$$- 2(n_2+1) \operatorname{tr}(GC^{-1}GC^{-1}HC^{-1})$$
$$- 2 \operatorname{tr}(GC^{-1}) \operatorname{tr}(GC^{-1}HC^{-1})$$
$$+ \operatorname{tr}(GC^{-1}HC^{-1}GC^{-1}HC^{-1}), \tag{D.15}$$

where we write $H_{\mu\nu} \equiv \mathbf{h}_\mu \cdot \mathbf{h}_\nu$.

We now must integrate over $\mathbf{h}_\mu$. The leading term is simply

$$\det(CD)^{-n_2/2} \exp\left( \frac{1}{2} \sum_{\mu,\nu=1}^{p} D_{\mu\nu}^{-1} J_{\mu\nu} \right) \tag{D.16}$$

where we have defined

$$D \equiv C^{-1} + \frac{1}{n_2} A. \tag{D.17}$$

and $J_{\mu\nu} \equiv \mathbf{j}_\mu \cdot \mathbf{j}_\nu$. Multiplying and dividing by this quantity, we can compute the perturbative correction from the quartic term using the fact that $\mathbf{h}_\mu$ then behaves as a Gaussian random vector of mean $\bar{\mathbf{h}}_\mu = \sum_{\nu=1}^{p} D_{\mu\nu}^{-1} \mathbf{j}_\nu$ and covariance $D_{\mu\nu}^{-1} I_{n_2}$. We denote expectations with respect to this distribution by $\langle\!\langle \cdot \rangle\!\rangle_h$, and define $\tilde{\mathbf{h}}_\mu \equiv \mathbf{h}_\mu - \bar{\mathbf{h}}_\mu$. Then, we have

$$\langle\!\langle H_{\mu\nu} \rangle\!\rangle_h = \langle\!\langle \mathbf{h}_\mu \cdot \mathbf{h}_\nu \rangle\!\rangle_h = \bar{\mathbf{h}}_\mu \cdot \bar{\mathbf{h}}_\nu + n_2 D_{\mu\nu}^{-1}, \tag{D.18}$$

and, by analogy to the corresponding four-point average for $\mathbf{q}_\mu$,

$$\langle\!\langle (\mathbf{h}_\nu \cdot \mathbf{h}_\rho)(\mathbf{h}_\lambda \cdot \mathbf{h}_\mu) \rangle\!\rangle_h = n_2^2 D_{\nu\rho}^{-1} D_{\lambda\mu}^{-1} + n_2 D_{\nu\lambda}^{-1} D_{\rho\mu}^{-1} + n_2 D_{\nu\mu}^{-1} D_{\rho\lambda}^{-1} + n_2 D_{\nu\rho}^{-1} (\bar{\mathbf{h}}_\lambda \cdot \bar{\mathbf{h}}_\mu)$$
$$+ D_{\nu\lambda}^{-1} (\bar{\mathbf{h}}_\rho \cdot \bar{\mathbf{h}}_\mu) + D_{\nu\mu}^{-1} (\bar{\mathbf{h}}_\rho \cdot \bar{\mathbf{h}}_\lambda)$$
$$+ D_{\rho\lambda}^{-1} (\bar{\mathbf{h}}_\nu \cdot \bar{\mathbf{h}}_\mu) + D_{\rho\mu}^{-1} (\bar{\mathbf{h}}_\nu \cdot \bar{\mathbf{h}}_\lambda)$$
$$+ n_2 (\bar{\mathbf{h}}_\nu \cdot \bar{\mathbf{h}}_\rho) D_{\mu\lambda}^{-1} + (\bar{\mathbf{h}}_\nu \cdot \bar{\mathbf{h}}_\rho)(\bar{\mathbf{h}}_\lambda \cdot \bar{\mathbf{h}}_\mu). \tag{D.19}$$

Then, the correction to the integral over $\mathbf{h}_\mu$ is proportional to

$$\sum_{\mu,\nu,\rho,\lambda=1}^{p} G_{\mu\nu}G_{\rho\lambda}\langle\!\langle(\mathbf{q}_\nu \cdot \mathbf{q}_\rho)(\mathbf{q}_\lambda \cdot \mathbf{q}_\mu)\rangle\!\rangle = n_2(n_2+1)\operatorname{tr}(GC^{-1}GC^{-1}) + n_2\operatorname{tr}(GC^{-1})^2$$

$$- 2(n_2+1)\operatorname{tr}(GC^{-1}GC^{-1}D^{-1}JD^{-1}C^{-1})$$
$$- 2n_2(n_2+1)\operatorname{tr}(GC^{-1}GC^{-1}D^{-1}C^{-1})$$
$$- 2\operatorname{tr}(GC^{-1})\operatorname{tr}(GC^{-1}D^{-1}JD^{-1}C^{-1})$$
$$- 2n_2\operatorname{tr}(GC^{-1})\operatorname{tr}(GC^{-1}D^{-1}C^{-1})$$
$$+ n_2(n_2+1)\operatorname{tr}(C^{-1}GC^{-1}D^{-1}C^{-1}GC^{-1}D^{-1})$$
$$+ n_2\operatorname{tr}(C^{-1}GC^{-1}D^{-1})^2$$
$$+ 2(n_2+1)\operatorname{tr}(C^{-1}GC^{-1}D^{-1}C^{-1}GC^{-1}D^{-1}JD^{-1})$$
$$+ 2\operatorname{tr}(C^{-1}GC^{-1}D^{-1})\operatorname{tr}(C^{-1}GC^{-1}D^{-1}JD^{-1})$$
$$+ \operatorname{tr}(C^{-1}GC^{-1}D^{-1}JD^{-1}C^{-1}GC^{-1}D^{-1}JD^{-1}),$$
$$\tag{D.20}$$

where we have noted that

$$\langle\!\langle\operatorname{tr}(GC^{-1}HC^{-1}GC^{-1}HC^{-1})\rangle\!\rangle_h$$

$$= \sum_{\mu,\nu,\rho,\lambda=1}^{p} (C^{-1}GC^{-1})_{\mu\nu}(C^{-1}GC^{-1})_{\rho\lambda}\langle\!\langle(\mathbf{h}_\nu \cdot \mathbf{h}_\rho)(\mathbf{h}_\lambda \cdot \mathbf{h}_\mu)\rangle\!\rangle_h \tag{D.21}$$

$$= n_2(n_2+1)\operatorname{tr}(C^{-1}GC^{-1}D^{-1}C^{-1}GC^{-1}D^{-1}) + n_2\operatorname{tr}(C^{-1}GC^{-1}D^{-1})^2$$
$$+ 2(n_2+1)\operatorname{tr}(C^{-1}GC^{-1}D^{-1}C^{-1}GC^{-1}D^{-1}JD^{-1})$$
$$+ 2\operatorname{tr}(C^{-1}GC^{-1}D^{-1})\operatorname{tr}(C^{-1}GC^{-1}D^{-1}JD^{-1})$$
$$+ \operatorname{tr}(C^{-1}GC^{-1}D^{-1}JD^{-1}C^{-1}GC^{-1}D^{-1}JD^{-1}) \tag{D.22}$$

by analogy with the corresponding quartic expectation for $\mathbf{q}_\mu$.

We must now expand our results in $n_1^{-1}$. The inverses of the matrices $C$ and $D$ have Neumann series

$$C^{-1} = G^{-1} + \frac{1}{n_1}G^{-1}BG^{-1} + \mathcal{O}(n_1^{-2}) \tag{D.23}$$

and

$$D^{-1} = \left(C^{-1} + \frac{1}{n_2}A\right)^{-1} \tag{D.24}$$

$$= \left(G^{-1} + \frac{1}{n_1}G^{-1}BG^{-1} + \frac{1}{n_2}A + \mathcal{O}(n_1^{-2})\right)^{-1} \tag{D.25}$$

$$= F^{-1}G - \frac{1}{n_1}F^{-1}BF^{-\top} + \mathcal{O}(n_1^{-2}) \tag{D.26}$$

where we have defined

$$F \equiv I_p + \frac{1}{n_2}GA \tag{D.27}$$

and we write $F^{-\top} = (F^{-1})^\top = (F^\top)^{-1}$. Then, using the series expansion of the log-determinant, we find that the logarithm of the leading term expands as

$$\frac{1}{2}\operatorname{tr}(D^{-1}J) - \frac{1}{2}n_2\log\det(CD) = \frac{1}{2}\operatorname{tr}(F^{-1}GJ) - \frac{1}{2}n_2\log\det(F)$$

$$- \frac{1}{2}\frac{1}{n_1}\operatorname{tr}(F^{-1}BF^{-\top}J) + \frac{1}{2}\frac{1}{n_1}\operatorname{tr}(F^{-1}BA)$$
$$+ \mathcal{O}(n_1^{-2}), \tag{D.28}$$

while the quartic correction simplifies to

$$\frac{1}{4}\frac{g}{n_1}\sum_{\mu,\nu,\rho,\lambda=1}^{p}G_{\mu\nu}G_{\rho\lambda}\langle\!\langle(\mathbf{q}_\nu\cdot\mathbf{q}_\rho)(\mathbf{q}_\lambda\cdot\mathbf{q}_\mu)\rangle\!\rangle$$

$$=\frac{1}{4}\frac{g}{n_1}n_2(n_2+p+1)p$$

$$+\frac{1}{4}\frac{n_2 g}{n_1}\left((n_2+1)\operatorname{tr}(F^{-2})+\operatorname{tr}(F^{-1})^2-2(n_2+p+1)\operatorname{tr}(F^{-1})\right)$$

$$+\frac{1}{2}\frac{g}{n_1}\left((n_2+1)\operatorname{tr}(F^{-3}GJ)+\operatorname{tr}(F^{-1})\operatorname{tr}(F^{-2}GJ)-(n_2+p+1)\operatorname{tr}(F^{-2}GJ)\right)$$

$$+\frac{1}{4}\frac{g}{n_1}\operatorname{tr}(F^{-2}GJF^{-2}GJ)$$

$$+\mathcal{O}(n_1^{-2}). \tag{D.29}$$

Combining these results, we find that the result of integrating out the layer to $\mathcal{O}(n_1^{-1})$ is

$$\log I = \frac{1}{2}\operatorname{tr}(F^{-1}GJ)-\frac{1}{2}n_2\log\det(F)$$

$$-\frac{1}{2}\frac{1}{n_1}\operatorname{tr}(F^{-1}BF^{-\top}J)+\frac{1}{2}\frac{1}{n_1}\operatorname{tr}(F^{-1}BA)$$

$$+\frac{1}{4}\frac{g}{n_1}n_2(n_2+p+1)p$$

$$+\frac{1}{4}\frac{n_2 g}{n_1}\left((n_2+1)\operatorname{tr}(F^{-2})+\operatorname{tr}(F^{-1})^2-2(n_2+p+1)\operatorname{tr}(F^{-1})\right)$$

$$+\frac{1}{2}\frac{g}{n_1}\left((n_2+1)\operatorname{tr}(F^{-3}GJ)+\operatorname{tr}(F^{-1})\operatorname{tr}(F^{-2}GJ)-(n_2+p+1)\operatorname{tr}(F^{-2}GJ)\right)$$

$$+\frac{1}{4}\frac{g}{n_1}\operatorname{tr}(F^{-2}GJF^{-2}GJ)$$

$$+\mathcal{O}(n_1^{-2}). \tag{D.30}$$

As this result is a continuous function of $G$, as the set of full-rank positive definite matrices is dense in the space of positive semidefinite matrices, this result holds for all positive-semidefinite $G$.

We now further expand this result in $n_2^{-1}$. This yields

$$F^{-1}=I_p-\frac{1}{n_2}GA+\frac{1}{n_2^2}GAGA+\mathcal{O}(n_2^{-3}) \tag{D.31}$$

and

$$\log\det(F)=\frac{1}{n_2}\operatorname{tr}(GA)-\frac{1}{2}\frac{1}{n_2^2}\operatorname{tr}(GAGA)+\mathcal{O}(n_2^{-3}), \tag{D.32}$$

hence we find that the logarithm of the leading term yields

$$\frac{1}{2}\operatorname{tr}(D^{-1}J)-\frac{1}{2}n_2\log\det(CD)=\frac{1}{2}\operatorname{tr}(GJ)-\frac{1}{2}\operatorname{tr}(GA)+\frac{1}{4}\frac{1}{n_2}\operatorname{tr}(GAGA)$$

$$-\frac{1}{2}\frac{1}{n_2}\operatorname{tr}(GAGJ)+\frac{1}{2}\frac{1}{n_1}\operatorname{tr}(B(A-J))$$

$$+\mathcal{O}(n_1^{-2},n_2^{-2},n_1^{-1}n_2^{-1}). \tag{D.33}$$

After some straightforward but tedious algebra, the quartic term reduces to

$$\frac{1}{4}\frac{g}{n_1}\sum_{\mu,\nu,\rho,\lambda=1}^{p}G_{\mu\nu}G_{\rho\lambda}\langle\!\langle(\mathbf{q}_\nu\cdot\mathbf{q}_\rho)(\mathbf{q}_\lambda\cdot\mathbf{q}_\mu)\rangle\!\rangle=\frac{1}{4}\frac{g}{n_1}\operatorname{tr}(G(A-J)G(A-J))$$

$$+\mathcal{O}(n_1^{-2},n_2^{-2},n_1^{-1}n_2^{-1}). \tag{D.34}$$

Combining these results, we find that the result of integrating out the layer is

$$\log I = \frac{1}{2}\operatorname{tr}(GJ) - \frac{1}{2}\operatorname{tr}(GA) + \frac{1}{4}\frac{1}{n_2}\left(1 + \frac{n_2}{n_1}g\right)\operatorname{tr}(GAGA)$$

$$- \frac{1}{2}\left(1 + \frac{n_2}{n_1}g\right)\operatorname{tr}(GAGJ) + \frac{1}{2}\frac{1}{n_1}\operatorname{tr}(B(A - J)) + \frac{1}{4}\frac{1}{n_1}g\operatorname{tr}(GJGJ)$$

$$+ \mathcal{O}(n_1^{-2}, n_2^{-2}, n_1^{-1}n_2^{-1}). \tag{D.35}$$

Again, this result is continuous in $G$, hence it holds even if $G$ is rank-deficient.

### D.3 Perturbative computation of the partition function of a deep linear network

We now apply the results of Appendix D.2 to compute the partition function for a deep linear network to the desired order. Our starting point is the effective action before any of the layers have been integrated out, including a source term:

$$S = -\frac{1}{2}\beta\sum_{\mu=1}^{p}\|\mathbf{h}_\mu^{(d)} - \mathbf{y}_\mu\|^2 + \sum_{\ell=1}^{d}\sum_{\mu=1}^{p} i\mathbf{q}_\mu^{(\ell)}\cdot\mathbf{h}_\mu^{(\ell)} - \frac{1}{2}\sum_{\mu,\nu=1}^{p}(\sigma_1^2 G_{xx})_{\mu\nu}(\mathbf{q}_\mu^{(1)}\cdot\mathbf{q}_\nu^{(1)})$$

$$- \frac{1}{2}\sum_{\ell=1}^{d-1}\frac{1}{n_\ell}\sum_{\mu,\nu=1}^{p}(J_{\mu\nu}^{(\ell)} + \sigma_{\ell+1}^2\mathbf{q}_\mu^{(\ell+1)}\cdot\mathbf{q}_\nu^{(\ell+1)})(\mathbf{h}_\mu^{(\ell)}\cdot\mathbf{h}_\nu^{(\ell)}). \tag{D.36}$$

Applying the results of Appendix D.2 with

$$G = \sigma_1^2 G_{xx},$$
$$\mathbf{j}_\mu = 0,$$
$$A = J^{(1)} + \sigma_2^2 Q^{(2)}, \tag{D.37}$$
$$B = 0, \quad \text{and}$$
$$g = 0,$$

we find that the effective action after integrating out the first layer is

$$S^{(1)} = -\frac{1}{2}\beta\sum_{\mu=1}^{p}\|\mathbf{h}_\mu^{(d)} - \mathbf{y}_\mu\|^2 + \sum_{\ell=2}^{d}\sum_{\mu=1}^{p} i\mathbf{q}_\mu^{(\ell)}\cdot\mathbf{h}_\mu^{(\ell)} - \frac{1}{2}\sum_{\mu,\nu=1}^{p}(m_2^2 G_{xx})_{\mu\nu}(\mathbf{q}_\mu^{(2)}\cdot\mathbf{q}_\nu^{(2)})$$

$$- \frac{1}{2}\sum_{\ell=2}^{d-1}\frac{1}{n_\ell}\sum_{\mu,\nu=1}^{p}(J_{\mu\nu}^{(\ell)} + \sigma_{\ell+1}^2\mathbf{q}_\mu^{(\ell+1)}\cdot\mathbf{q}_\nu^{(\ell+1)})(\mathbf{h}_\mu^{(\ell)}\cdot\mathbf{h}_\nu^{(\ell)})$$

$$+ \frac{1}{4}\frac{g_1}{n_1}m_2^4\operatorname{tr}(G_{xx}Q^{(2)}G_{xx}Q^{(2)}) + \frac{1}{2}\frac{g_1}{n_1}m_2^2\operatorname{tr}(G_{xx}\tilde{J}_1 G_{xx}Q^{(2)})$$

$$- \frac{1}{2}\operatorname{tr}(m_1^2 G_{xx}J^{(1)}) + \frac{1}{4}\frac{g_1}{n_1}m_1^4\operatorname{tr}(G_{xx}J^{(1)}G_{xx}J^{(1)})$$

$$+ \mathcal{O}(n^{-2}), \tag{D.38}$$

where we have defined

$$m_1 \equiv \sigma_1,$$
$$m_2 \equiv \sigma_2 m_1,$$
$$g_1 \equiv 1, \quad \text{and} \tag{D.39}$$
$$\tilde{J}_1 \equiv m_1^2 J^{(1)}.$$

Assuming that the network has more than one hidden layer, if we now again apply the results of Appendix D.2 with

$$G = m_2^2 G_{xx},$$
$$\mathbf{j}_\mu = 0,$$
$$A = J^{(2)} + \sigma_3^2 Q^{(3)}, \tag{D.40}$$
$$B = g_1 m_2^2 G_{xx}\tilde{J}_1 G_{xx}, \quad \text{and}$$
$$g = g_1,$$

S16

we find that the effective action after integrating out the first two layers is

$$S^{(2)} = -\frac{1}{2}\beta \sum_{\mu=1}^{p} \|\mathbf{h}_\mu^{(d)} - \mathbf{y}_\mu\|^2 + \sum_{\ell=3}^{d}\sum_{\mu=1}^{p} i\mathbf{q}_\mu^{(\ell)} \cdot \mathbf{h}_\mu^{(\ell)} - \frac{1}{2}\sum_{\mu,\nu=1}^{p} (m_3^2 G_{xx})_{\mu\nu}(\mathbf{q}_\mu^{(3)} \cdot \mathbf{q}_\nu^{(3)})$$

$$-\frac{1}{2}\sum_{\ell=3}^{d-1}\frac{1}{n_\ell}\sum_{\mu,\nu=1}^{p}(J_{\mu\nu}^{(\ell)} + \sigma_{\ell+1}^2 \mathbf{q}_\mu^{(\ell+1)} \cdot \mathbf{q}_\nu^{(\ell+1)})(\mathbf{h}_\mu^{(\ell)} \cdot \mathbf{h}_\nu^{(\ell)})$$

$$+\frac{1}{4}\frac{g_2}{n_2}m_3^4 \operatorname{tr}(G_{xx}Q^{(3)}G_{xx}Q^{(3)})$$

$$+\frac{1}{2}\frac{g_2}{n_2}m_3^2 \operatorname{tr}(G_{xx}\tilde{J}_2 G_{xx}Q^{(3)})$$

$$-\frac{1}{2}\operatorname{tr}(m_1^2 G_{xx}J^{(1)}) - \frac{1}{2}\operatorname{tr}(m_2^2 G_{xx}J^{(2)})$$

$$+\frac{1}{4}\frac{g_1}{n_1}m_1^4 \operatorname{tr}(G_{xx}J^{(1)}G_{xx}J^{(1)}) + \frac{1}{4}\frac{g_2}{n_2}m_2^4 \operatorname{tr}(G_{xx}J^{(2)}G_{xx}J^{(2)})$$

$$+\frac{1}{2}\frac{g_1}{n_1}m_2^2 \operatorname{tr}(G_{xx}\tilde{J}_1 G_{xx}J^{(2)})$$

$$+\mathcal{O}(n^{-2}), \tag{D.41}$$

where we have defined

$$m_3 \equiv \sigma_3 m_2,$$

$$g_2 \equiv 1 + \frac{n_2}{n_1}g_1, \quad \text{and} \tag{D.42}$$

$$\tilde{J}_2 \equiv m_2^2 J^{(2)} + \frac{n_2}{n_1}\frac{g_1}{g_2}\tilde{J}_1.$$

Then, by induction, we can see that we can iterate this procedure to integrate out all of the hidden layers, yielding

$$S^{(d-1)} = -\frac{1}{2}\beta \sum_{\mu=1}^{p} \|\mathbf{h}_\mu^{(d)} - \mathbf{y}_\mu\|^2 + \sum_{\mu=1}^{p} i\mathbf{q}_\mu^{(d)} \cdot \mathbf{h}_\mu^{(d)} - \frac{1}{2}\sum_{\mu,\nu=1}^{p}(m_d^2 G_{xx})_{\mu\nu}(\mathbf{q}_\mu^{(d)} \cdot \mathbf{q}_\nu^{(d)})$$

$$+\frac{1}{4}\frac{g_{d-1}}{n_{d-1}}m_d^4 \operatorname{tr}(G_{xx}Q^{(d)}G_{xx}Q^{(d)})$$

$$+\frac{1}{2}\frac{g_{d-1}}{n_{d-1}}m_d^2 \operatorname{tr}(G_{xx}\tilde{J}_{d-1}G_{xx}Q^{(d)})$$

$$-\frac{1}{2}\sum_{\ell=1}^{d-1}\operatorname{tr}(m_\ell^2 G_{xx}J^{(\ell)})$$

$$+\frac{1}{4}\sum_{\ell=1}^{d-1}\frac{g_\ell}{n_\ell}m_\ell^4 \operatorname{tr}(G_{xx}J^{(\ell)}G_{xx}J^{(\ell)})$$

$$+\frac{1}{2}\sum_{\ell=1}^{d-2}\frac{g_\ell}{n_\ell}m_{\ell+1}^2 \operatorname{tr}(G_{xx}\tilde{J}_\ell G_{xx}J^{(\ell+1)})$$

$$+\mathcal{O}(n^{-2}), \tag{D.43}$$

where $m_d$, $g_{d-1}$, and $\tilde{J}_{d-1}$ are defined by the closed recurrences

$$m_\ell \equiv \sigma_\ell m_{\ell-1}, \tag{D.44}$$

$$g_\ell \equiv 1 + \frac{n_\ell}{n_{\ell-1}}g_{\ell-1}, \quad \text{and} \tag{D.45}$$

$$\tilde{J}_\ell \equiv m_\ell^2 J^{(\ell)} + \frac{n_\ell}{n_{\ell-1}}\frac{g_{\ell-1}}{g_\ell}\tilde{J}_{\ell-1}. \tag{D.46}$$

Applying the results of Appendix D.2 one final time with

$$
\begin{aligned}
G &= m_d^2 G_{xx}, \\
\mathbf{j}_\mu &= \beta \mathbf{y}, \\
A &= \beta n_d I_p, \\
B &= g_{d-1} m_d^2 G_{xx} \tilde{J}_{d-1} G_{xx}, \quad \text{and} \\
g &= g_{d-1},
\end{aligned}
$$
(D.47)

we conclude that

$$
\begin{aligned}
\log Z ={}& -\frac{1}{2}\beta n_d \operatorname{tr}(\tilde{\Gamma}^{-1} G_{yy}) - \frac{1}{2} n_d \log \det(\tilde{\Gamma}) \\
&+ \frac{1}{4}\frac{n_d g_{d-1}}{n_{d-1}}\left( (n_d + p + 1)p + (n_d + 1)\operatorname{tr}(\tilde{\Gamma}^{-2}) + \operatorname{tr}(\tilde{\Gamma}^{-1})^2 - 2(n_d + p + 1)\operatorname{tr}(\tilde{\Gamma}^{-1}) \right) \\
&+ \frac{1}{2}\frac{g_{d-1}}{n_{d-1}}\beta^2 n_d m_d^2 \bigg( (n_d + 1)\operatorname{tr}(\tilde{\Gamma}^{-3} G_{xx} G_{yy}) + \operatorname{tr}(\tilde{\Gamma}^{-1})\operatorname{tr}(\tilde{\Gamma}^{-2} G_{xx} G_{yy}) \\
&\qquad\qquad\qquad - (n_d + p + 1)\operatorname{tr}(\tilde{\Gamma}^{-2} G_{xx} G_{yy}) \bigg) \\
&+ \frac{1}{4}\frac{g_{d-1}}{n_{d-1}}\beta^4 n_d^2 m_d^4 \operatorname{tr}(\tilde{\Gamma}^{-2} G_{xx} G_{yy} \tilde{\Gamma}^{-2} G_{xx} G_{yy}) \\
&- \frac{1}{2}\frac{g_{d-1}}{n_{d-1}}n_d m_d^2 \operatorname{tr}\left[ \left( \beta^2 G_{xx}\tilde{\Gamma}^{-1} G_{yy}\tilde{\Gamma}^{-1} G_{xx} - \beta G_{xx}\tilde{\Gamma}^{-1} G_{xx} \right)\tilde{J}_{d-1} \right] \\
&- \frac{1}{2}\sum_{\ell=1}^{d-1}\operatorname{tr}(m_\ell^2 G_{xx} J^{(\ell)}) \\
&+ \frac{1}{4}\sum_{\ell=1}^{d-1}\frac{g_\ell}{n_\ell}m_\ell^4 \operatorname{tr}(G_{xx} J^{(\ell)} G_{xx} J^{(\ell)}) \\
&+ \frac{1}{2}\sum_{\ell=1}^{d-2}\frac{g_\ell}{n_\ell}m_{\ell+1}^2 \operatorname{tr}(G_{xx}\tilde{J}_\ell G_{xx} J^{(\ell+1)}) \\
&+ \mathcal{O}(n^{-2}),
\end{aligned}
$$
(D.48)

where we have defined the matrix

$$
\tilde{\Gamma} \equiv I_p + \beta m_d^2 G_{xx}
$$
(D.49)

and absorbed the normalizing constant using the fact that $I_p - \beta m_d^2 \tilde{\Gamma}^{-1} G_{xx} = \tilde{\Gamma}^{-1}$. As was the case for the individual layer integrals, a continuity argument implies that this expression can be applied even if $G_{xx}$ is rank-deficient.

## D.4  Computing the average hidden layer kernels of a deep linear network

With the relevant partition function in hand, we can finally compute the average hidden layer kernels. In particular, we can immediately read off that

$$
\begin{aligned}
\langle K^{(\ell)} \rangle ={}& m_\ell^2 G_{xx} \\
&+ \frac{g_{d-1}}{n_{d-1}}n_d m_d^2 \operatorname{tr}\left[ \left( \beta^2 G_{xx}\tilde{\Gamma}^{-1} G_{yy}\tilde{\Gamma}^{-1} G_{xx} - \beta G_{xx}\tilde{\Gamma}^{-1} G_{xx} \right)\frac{\delta \tilde{J}_{d-1}}{\delta J^{(\ell)}}\bigg|_{J^{(\ell)}=0} \right] \\
&+ \mathcal{O}(n^{-2}),
\end{aligned}
$$
(D.50)

hence our only task is to determine how the effective source $\tilde{J}_{d-1}$ depends on the source for a given layer. Fortunately, the recurrence relation for the effective source is extremely easy to solve, yielding

$$
\tilde{J}_{d-1} = \sum_{\ell=1}^{d-1} m_\ell^2 \frac{n_{d-1}}{n_\ell}\frac{g_\ell}{g_{d-1}} J^{(\ell)}.
$$
(D.51)

Thus, defining the matrix

$$\Gamma \equiv \frac{1}{\beta m_d^2}\tilde{\Gamma} = G_{xx} + \frac{1}{\beta m_d^2}I_p,$$ (D.52)

we find that

$$\langle K^{(\ell)}\rangle = m_\ell^2 G_{xx} + \frac{g_\ell}{n_\ell}n_d m_\ell^2\left(m_d^{-2}G_{xx}\Gamma^{-1}G_{yy}\Gamma^{-1}G_{xx} - G_{xx}\Gamma^{-1}G_{xx}\right) + \mathcal{O}(n^{-2}).$$ (D.53)

To obtain the expression listed in the main text, we note that

$$\frac{g_\ell}{n_\ell} = \frac{1}{n_\ell} + \frac{g_{\ell-1}}{n_{\ell-1}},$$ (D.54)

hence we have

$$\frac{g_\ell}{n_\ell} = \sum_{\ell'=1}^{\ell}\frac{1}{n_{\ell'}},$$ (D.55)

mirroring the width dependence found by Yaida [8] in his study of the prior of deep linear networks.

# E  Average kernels in a deep feedforward linear network with skip connections

In this appendix, we show that Conjecture 1 holds perturbatively for a linear feedforward network with arbitrary skip connections, following the method of Appendix D. Concretely, we consider a network defined as

$$\mathbf{h}^{(0)} = \mathbf{x}$$ (E.1)

$$\mathbf{h}^{(\ell)} = \sum_{\ell'=0}^{\ell-1}\frac{\sigma_{\ell,\ell'}}{\sqrt{n_{\ell'}}}W^{(\ell,\ell')}\mathbf{h}^{(\ell')} \qquad \ell = 1,\dots,d$$ (E.2)

$$\mathbf{f} = \mathbf{h}^{(d)},$$ (E.3)

where $\sigma_{\ell,\ell'}$ is positive if layer $\ell$ receives input from an earlier layer $\ell' < \ell$, and zero otherwise.

## E.1  Perturbative computation of the partition function

Upon integrating out the weights, we obtain an effective action for the preactivations and the corresponding Lagrange multipliers of

$$\begin{aligned}
S = {}&-\beta\sum_{\mu=1}^{p}\varepsilon(\mathbf{h}_\mu^{(d)}, \mathbf{y}_\mu) + \sum_{\mu=1}^{p}\sum_{\ell=1}^{d}i\mathbf{q}_\mu^{(\ell)}\cdot\mathbf{h}_\mu^{(\ell)}\\
&-\frac{1}{2}\sum_{\ell=1}^{d-1}\frac{1}{n_\ell}\sum_{\mu,\nu=1}^{p}\left[J^{(\ell)} + \sum_{\ell'=\ell+1}^{d}\sigma_{\ell',\ell}^2(\mathbf{q}_\mu^{(\ell')}\cdot\mathbf{q}_\nu^{(\ell')})\right](\mathbf{h}_\mu^{(\ell)}\cdot\mathbf{h}_\nu^{(\ell)})\\
&-\frac{1}{2}\sum_{\ell=1}^{d}\sigma_{\ell,0}^2\sum_{\mu,\nu=1}^{p}(G_{xx})_{\mu\nu}(\mathbf{q}_\mu^{(\ell)}\cdot\mathbf{q}_\nu^{(\ell)}).
\end{aligned}$$ (E.4)

Applying the result of Appendix D.2 with

$$\begin{aligned}
G &= \sigma_{1,0}^2 G_{xx},\\
\mathbf{j}_\mu &= \mathbf{0},\\
A &= J^{(1)} + \sum_{\ell'=2}^{d}\sigma_{\ell',1}^2 Q^{(\ell')},\\
B &= 0, \quad\text{and}\\
g &= 0,
\end{aligned}$$ (E.5)

we find that the effective action after integrating out the first layer is

$$
\begin{aligned}
S^{(1)} = &-\beta \sum_{\mu=1}^{p} \varepsilon(\mathbf{h}_\mu^{(d)}, \mathbf{y}_\mu) + \sum_{\mu=1}^{p}\sum_{\ell=2}^{d} i\mathbf{q}_\mu^{(\ell)} \cdot \mathbf{h}_\mu^{(\ell)} \\
&- \frac{1}{2}\sum_{\ell=2}^{d-1} \frac{1}{n_\ell} \sum_{\mu,\nu=1}^{p} \left[ J^{(\ell)} + \sum_{\ell'=\ell+1}^{d} \sigma_{\ell',\ell}^2 (\mathbf{q}_\mu^{(\ell')} \cdot \mathbf{q}_\nu^{(\ell')}) \right] (\mathbf{h}_\mu^{(\ell)} \cdot \mathbf{h}_\nu^{(\ell)}) \\
&- \frac{1}{2}\sum_{\ell=2}^{d} m_{\ell,1}^2 \operatorname{tr}(G_{xx}Q^{(\ell)}) + \frac{1}{4}\frac{1}{n_1} \sum_{\ell,\ell'=2}^{d} g_{\ell,\ell',1} \operatorname{tr}(G_{xx}Q^{(\ell)}G_{xx}Q^{(\ell')}) \\
&+ \frac{1}{2}\frac{1}{n_1} \sum_{\ell=2}^{d} \operatorname{tr}(G_{xx}\tilde{J}_{\ell,1}G_{xx}Q^{(\ell)}) \\
&- \frac{1}{2}m_{1,0}^2 \operatorname{tr}(G_{xx}J^{(1)}) \\
&+ \frac{1}{4}\frac{1}{n_1}\sigma_{1,0}^4 \operatorname{tr}(G_{xx}J^{(1)}G_{xx}J^{(1)}) \\
&+ \mathcal{O}(n^{-2}),
\end{aligned}
\tag{E.6}
$$

where we have defined

$$
m_{\ell,0}^2 \equiv \sigma_{\ell,0}^2,
\tag{E.7}
$$

$$
m_{\ell,1}^2 \equiv m_{\ell,0}^2 + \sigma_{\ell,1}^2 m_{1,0}^2
\tag{E.8}
$$

$$
g_{\ell,\ell',1} \equiv \sigma_{\ell,1}^2 \sigma_{\ell',1}^2 \sigma_{1,0}^4, \quad \text{and}
\tag{E.9}
$$

$$
\tilde{J}_{\ell,1} \equiv \sigma_{\ell,1}^2 \sigma_{1,0}^4 J^{(1)},
\tag{E.10}
$$

where $\ell, \ell' > 1$ for all cases but $m_{1,0}^2$. Assuming the network has more than one hidden layer, if we now again apply the results of Appendix D.2 with

$$
\begin{aligned}
&G = m_{2,1}^2 G_{xx}, \\
&\mathbf{j}_\mu = \mathbf{0}, \\
&A = J^{(2)} + \sum_{\ell'=3}^{d} \sigma_{\ell',2}^2 Q^{(\ell')}, \\
&B = G_{xx}\tilde{J}_{2,1}G_{xx} + \sum_{\ell'=3}^{d} g_{\ell',2,1}G_{xx}Q^{(\ell')}G_{xx}, \quad \text{and} \\
&g = g_{2,2,1}/m_{2,1}^4,
\end{aligned}
\tag{E.11}
$$

we find that the effective action after integrating out the first two layers of the network is

$$
\begin{aligned}
S^{(2)} = {} & -\beta \sum_{\mu=1}^{p} \varepsilon(\mathbf{h}_\mu^{(d)}, \mathbf{y}_\mu) + \sum_{\mu=1}^{p} \sum_{\ell=1}^{d} i\mathbf{q}_\mu^{(\ell)} \cdot \mathbf{h}_\mu^{(\ell)} \\
& - \frac{1}{2} \sum_{\ell=3}^{d-1} \frac{1}{n_\ell} \sum_{\mu,\nu=1}^{p} \left[ J_{\mu\nu}^{(\ell)} + \sum_{\ell'=\ell+1}^{d} \sigma_{\ell',\ell}^2 (\mathbf{q}_\mu^{(\ell')} \cdot \mathbf{q}_\nu^{(\ell')}) \right] (\mathbf{h}_\mu^{(\ell)} \cdot \mathbf{h}_\nu^{(\ell)}) \\
& - \frac{1}{2} \sum_{\ell=3}^{d} m_{\ell,2}^2 \operatorname{tr}(G_{xx} Q^{(\ell)}) + \frac{1}{4} \frac{1}{n_2} \sum_{\ell,\ell'=3}^{d} g_{\ell,\ell',2} \operatorname{tr}(G_{xx} Q^{(\ell)} G_{xx} Q^{(\ell')}) \\
& + \frac{1}{2} \frac{1}{n_2} \sum_{\ell=3}^{d} \operatorname{tr}(G_{xx} \tilde{J}_{\ell,2} G_{xx} Q^{(\ell)}) \\
& - \frac{1}{2} \sum_{\ell=1}^{2} m_{\ell,\ell-1}^2 \operatorname{tr}(G_{xx} J^{(\ell)}) \\
& + \frac{1}{4} \frac{1}{n_1} m_{1,0}^4 \operatorname{tr}(G_{xx} J^{(1)} G_{xx} J^{(1)}) + \frac{1}{4} \frac{1}{n_2} \left( m_{2,1}^4 + \frac{n_2}{n_1} g_{2,2,1} \right) \operatorname{tr}(G_{xx} J^{(2)} G_{xx} J^{(2)}) \\
& + \frac{1}{2} \frac{1}{n_1} \operatorname{tr}(G_{xx} \tilde{J}_{2,1} G_{xx} J^{(2)}) \\
& + \mathcal{O}(n^{-2}),
\end{aligned}
\tag{E.12}
$$

where we now define

$$
m_{\ell,2}^2 \equiv m_{\ell,1}^2 + m_{2,1}^2 \sigma_{\ell,2}^2,
\tag{E.13}
$$

$$
g_{\ell,\ell',2} \equiv m_{2,1}^4 + \frac{n_2}{n_1} \left( g_{\ell,\ell',1} + g_{2,2,1} \sigma_{\ell,2}^2 \sigma_{\ell',2}^2 + g_{\ell,2,1} \sigma_{\ell',2}^2 + \sigma_{\ell,2}^2 g_{2,\ell',1} \right), \quad \text{and}
\tag{E.14}
$$

$$
\tilde{J}_{\ell,2} \equiv \frac{n_2}{n_1} \tilde{J}_{\ell,1} + \left( m_{2,1}^4 + \frac{n_2}{n_1} g_{2,2,1} \right) \sigma_{\ell,2}^2 J^{(2)} + \frac{n_2}{n_1} \sigma_{\ell,2}^2 \tilde{J}_{2,1} + \frac{n_2}{n_1} g_{\ell,2,1} J^{(2)}
\tag{E.15}
$$

for $\ell, \ell' > 2$. We can now see that we can repeat this procedure to integrate out all of the hidden layers of the network, yielding an effective action of

$$
\begin{aligned}
S^{(d-1)} = {} & -\beta \sum_{\mu=1}^{p} \varepsilon(\mathbf{h}_\mu^{(d)}, \mathbf{y}_\mu) + \sum_{\mu=1}^{p} i\mathbf{q}_\mu^{(d)} \cdot \mathbf{h}_\mu^{(d)} \\
& - \frac{1}{2} m_{d,d-1}^2 \operatorname{tr}(G_{xx} Q^{(d)}) + \frac{1}{4} \frac{1}{n_{d-1}} g_{d,d,d-1} \operatorname{tr}(G_{xx} Q^{(d)} G_{xx} Q^{(d)}) \\
& + \frac{1}{2} \frac{1}{n_{d-1}} \operatorname{tr}(G_{xx} \tilde{J}_{d,d-1} G_{xx} Q^{(d)}) \\
& - \frac{1}{2} \sum_{\tau=1}^{d-1} m_{\tau,\tau-1}^2 \operatorname{tr}(G_{xx} J^{(\tau)}) \\
& + \frac{1}{4} \sum_{\tau=2}^{d-1} \left( \frac{1}{n_\tau} m_{\tau,\tau-1}^4 + \frac{1}{n_{\tau-1}} g_{\tau,\tau,\tau-1} \right) \operatorname{tr}(G_{xx} J^{(\tau)} G_{xx} J^{(\tau)}) \\
& + \frac{1}{2} \sum_{\tau=2}^{d-1} \frac{1}{n_{\tau-1}} \operatorname{tr}(G_{xx} \tilde{J}_{\tau,\tau-1} G_{xx} J^{(\tau)}) \\
& + \mathcal{O}(n^{-2}),
\end{aligned}
\tag{E.16}
$$

where the coupling constants and effective source obey the recurrences

$$m_{\ell,\tau}^2 \equiv m_{\ell,\tau-1}^2 + m_{\tau,\tau-1}^2 \sigma_{\ell,\tau}^2, \tag{E.17}$$

$$
\begin{aligned}
g_{\ell,\ell',\tau} \equiv{}& m_{\tau,\tau-1}^4 \sigma_{\ell,\tau}^2 \sigma_{\ell',\tau}^2 \\
&+ \frac{n_\tau}{n_{\tau-1}}\left( g_{\ell,\ell',\tau-1} + g_{\tau,\tau,\tau-1}\sigma_{\ell,\tau}^2\sigma_{\ell',\tau}^2 + g_{\ell,\tau,\tau-1}\sigma_{\ell',\tau}^2 + \sigma_{\ell,\tau}^2 g_{\tau,\ell',\tau-1} \right), \quad \text{and} \tag{E.18}
\end{aligned}
$$

$$
\begin{aligned}
\tilde{J}_{\ell,\tau} \equiv{}& \frac{n_\tau}{n_{\tau-1}}\tilde{J}_{\ell,\tau-1} + \frac{n_\tau}{n_{\tau-1}}\sigma_{\ell,\tau}^2 \tilde{J}_{\tau,\tau-1} \\
&+ \left( m_{\tau,\tau-1}^4 \sigma_{\ell,\tau}^2 + \frac{n_\tau}{n_{\tau-1}}g_{\tau,\tau,\tau-1}\sigma_{\ell,\tau}^2 + \frac{n_\tau}{n_{\tau-1}}g_{\ell,\tau,\tau-1} \right) J^{(\tau)} \tag{E.19}
\end{aligned}
$$

for $\ell, \ell' > \tau$. Applying the results of Appendix D.2 once more with

$$
\begin{aligned}
G &= m_{d,d-1}^2 G_{xx}, \\
\mathbf{j}_\mu &= \beta \mathbf{y}_\mu, \\
A &= \beta n_d I_p, \\
B &= G_{xx}\tilde{J}_{d,d-1}G_{xx}, \quad \text{and} \\
g &= g_{d,d,d-1}/m_{d,d-1}^4, \tag{E.20}
\end{aligned}
$$

we find the source-dependent terms in the logarithm of the partition function are

$$
\begin{aligned}
\log Z \supset{}& -\frac{1}{2}\frac{1}{n_1}\beta^2 n_d \operatorname{tr}(\Gamma^{-1}G_{xx}\tilde{J}_{d,d-1}G_{xx}\Gamma^{-1}G_{yy}) + \frac{1}{2}\frac{1}{n_{d-1}}\beta n_d \operatorname{tr}(\Gamma^{-1}G_{xx}\tilde{J}_{d,d-1}G_{xx}) \\
&- \frac{1}{2}\sum_{\tau=1}^{d-1} m_{\tau,\tau-1}^2 \operatorname{tr}(G_{xx}J^{(\tau)}) \\
&+ \frac{1}{4}\sum_{\tau=2}^{d-1}\left(\frac{1}{n_\tau}m_{\tau,\tau-1}^4 + \frac{1}{n_{\tau-1}}g_{\tau,\tau,\tau-1}\right)\operatorname{tr}(G_{xx}J^{(\tau)}G_{xx}J^{(\tau)}) \\
&+ \frac{1}{2}\sum_{\tau=2}^{d-1}\frac{1}{n_{\tau-1}}\operatorname{tr}(G_{xx}\tilde{J}_{\tau,\tau-1}G_{xx}J^{(\tau)}) \\
&+ \mathcal{O}(n^{-2}), \tag{E.21}
\end{aligned}
$$

where

$$\Gamma \equiv I_p + \beta m_{d,d-1}^2 G_{xx}. \tag{E.22}$$

## E.2 Computing the average hidden layer kernels

With the source-dependent terms of the relevant partition function in hand, we can compute the average hidden layer kernels for a feedforward linear network with arbitrary skip connections. We can immediately read off that

$$
\begin{aligned}
\langle K^{(\ell)}\rangle ={}& m_{\ell,\ell-1}^2 G_{xx} \\
&+ \frac{n_d}{n_{d-1}}\operatorname{tr}\left[\left(\beta^2 G_{xx}\Gamma^{-1}G_{yy}\Gamma^{-1}G_{xx} - \beta G_{xx}\Gamma^{-1}G_{xx}\right)\frac{\delta\tilde{J}_{d,d-1}}{\delta J^{(\ell)}}\bigg|_{J^{(\ell)}=0}\right] \\
&+ \mathcal{O}(n^{-2}), \tag{E.23}
\end{aligned}
$$

hence our only task is to compute the derivative of the effective source $\tilde{J}_{d,d-1}$ with respect to the source for the $\ell$-th hidden layer. Singling out the $\ell$-th layer, we can set all sources except $J^{(\ell)}$ to zero. Then, the 'earliest' effective source to be non-zero is

$$\tilde{J}_{\ell',\ell} = \left( m_{\ell,\ell-1}^4 \sigma_{\ell',\ell}^2 + \frac{n_\ell}{n_{\ell-1}}g_{\ell,\ell,\ell-1}\sigma_{\ell',\ell}^2 + \frac{n_\ell}{n_{\ell-1}}g_{\ell',\ell,\ell-1}\right) J^{(\ell)}, \tag{E.24}$$

for $\ell' > \ell$, and the recurrence relation for $\tau > \ell$ is

$$\tilde{J}_{\ell',\tau} = \frac{n_\tau}{n_{\tau-1}} \left( \tilde{J}_{\ell',\tau-1} + \sigma^2_{\ell',\tau} \tilde{J}_{\tau,\tau-1} \right). \tag{E.25}$$

From the form of these recurrences, we can see that

$$
\begin{aligned}
\langle K^{(\ell)} \rangle = {} & m^2_{\ell,\ell-1} G_{xx} \\
& + \frac{n_d}{n_{d-1}} \tilde{g}_\ell G_{xx} (\beta^2 \Gamma^{-1} G_{yy} \Gamma^{-1} - \beta \Gamma^{-1}) G_{xx} \\
& + \mathcal{O}(n^{-2}),
\end{aligned}
\tag{E.26}
$$

where $\tilde{g}_\ell$ is a layer-dependent scalar. Even without explicitly solving the recurrences to obtain $\tilde{g}_\ell$, this shows that Conjecture 1 holds perturbatively for linear networks with arbitrary skip connections. We leave detailed study of these recurrences—and therefore of the precise dependence of the corrections on width, depth, and skip connection structure—as an interesting objective for future work.

# F Comparison to the results of Aitchison [10] and Li and Sompolinsky [11]

In this appendix, we compare our results for the average kernels of deep linear networks to those of Aitchison [10] and Li and Sompolinsky [11].

## F.1 Comparison to the results of Aitchison [10]

We first show that our result (9) for the low-temperature limit of the average kernels of a deep linear network can be recovered from the results of Aitchison [10]. Working in what corresponds to the zero-temperature limit of our setup, Aitchison derives the following implicit recurrence

$$0 = -(n_{\ell+1} - n_\ell)(K^{(\ell)})^{-1} + n_{\ell+1}(K^{(\ell)})^{-1}(K^{(\ell+1)})(K^{(\ell)})^{-1} - n_\ell(K^{(\ell-1)})^{-1}, \tag{F.1}$$

for $\ell = 1, \ldots, d-1$, where the boundary conditions of the recurrence are $K^{(0)} = G_{xx}$ and $K^{(d)} = G_{yy}$. We will self-consistently solve this recurrence relation in the limit $n_1, \ldots, n_{d-1} \to \infty$, $n_0, n_d, p = \mathcal{O}(1)$. Concretely, we make the ansatz that the zero-temperature kernels are of the form

$$K^{(\ell)} = K_\infty^{(\ell)} + \frac{1}{n_\ell} K_1^{(\ell)} + \mathcal{O}(n_\ell^{-2}), \tag{F.2}$$

and solve the recurrence relations order-by-order using the resulting Neumann series

$$(K^{(\ell)})^{-1} = (K_\infty^{(\ell)})^{-1} - \frac{1}{n_\ell} (K_\infty^{(\ell)})^{-1} K_1^{(\ell)} (K_\infty^{(\ell)})^{-1} + \mathcal{O}(n_\ell^{-2}). \tag{F.3}$$

The leading-order recurrence is simply

$$0 = \left(1 - \frac{n_{\ell+1}}{n_\ell}\right) (K_\infty^{(\ell)})^{-1} + \frac{n_{\ell+1}}{n_\ell} (K_\infty^{(\ell)})^{-1}(K_\infty^{(\ell+1)})(K_\infty^{(\ell)})^{-1} - (K_\infty^{(\ell-1)})^{-1}, \tag{F.4}$$

with boundary conditions $K_\infty^{(0)} = G_{xx}$ and $K_\infty^{(d)} = G_{yy}$. For the last hidden layer, we have $n_{\ell+1}/n_\ell = n_d/n_{d-1} \to 0$, hence the recurrence reduces to

$$K_\infty^{(d-1)} = K_\infty^{(d-2)}. \tag{F.5}$$

If we iterate this procedure backwards through the network, it is easy to see that the $n_{\ell+1}/n_\ell$-dependent terms at each layer will cancel, leaving

$$K_\infty^{(d-1)} = K_\infty^{(d-2)} = \cdots = K_\infty^{(1)} = G_{xx}. \tag{F.6}$$

We now consider the leading finite-width correction. For the last hidden layer, we obtain

$$0 = n_d(G_{yy} - G_{xx}) - K_1^{(d-1)} + \frac{n_{d-1}}{n_{d-2}} K_1^{(d-2)} \tag{F.7}$$

after dropping all terms that are of $\mathcal{O}(n^{-2})$ and multiplying on the left and right by $G_{xx}$. For the first hidden layer, we have

$$0 = K_1^{(2)} - \left(1 + \frac{n_2}{n_1}\right) K_1^{(1)}. \tag{F.8}$$

Finally, for intermediate hidden layers (i.e., $\ell = 2, 3, \ldots, d - 2$), we have

$$0 = K_1^{(\ell+1)} - \left(1 + \frac{n_{\ell+1}}{n_\ell}\right) K_1^{(\ell)} + \frac{n_\ell}{n_{\ell-1}} K_1^{(\ell-1)}. \tag{F.9}$$

Based on the form of these recurrences, we make the ansatz that the solution is of the form

$$K_1^{(\ell)} = n_d a_\ell (G_{yy} - G_{xx}) \tag{F.10}$$

for some sequence $a_\ell$, where we assume that $G_{yy} \neq G_{xx}$. Then, the recurrence for the last hidden layer is satisfied provided that

$$a_{d-1} = 1 + \frac{n_{d-1}}{n_{d-2}} a_{d-2}, \tag{F.11}$$

those for the intermediate layers if

$$0 = a_{\ell+1} - \left(1 + \frac{n_{\ell+1}}{n_\ell}\right) a_\ell + \frac{n_\ell}{n_{\ell-1}} a_{\ell-1}, \tag{F.12}$$

and that for the first hidden layer if

$$a_2 = \left(1 + \frac{n_2}{n_1}\right) a_1. \tag{F.13}$$

Substituting the expression for $a_{d-1}$ into the condition resulting from the recurrence relation centered on $a_{d-2}$, we find that we must have

$$a_{d-2} = 1 + \frac{n_{d-2}}{n_{d-3}} a_{d-3}, \tag{F.14}$$

hence we can iterate this process backwards to the second hidden layer, yielding

$$a_\ell = 1 + \frac{n_\ell}{n_{\ell-1}} a_{\ell-1} \tag{F.15}$$

for $\ell = 2, 3, \ldots, d - 1$. Then, the condition relating $a_2$ and $a_1$ resulting from the recurrence relation for the first layer implies that we must have $a_1 = 1$. Thus, we recover our zero-temperature result from solving Aitchison's recurrence relations order-by-order.

## F.2 Comparison to the results of Li and Sompolinsky [11]

We now show that our result (9) for the low-temperature limit of the average kernels of a deep linear network can be recovered as a limiting case of the result of Li and Sompolinsky [11]. Their result for the zero-temperature kernel in the limit $n_0, n, p \to \infty$ with $n_1 = n_2 = \cdots = n_{d-1} = n$, $n_0/n \in (0, \infty)$, $\alpha \equiv p/n \in (0, \infty)$, and $\sigma_1 = \cdots = \sigma_d = \sigma$ is, in our notation,

$$\sigma^{-2(\ell+1)} \langle K^{(\ell)} \rangle \sim \left(1 - \frac{n_d}{n}\right)^\ell G_{xx} + \frac{1}{n} \sigma^{-2d} Y V M_\ell V^\top Y^\top, \tag{F.16}$$

where $Y \in \mathbb{R}^{p \times n_d}$ is the matrix of targets and $M_\ell \in \mathbb{R}^{n_d \times n_d}$ is a diagonal matrix with non-zero elements

$$[M_\ell]_{kk} = z_k^{-(d-1)} \frac{z_k^\ell - 1}{z_k - 1}. \tag{F.17}$$

Here, the orthogonal matrix $V$ is the matrix of eigenvectors of

$$R = \frac{1}{\sigma^2 p} Y^\top G_{xx}^+ Y = V \Omega V^\top, \tag{F.18}$$

for $G_{xx}^+$ the pseudoinverse of $G_{xx}$, and the scalars $z_k$ are in turn defined in terms of the eigenvalues $\Omega_{kk} = \omega_k$ as

$$1 - \alpha = z_k - \alpha\sigma^{-2(d-1)}z_k^{-(d-1)}\omega_k; \tag{F.19}$$

we note that Li and Sompolinsky [11] use variables $u_{k0} = \sigma^2 z_k$.

As we are interested in the limit $\alpha \downarrow 0$, it is useful to write the implicit equation for $z_k$ as

$$z_k = 1 + \alpha(\sigma^{-2L}z_k^{-(d-1)}\omega_k - 1), \tag{F.20}$$

hence we expect $z_k \to 1$ as $\alpha \downarrow 0$. Thus, we have

$$[M_\ell]_{kk} \to \ell, \tag{F.21}$$

which gives

$$V M_\ell V^\top \to \ell I_{n_d}. \tag{F.22}$$

Using the expansion $(1 - n_d/n)^\ell = 1 - n_d\ell/n + \mathcal{O}(n^{-2})$, we therefore find that

$$\sigma^{-2(\ell+1)}\langle K^{(\ell)}\rangle \sim G_{xx} + \frac{n_d\ell}{n}(\sigma^{-2d}G_{yy} - G_{xx}) \tag{F.23}$$

in the limit in which $n_d/n \downarrow 0$ and $p/n \downarrow 0$. Therefore, combining this result with that of the previous subsection, our result (9) agrees with those of Aitchison [10] and of Li and Sompolinsky [11] in the appropriate limit. Whether the full result of Li and Sompolinsky [11] agrees with that of Aitchison [10] is an interesting question, but is well beyond the scope of the present work.

## G   Predictor statistics and generalization in deep linear networks

Though the main focus of our work is on the asymptotics of representation learning, we have also computed the leading finite-width corrections to the predictor statistics. Though one can derive the analogy of Conjecture 1 for the predictor statistics of a general BNN with linear readout, the resulting formula is not particularly illuminating. We will therefore present results only for linear networks. As was true of the hidden layer kernels of deep linear networks, this calculation can be performed either using methods similar to those described in Appendix B or Appendix D. As the steps are largely identical to those calculations, we only briefly summarize the results.

In short, we fix a test dataset $\hat{\mathcal{D}} = \{(\hat{\mathbf{x}}_\mu, \hat{\mathbf{y}}_\mu)\}_{\mu=1}^{\hat{p}}$ of $\hat{p}$ examples, and define the Gram matrices

$$(G_{\hat{x}\hat{x}})_{\hat{\mu}\hat{\nu}} \equiv n_0^{-1}\hat{\mathbf{x}}_{\hat{\mu}} \cdot \hat{\mathbf{x}}_{\hat{\nu}}, \tag{G.1}$$

$$(G_{\hat{y}\hat{y}})_{\hat{\mu}\hat{\nu}} \equiv n_d^{-1}\hat{\mathbf{y}}_{\hat{\mu}} \cdot \hat{\mathbf{y}}_{\hat{\nu}}, \tag{G.2}$$

$$(G_{x\hat{x}})_{\mu\hat{\mu}} \equiv n_0^{-1}\mathbf{x}_\mu \cdot \hat{\mathbf{x}}_{\hat{\mu}}, \quad \text{and} \tag{G.3}$$

$$(G_{y\hat{y}})_{\mu\hat{\nu}} \equiv n_d^{-1}\mathbf{y}_\mu \cdot \hat{\mathbf{y}}_{\hat{\nu}}. \tag{G.4}$$

Introducing appropriate source terms to allow us to compute predictor statistics, we then proceed perturbatively as before, assuming that the combined input Gram matrix

$$\begin{bmatrix} G_{xx} & G_{x\hat{x}} \\ G_{x\hat{x}}^\top & G_{\hat{x}\hat{x}} \end{bmatrix} \tag{G.5}$$

is invertible. Again, the final result can be extended to the case in which this matrix is not invertible by a continuity argument.

Our notation in this appendix will follow that of Appendix B rather than Appendix D in that we will introduce matrices

$$K_\infty \equiv \sigma_1^2 \cdots \sigma_{d-1}^2 G_{xx}, \tag{G.6}$$

$$\hat{R}_\infty \equiv \sigma_1^2 \cdots \sigma_{d-1}^2 G_{x\hat{x}}, \quad \text{and} \tag{G.7}$$

$$\hat{K}_\infty \equiv \sigma_1^2 \cdots \sigma_{d-1}^2 G_{\hat{x}\hat{x}} \tag{G.8}$$

to denote the blocks of the infinite-width kernel of the last hidden layer, rather than introducing scalar parameters to represent the products of variances. This will make our expressions somewhat more compact than they would be under the conventions of Appendix D.

## G.1 Predictor statistics

Defining the matrix $\hat{F}_{\hat{\mu}j} \equiv f_j(\hat{\mathbf{x}}_{\hat{\mu}})$, we find that the mean predictor can be written compactly as

$$\langle \hat{F} \rangle = \hat{R}_\infty^\top \left[ \Gamma^{-1} - \frac{1}{\beta \sigma_d^2} \left( \sum_{\ell=1}^{d-1} \frac{1}{n_\ell} \right) \Gamma^{-1} M \Gamma^{-1} \right] Y + \mathcal{O}(n^{-2}) \tag{G.9}$$

for

$$M \equiv \Gamma^{-1} K_\infty + \text{tr}(\Gamma^{-1} K_\infty) I_p - n_d(\sigma_d^{-2} \Gamma^{-1} G_{yy} \Gamma^{-1} - \Gamma^{-1}) K_\infty. \tag{G.10}$$

The predictor covariance is given as

$$\begin{aligned}
\sigma_d^{-2} &\, \text{cov}(\hat{F}_{\hat{\mu}j}, \hat{F}_{\hat{\nu}k}) \\
&= (\hat{K}_\infty - \hat{R}_\infty^\top \Gamma^{-1} \hat{R}_\infty)_{\hat{\mu}\hat{\nu}} \delta_{jk} \\
&\quad + \left( \sum_{\ell=1}^{d-1} \frac{1}{n_\ell} \right) \Bigg[ \hat{M}_{\hat{\mu}\hat{\nu}} \delta_{jk} \\
&\qquad\qquad + \sigma_d^{-2} (Y^\top \Gamma^{-1} K_\infty \Gamma^{-1} Y)_{jk} (\hat{K}_\infty - \hat{R}_\infty^\top \Gamma^{-1} \hat{R}_\infty)_{\hat{\mu}\hat{\nu}} \\
&\qquad\qquad - \frac{1}{\beta \sigma_d^4} (Y^\top \Gamma^{-1} K_\infty \Gamma^{-1} Y)_{jk} (\hat{R}_\infty^\top \Gamma^{-2} \hat{R}_\infty)_{\hat{\mu}\hat{\nu}} \\
&\qquad\qquad + \frac{1}{\beta^2 \sigma_d^6} (Y^\top \Gamma^{-2} \hat{R}_\infty)_{j\hat{\nu}} (Y^\top \Gamma^{-2} \hat{R}_\infty)_{k\hat{\mu}} \Bigg] \\
&\quad + \mathcal{O}(n^{-2})
\end{aligned} \tag{G.11}$$

for

$$\begin{aligned}
\hat{M} &\equiv -\text{tr}(\Gamma^{-1} K_\infty)(\hat{K}_\infty - \hat{R}_\infty^\top \Gamma^{-1} \hat{R}_\infty) + \frac{1}{\beta \sigma_d^2} \text{tr}(\Gamma^{-1} K_\infty) \hat{R}_\infty^\top \Gamma^{-2} \hat{R}_\infty - \frac{1}{\beta^2 \sigma_d^4} \hat{R}_\infty^\top \Gamma^{-3} \hat{R}_\infty \\
&\quad + n_d \frac{1}{\beta^2 \sigma_d^4} \hat{R}_\infty^\top \Gamma^{-1} (\sigma_d^{-2} \Gamma^{-1} G_{yy} \Gamma^{-1} - \Gamma^{-1}) \Gamma^{-1} \hat{R}_\infty.
\end{aligned} \tag{G.12}$$

The mean and covariance of the training set predictor $F_{\mu j} \equiv f_j(\mathbf{x}_\mu)$ can be obtained by setting $\hat{R}_\infty$ and $\hat{K}_\infty$ to $K_\infty$ in the above expressions.

## G.2 Bias-variance decompositions and the low-temperature limit

These results allow us to define thermal bias-variance decompositions of the form

$$\langle E \rangle = \frac{1}{2} \sum_{\mu=1}^p \| \langle \mathbf{f}(\mathbf{x}_\mu) \rangle - \mathbf{y}_\mu \|_2^2 + \frac{1}{2} \sum_{\mu=1}^p \sum_{k=1}^{n_d} \text{cov}[f_k(\mathbf{x}_\mu), f_k(\mathbf{x}_\mu)] \equiv E_b + E_v \tag{G.13}$$

for the mean training and test errors. However, the resulting expressions are not particularly illuminating except in the low-temperature limit $\beta \to \infty$. We will focus on the regime in which $G_{xx}$ (and thus $K_\infty$) is invertible, in which the underlying linear system $XW = Y$ is underdetermined and the training set can be interpolated. In this regime, $\Gamma^{-1} = K_\infty^{-1} + \mathcal{O}(\beta^{-1})$, and the mean predictor reduces to the least-norm pseudoinverse solution to the linear system, with mean training and test predictions of

$$\langle F \rangle = Y + \mathcal{O}(\beta^{-1}) \tag{G.14}$$

and

$$\langle \hat{F} \rangle = \hat{R}_\infty^\top K_\infty^{-1} Y + \mathcal{O}(\beta^{-1}) = G_{\hat{x}x}^\top G_{xx}^{-1} Y + \mathcal{O}(\beta^{-1}) = \hat{X} X^\top (XX^\top)^{-1} Y + \mathcal{O}(\beta^{-1}), \tag{G.15}$$

respectively. The training and test set covariances have low-temperature limits of

$$\text{cov}(F_{\mu j}, F_{\nu k}) = \mathcal{O}(\beta^{-1}) \tag{G.16}$$

and

$$\text{cov}(\hat{F}_{\hat{\mu}j}, \hat{F}_{\hat{\nu}k}) = \sigma_d^2(\hat{K}_\infty - \hat{R}_\infty^\top K_\infty^{-1}\hat{R}_\infty)_{\hat{\mu}\hat{\nu}}\left[\delta_{jk} + \left(\sum_{\ell=1}^{d-1}\frac{1}{n_\ell}\right)(\sigma_d^{-2}Y^\top K_\infty^{-1}Y - pI_{n_2})_{jk}\right]$$
$$+ \mathcal{O}(\beta^{-1}, n^{-2}), \tag{G.17}$$

respectively. Then, it is easy to see that both $E_b$ and $E_v$ are $\mathcal{O}(\beta^{-1})$, while

$$\hat{E}_b = \frac{1}{2}\|\hat{R}_\infty^\top K_\infty^{-1}Y - \hat{Y}\|_F^2 + \mathcal{O}(\beta^{-1}) \tag{G.18}$$

and

$$\hat{E}_v = n_d\sigma_d^2\text{tr}(\hat{K}_\infty - \hat{R}_\infty^\top K_\infty^{-1}\hat{R}_\infty)\left[1 + \left(\sum_{\ell=1}^{d-1}\frac{1}{n_\ell}\right)(\sigma_d^{-2}\text{tr}(K_\infty^{-1}G_{yy}) - p)\right]$$
$$+ \mathcal{O}(\beta^{-1}, n^{-2}). \tag{G.19}$$

Thus, at least to leading order, width affects the low-temperature test error only through the variance term. Substituting in the definition of $K_\infty$, we find that to leading order the test error decreases with increasing width if

$$\frac{1}{p}\text{tr}(G_{xx}^{-1}G_{yy}) > \sigma_1^2\cdots\sigma_d^2 \tag{G.20}$$

and increases with increasing width otherwise. This small-initialization condition is the generalization of that found by Li and Sompolinsky [11] to our asymptotic regime.

## G.3 Effects of alternative regularization temperature-dependence

In this appendix, we comment on the possibility of alternative temperature-dependent posteriors. This possibility arises from the interpretation of the Bayes posterior (B.4) as the equilibrium distribution of the Langevin dynamics

$$d\Theta^{(\ell)}(t) = -(\lambda(\beta)\Sigma\Theta + \nabla_\Theta E)dt + \sqrt{2\beta^{-1}}dB^{(\ell)}(t) \tag{G.21}$$

at inverse temperature $\beta$, where $B^{(\ell)}(t)$ is a standard Wiener process, $\Sigma$ is the diagonal matrix of prior variances, and $\lambda(\beta) = 1/\beta$. As elsewhere, we focus on the regime in which the training dataset can be linearly interpolated, in which the thermal variance of the test set predictions need not vanish. Moreover, it suffices to consider only the GP contributions; the finite-width corrections computed above do not change the qualitative results. In these statistics, the case of general $\lambda(\beta)$ is related to $\lambda(\beta) = 1/\beta$ by the replacement

$$\sigma_1^2\cdots\sigma_d^2 \leftarrow \frac{\sigma_1^2\cdots\sigma_d^2}{\beta^d\lambda(\beta)^d}. \tag{G.22}$$

Then, if we assume a low-temperature power-law dependence $\lambda(\beta) \sim \beta^\omega$ for simplicity, we find that the zero-temperature limits of the training set predictor mean and covariance are

$$\lim_{\beta\to\infty}\langle F\rangle = \begin{cases} 0 & \omega > 1/d - 1 \\ K_\infty(\sigma_d^{-2}I_p + K_\infty)^{-1}Y & \omega = 1/d - 1 \\ Y & \omega < 1/d - 1 \end{cases} \tag{G.23}$$

and

$$\lim_{\beta\to\infty}\text{cov}(F_{\mu j}, F_{\nu k}) = 0, \tag{G.24}$$

respectively, while those of the test set mean and covariance are

$$\lim_{\beta\to\infty}\langle\hat{F}\rangle = \begin{cases} 0 & \omega > 1/d - 1 \\ \hat{R}_\infty^\top(\sigma_d^{-2}I_p + K_\infty)^{-1}Y & \omega = 1/d - 1 \\ \hat{R}_\infty^\top K_\infty^{-1}Y & \omega < 1/d - 1 \end{cases} \tag{G.25}$$

and

$$\lim_{\beta\to\infty}\text{cov}(\hat{F}_{\hat{\mu}j}, \hat{F}_{\hat{\nu}k}) = \begin{cases} 0 & \omega > -1 \\ \sigma_d^2(\hat{K}_\infty - \hat{R}^\top K_\infty^{-1}\hat{R}_\infty)_{\hat{\mu}\hat{\nu}}\delta_{jk} & \omega = -1 \\ \infty & \omega < -1, \end{cases} \tag{G.26}$$

respectively. Therefore, taking $\lambda(\beta) = 1/\beta$ yields sensible zero-temperature infinite-width behavior for a linear network of any depth in the underdetermined regime.

## H Derivation of the average kernels for a depth-two network

In this appendix, we derive the average feature kernel for a network with a single (possibly nonlinear) hidden layer and a linear readout. This derivation is a simple extension of the perturbative derivation of Conjecture 1 in Appendix B, using the fact that the size of the terms in the expansion for two-layer networks can be directly controlled in terms of the inverse hidden layer width.

Concretely, we consider a network defined as

$$\mathbf{h}^{(1)} = \frac{\sigma_1}{\sqrt{n_0}} W^{(1)} \mathbf{x} \tag{H.1}$$

$$\mathbf{h}^{(2)} = \frac{\sigma_2}{\sqrt{n_1}} W^{(2)} \phi(\mathbf{h}^{(1)}) \tag{H.2}$$

$$\mathbf{f} = \mathbf{h}^{(2)}. \tag{H.3}$$

Our task is to control the prior cumulants of the hidden layer feature kernel

$$K_{\mu\nu} \equiv \frac{1}{n_1} \phi(\mathbf{h}_\mu^{(1)}) \cdot \phi(\mathbf{h}_\nu^{(1)}). \tag{H.4}$$

We can use the fact that the rows $[\mathbf{w}_j^{(1)}]^\top$ of $W^{(1)}$ are independent and identically distributed under the prior to obtain

$$[K_\infty]_{\mu\nu} = \mathbb{E}_{\mathcal{W}} K_{\mu\nu} \tag{H.5}$$

$$= \frac{1}{n_1} \sum_{j=1}^{n_1} \mathbb{E}_{\mathbf{w}_j^{(1)}} \left[ \phi\left( \frac{\sigma_1}{\sqrt{n_0}} \mathbf{w}_j^{(1)} \cdot \mathbf{x}_\mu \right) \phi\left( \frac{\sigma_1}{\sqrt{n_0}} \mathbf{w}_j^{(1)} \cdot \mathbf{x}_\nu \right) \right] \tag{H.6}$$

$$= \mathbb{E}[\phi(h_\mu^{(1)})\phi(h_\nu^{(1)}) : \mathbf{h}^{(1)} \sim \mathcal{N}(\mathbf{0}, \sigma_1^2 G_{xx})] \tag{H.7}$$

at any hidden layer width [12, 13]. Similarly, we can easily see that

$$\text{cov}_{\mathcal{W}}(K_{\mu\nu}, K_{\rho\lambda}) = \frac{1}{n_1} \left( \mathbb{E}[\phi(h_\mu^{(1)})\phi(h_\nu^{(1)})\phi(h_\rho^{(1)})\phi(h_\lambda^{(1)})] - [K_\infty]_{\mu\nu}[K_\infty]_{\rho\lambda} \right), \tag{H.8}$$

where $\mathbf{h}^{(1)} \sim \mathcal{N}(\mathbf{0}, \sigma_1^2 G_{xx})$, and that higher cumulants are $\mathcal{O}(n_1^{-2})$. Then, we can directly apply the result of Appendix B to conclude that

$$\langle K_{\mu\nu} \rangle = [K_\infty]_{\mu\nu} + \frac{1}{2} n_d \sum_{\rho,\lambda=1}^{p} (\sigma_d^{-2} \Gamma^{-1} G_{yy} \Gamma^{-1} - \Gamma^{-1})_{\rho\lambda} \text{cov}_{\mathcal{W}}(K_{\mu\nu}, K_{\rho\lambda}) + \mathcal{O}(n_1^{-2}) \tag{H.9}$$

for $\Gamma = \sigma_1^2 K_\infty + I_p/\beta\sigma_2^2$. Depending on the nonlinearity, this result may be continuous in $G_{xx}$, and therefore extensible to the non-invertible case via a continuity argument. In particular, as noted in Appendix D, this holds for a linear network.

To gain some intuition for how different choices of nonlinear activation function affect the learned representations, we consider the case in which $G_{xx}$ is diagonal. In this special case, the four-point term simplifies dramatically. In particular, we have

$$(K_\infty)_{\mu\nu} = \text{var}[\phi(h_\mu^{(1)})]\delta_{\mu\nu} + \mathbb{E}[\phi(h_\mu^{(1)})]\mathbb{E}[\phi(h_\nu^{(1)})] \tag{H.10}$$

and

$$\text{cov}_{\mathcal{W}}(K_{\mu\nu}, K_{\rho\lambda}) = \frac{1}{n_1} \Bigg( \text{var}[\phi(h_\mu^{(1)})^2]\delta_{\mu\nu}\delta_{\mu\rho}\delta_{\mu\lambda}$$

$$+ \text{var}[\phi(h_\mu^{(1)})] \text{var}[\phi(h_\nu^{(1)})](1 - \delta_{\mu\nu})(\delta_{\mu\rho}\delta_{\nu\lambda} + \delta_{\mu\lambda}\delta_{\nu\rho}) \Bigg), \tag{H.11}$$

which yields

$$\langle K_{\mu\nu} \rangle = (K_\infty)_{\mu\nu} + \frac{1}{2} \frac{n_2}{n_1} (\sigma_2^{-2} \Gamma^{-1} G_{yy} \Gamma^{-1} - \Gamma^{-1})_{\mu\nu}$$

$$\times \left[ \text{var}[\phi(h_\mu^{(1)})^2]\delta_{\mu\nu} + 2 \text{var}[\phi(h_\mu^{(1)})] \text{var}[\phi(h_\nu^{(1)})](1 - \delta_{\mu\nu}) \right] + \mathcal{O}(n_1^{-2}). \tag{H.12}$$

S28

Moreover, applying the Sherman-Morrison formula [3], we have

$$\frac{1}{\beta\sigma_2^2}\Gamma_{\mu\nu}^{-1} = \frac{\delta_{\mu\nu}}{\gamma_\mu} - \frac{1}{1 + \sum_{\rho=1}^{p} \mathbb{E}[\phi(h_\rho^{(1)})]^2/\gamma_\rho} \frac{\mathbb{E}[\phi(h_\mu^{(1)})]}{\gamma_\mu} \frac{\mathbb{E}[\phi(h_\nu^{(1)})]}{\gamma_\nu}, \tag{H.13}$$

where we have defined the vector $\gamma_\mu \equiv 1 + \beta\sigma_2^2 \operatorname{var}[\phi(h_\mu^{(1)})]$ for brevity. Thus, in this simple setting, activation functions with $\mathbb{E}\phi(h) \neq 0$ yield qualitatively different behavior from those with $\mathbb{E}\phi(h) = 0$: non-vanishing $\mathbb{E}\phi(h)$ introduces a rank-1 component in the GP kernel, which in turn couples elements of $G_{yy}$ in the leading finite-width correction.

# I   Numerical methods

In this appendix, we describe the numerical methods used in our experiments. We perform our simulations by sampling network parameters at each time step of the Langevin update G.21 after some large burn-in period when the loss function stabilizes around a fixed number. We used Euler-Maruyama method [14] to obtain the discretized Langevin equation:

$$\Theta(t+1) - \Theta(t) = -\beta^{-1}\Theta(t)dt - \nabla_\Theta E(t)dt + \xi\sqrt{2\beta^{-1}dt}, \tag{I.1}$$

where $\xi \sim \mathcal{N}(0,1)$ is a standard Gaussian random variable sampled i.i.d. at each time step and $dt$ is the time step. The first, second and last terms represent the weight decay, the gradient descent update and the stochastic Wiener process, respectively. We also used stochastic gradient methods for posterior sampling such as Stochastic Gradient Langevin Dynamics [15, 16], although obtained the best results via full-batch gradient updates.

We used the Neural Tangents framework [17] and PyTorch deep learning library [18] to generate the neural networks and trained them according to the discretized full-batch Langevin update rule. A typical burn-in time was $\sim 2 \times 10^6$ iterations and after that the parameters were sampled over $\sim 2 \times 10^6$ iterations where we chose a learning rate of $dt \sim 10^{-4}$. Simulations have been performed on a cluster with NVIDIA Tesla V100 GPU's with 32 GB RAM and a typical simulation run took $\sim 2 - 6$ hr depending on the architecture and the network width. All code used throughout this work can be reached at `https://github.com/Pehlevan-Group/finite-width-bayesian/`.

All figures shown here are results of a single instance of a trained neural network on a fixed dataset. Since we performed all our experiments with $\beta = 1$, we observed that the different initializations of a network did not influence the final posterior mean due to the weight decay and long burn-in periods.

Throughout all experiments, the MNIST digits were downsized from $28 \times 28$ pixels to $10 \times 10$ pixels without distorting the original digits. This was done to accelerate the training process since large input dimensions would take an order of magnitude more time to obtain well estimated posterior means. We considered 10-dimensional outputs corresponding to one-hot encoded digits. Both inputs and labels were ordered according to their class. Figure 2 shows an example of MNIST digits and the input $G_{xx}$ and output $G_{yy}$ Gram matrices.