# OpenReview forum: "Asymptotics of representation learning in finite Bayesian neural networks"
_NeurIPS.cc/2021/Conference — NeurIPS 2021 Poster_

### Official Review · Reviewer_jCKB · 2021-06-25

**Rating:** 6
**Confidence:** 3

**Summary:**

This paper studies the problem of leading-order perturbations to average feature kernels of Bayesian finite-but-wide neural networks (NNs). The paper is mainly of theoretical contribution and the main result establishes a conjecture for the leading order corrections for general NN architectures under linear readout and squared loss. The authors then delve into limiting situations of tempered posteriors before analysing two tractable cases: deep linear NNs and single-hidden layer non-linear NNs. Experimental evidence on toy settings support their claims.

**Limitations And Societal Impact:**

The authors adequately address the limitations of their work in the conclusion section. They do not discuss potential negative societal impacts though I do not believe there are any.

**Main Review:**

*Originality*: This is to my knowledge the first paper to study the leading-order pertubations to average feature kernels of general wide BNNs, and certainly I believe Conjecture 1 is original. The case of deep linear networks is closely linked to the works of Aitchison [9] and Li and Sompolinsky [15], and the authors detail this comparison in Appendix E, though the authors stress that their contributions of leading-order corrections extend these previous works.

*Quality*: I believe the submission is technically sound, and written in a fair way without overstating their claims (which I commend). However, the work seems a bit light and incomplete and thus is slightly unsatisfying. Indeed, the main contribution is a non-rigorous conjecture and the analytic cases considered do not give me a great deal of insight into the inner workings of feature learning in BNNs (which is a very interesting question).

Some of the questions/thoughts I have which I believe would improve the quality of this work if addressed:
- How can you make conjecture 1 rigorous?
- Do you get posterior concentration of the feature kernels with large p? I realise you say this is left for future work in  line 277-278, but this is quite a crucial question I think for a Bayesian perspective of wide NNs
- Related to above, can we say anything about other moments of the finite-width corrections to feature kernels besides the posterior mean?
- I realise that analytic tractability restricts you to deep linear and single hidden-layer, but unfortunately that means we don't see any notion of 'hierarchical' feature learning (i.e. simpler features being learned in the early layers), which intuitively is important for (B)NNs.
- Is it wrong to think of having n_d outputs per input, as having a dataset size of $p\times n_d$ instead of p? In which case makes the paragraph from lines 130-134 slightly unsurprising. Indeed I would expect the second term in eq 8 to scale as $O(p\times n_d)$, but the dependence on p isn't clear?
- Can the authors empirically demonstrate that your theory holds in more practically relevant settings. E.g. using the whole MNIST dataset as opposed to 100 points. What about CNNs with many channels?

*Clarity* The paper is well written and organised.

*Significance* The question of feature learning (or lack of it) in wide (B)NNs is interesting to me and I believe to the community as a whole. The results in this submission are a new angle towards understanding the interplay between the infinite-width 'lazy' GP limit and the finite-width 'feature-learning' regime, and in that sense are significant. Although see my comments in *Quality* about questions that I wish were addressed in this submission.

Missed citations:
- https://arxiv.org/abs/2106.06615 concurrent work to Zavatone-Veth and Pehlevan [6]
- https://arxiv.org/abs/2006.10541 prove limiting posterior converges weakly to GP limit.

Minor points:
- the notation p for dataset size is confusing.
- The conclusion focuses in on the NNGP kernel but this seems like you're selling yourself short as you also discuss earlier layers' kernels.
- The expectation with respect to \mathcal{W} e.g. in Eq 8 doesn't seem to be defined. I take it means the prior over weights?

**Time Spent Reviewing:**

4

---

> ### Author Response · Authors · 2021-08-10
> **Response to Reviewer jCKB**
>
> *[...]*
>
> ***Quality***
>
> *I believe the submission is technically sound, and written in a fair way without overstating their claims (which I commend). However, the work seems a bit light and incomplete and thus is slightly unsatisfying. Indeed, the main contribution is a non-rigorous conjecture and the analytic cases considered do not give me a great deal of insight into the inner workings of feature learning in BNNs (which is a very interesting question).*
>
> Thank you for this comment. As we discuss in detail in our responses to "Common concerns," approaches which rely on prior cumulants are limited to simple analytic cases. Moreover, we hope our proposed changes to the statement of Conjecture 1 help address your concerns regarding the completeness of our work.
>
> *Some of the questions/thoughts I have which I believe would improve the quality of this work if addressed:*
>
> - *How can you make conjecture 1 rigorous?*
>
>     Thank you for this question. One approach would be to perform a rigorous version of the layer-by-layer analysis of Yaida [11] or of Appendix D of our work. Such an analysis would be similar in spirit to recent work by Hanin on the infinite-width limit of fully connected networks (https://arxiv.org/abs/2107.01562). However, this would likely require separate proofs for different network architectures. An alternative approach would be to attempt to prove an analog of the 'Master Theorem' of Yang's Tensor Programs framework (e.g. 2.10 of https://arxiv.org/abs/2009.10685) that describes finite-width corrections to expectations of functions of variables within the program (rather than limiting values of empirical means). This approach would likely require significant additional effort, but could allow asymptotic analysis of a broader range of architectures. Though we believe a rigorous proof of Conjecture 1 is beyond the scope of the present work, we can include these speculative comments in our discussion if you believe that they would improve the paper.
>
> - *Do you get posterior concentration of the feature kernels with large p? I realise you say this is left for future work in line 277-278, but this is quite a crucial question I think for a Bayesian perspective of wide NNs*
>
>     Thank you for emphasizing what we agree is an important point. Unfortunately, the perturbative framework used in this paper cannot easily be adapted to the regime in which $p$ is comparable to $n$, since the particular dependence on dataset size is both implicit and data-dependent in our setup. We note that recent work by Naveh and Ringel (https://arxiv.org/abs/2106.04110) has aimed to address the large-$p$ regime in a qualiatively similar manner (describing the prior in terms of cumulants), but their approach shares many limitations with ours (in particular, the need to compute prior cumulants, which is generally intractable for deep networks). Therefore, we must reserve careful study of posterior concentration for future work.
>
> - *Related to above, can we say anything about other moments of the finite-width corrections to feature kernels besides the posterior mean?*
>
>     Thank you for this suggestion. Our approach - perturbative approximation of the joint posterior cumulant generating function of the feature kernels - could certainly be used to compute higher moments of the kernels. In particular, one could obtain perturbative expansions for higher posterior cumulants in terms of higher prior cumulants similar to that given in (8) for the posterior mean. In our revised manuscript, we will note this point explicitly.
>
> - *I realise that analytic tractability restricts you to deep linear and single hidden-layer, but unfortunately that means we don't see any notion of 'hierarchical' feature learning (i.e. simpler features being learned in the early layers), which intuitively is important for (B)NNs.*
>
>     We agree that the inability to study hierarchical' feature learning is an important limitation of our methods, which, as we discuss in our responses to "Common concerns," is shared by many recent studies of BNNs at large but finite width. We will comment on this point in our revised discussion.
>
> - *Is it wrong to think of having $n\_d$ outputs per input, as having a dataset size of $p \times n\_{d}$ instead of $p$? In which case makes the paragraph from lines 130-134 slightly unsurprising. Indeed I would expect the second term in eq 8 to scale as $O(p \times n\_{d})$, but the dependence on $p$ isn't clear?*
>
>     Thank you for this comment; we agree that this intuition is reasonable. As you note, the dependence on $p$ is implicit in the dimensionality of the sums.
>
> - *Can the authors empirically demonstrate that your theory holds in more practically relevant settings. E.g. using the whole MNIST dataset as opposed to 100 points. What about CNNs with many channels?*
>
>     Thank you for this suggestion. As we discuss in detail in our response to questions regarding application of our results to more realistic settings, we are currently working on experiments that use larger datasets. As we currently lack exact formulae for the required covariances for deep linear CNNs with many channels, we must reserve such tests for future work.
>
> *[...]*
>
> ***Missed citations:***
>
> - *https://arxiv.org/abs/2106.06615 concurrent work to Zavatone-Veth and Pehlevan [6]*
>
>     As noted in our response to Reviewer **QD2J**, this paper was not posted until after the submission deadline. We will cite it in our revised manuscript.
>
> - *https://arxiv.org/abs/2006.10541 prove limiting posterior converges weakly to GP limit.*
>
>     Thank you for this suggestion; we will add this reference to our revised manuscript.
>
> ***Minor points:***
>
> - *the notation p for dataset size is confusing.*
>
>     Thank you for this comment. Using $p$ to denote the dataset size is a common convention among the community of physicists working on neural networks (see, for instance, Li and Sompolinsky as cited above, or Engel and van den Broeck, *The Statistical Physics of Learning*).  We note that we already use $n$ to denote layer widths. If you believe another choice would be clearer, we are happy to adopt a different convention.
>
> - *The conclusion focuses in on the NNGP kernel but this seems like you're selling yourself short as you also discuss earlier layers' kernels.*
>
>     Thank you for this suggestion. We will revise the conclusion to more clearly state our contribution. Moreover, to avoid confusion, we intend to adopt the notation $K\_{\infty}^{(\ell)}$ throughout for the deterministic infinite-width values of the kernels, instead of $K\_{\mathrm{GP}}^{(\ell)}$ as in the submitted manuscript. We appreciate your feedback on this point.
>
> - *The expectation with respect to $\mathcal{W}$ e.g. in Eq 8 doesn't seem to be defined. I take it means the prior over weights?*
>
>     Thank you for this comment; your understanding is correct. We previously defined $\mathbb{E}\_{\mathcal{W}}$ only in words below (8); we will do so more explicitly.

---

> > ### Author Response · Authors · 2021-08-20
> > **Additional response to Reviewer jCKB**
> >
> > To concretely address the referee's question regarding finite-width corrections to cumulants beyond the posterior mean, we have computed the leading correction to the posterior covariance of a given hidden layer observable $O$:
> > $$
> > \mathrm{cov}(O\_{\rho \gamma},O\_{\omega \chi}) = \mathrm{cov}\_{\mathcal{W}}(O\_{\rho\gamma}, O\_{\omega \chi} ) + \frac{1}{2} n\_{d} \sigma\_{d}^{2} \sum\_{\mu,\nu=1}^{p} (\beta^2 \Gamma^{-1} G\_{yy} \Gamma^{-1} - \beta \Gamma^{-1})\_{\mu\nu} \mathbb{E}\_{\mathcal{W}}[ (O - \mathbb{E}\_{\mathcal{W}} O)\_{\rho\gamma} (O - \mathbb{E}\_{\mathcal{W}} O)\_{\omega \chi} (K^{(d-1)} - \mathbb{E}\_{\mathcal{W}} K^{(d-1)})\_{\mu\nu} ] + \ldots .
> > $$
> > This result is of the same form as the correction to the posterior mean. For an MLP, considering a hidden-layer kernel $K^{(\ell)}$, the prior covariance $\mathrm{cov}\_{\mathcal{W}}(K^{(\ell)}\_{\rho\gamma}, K^{(\ell)}\_{\omega \chi} )$ will be $\mathcal{O}(n^{-1})$, while the third cumulant $\mathbb{E}\_{\mathcal{W}}[ (K^{(\ell)} - \mathbb{E}\_{\mathcal{W}} O)\_{\rho\gamma} (K^{(\ell)} - \mathbb{E}\_{\mathcal{W}} O)_{\omega \chi} (K^{(d-1)} - \mathbb{E}\_{\mathcal{W}} K^{(d-1)})\_{\mu\nu} ]$ will be $\mathcal{O}(n^{-2})$. Higher-order posterior cumulants will behave similarly: for the $q$-th posterior cumulant, the leading term will be the $q$-th prior cumulant, while the first subleading correction will involve a joint prior cumulant of the observable of interest with the kernel $K^{(d-1)}$. In our revised manuscript, we will state these points explicitly, and give a revised perturbative derivation of Conjecture 1 that makes the behavior of higher-order cumulants more transparent.

---

> > > ### Comment · Reviewer_jCKB · 2021-08-23
> > > **Thanks**
> > >
> > > Thank you for the detailed response, particularly for the explanation of ideas to make conjecture 1 rigorous (it might be nice to see this in a discussion section), and the more general version of conjecture 1. I am raising my rating to a 6 as a result. I hope to see the new simulations in a later version of the paper. It is also apparent to me that my confidence score was too high (as you can likely tell I am not from a physics background), so I am lowering it to 3.

---

### Official Review · Reviewer_1KNh · 2021-07-03

**Rating:** 7
**Confidence:** 3

**Summary:**

Work analyzing representation learning using finite-width corrections to Bayesian neural networks.  Specifically, they look at deep linear networks and networks with one nonlinear hidden layer.


**Limitations And Societal Impact:**

The idea that the paper is "non-quantitative" is a bit strong!

**Main Review:**

Work presenting insightful results on neural network theory in simple cases.  The methods are clearly developed in the Appendix.  I have no technical concerns.  There are a couple of points that I'd be interested to see discussed further (but this can easily be done in a revision):

Possibly the key weakness of this work is that it uses simplistic deep linear networks, or networks with one nonlinear layer.  Of course, this allows the authors to come up with simple, interpretable results that give solid insight into the underlying processes.  But it would be worth presenting the case that these corrections are relevant in "real world" cases (i.e. deep nonlinear networks).

Likewise, the empirical findings of Aitchison (2019) that upper-layer kernels are radically different from the infinite limit.  I'd be very interested to know whether the results of Aitchison (2019) differ from the perturbative approach, and whether that tells us anything about "how far" these low-order perturbations can take you from the underlying infinite limit.

Appendix is very clear!  The "Preliminary Technical Results" section is great.

Good literature review, and extensive connections to existing literature.

While Zavatone-Veth and Pehlevan (2021) appears in the Related Work, I wasn't at all clear about the relationship between the papers (at least from the discussion there).  Would being _really_ clear about the high-level and technical relationship.  At a high-level, my guess would be that this papers looks at representation learning, while that paper focuses on the distribution of a single output variable?  But it would also be useful to sketch the technical relationship, if any!

**Time Spent Reviewing:**

3

---

> ### Author Response · Authors · 2021-08-10
> **Response to Reviewer 1KNh**
>
> *[...]*
>
> *Possibly the key weakness of this work is that it uses simplistic deep linear networks, or networks with one nonlinear layer. Of course, this allows the authors to come up with simple, interpretable results that give solid insight into the underlying processes. But it would be worth presenting the case that these corrections are relevant in "real world" cases (i.e. deep nonlinear networks).*
>
> Thank you for this suggestion. As detailed in our response to "Common concerns," we are currently working to test our perturbative predictions against experimental results for BNNs trained on larger datasets (MNIST and CIFAR-10).
>
> *Likewise, the empirical findings of Aitchison (2019) that upper-layer kernels are radically different from the infinite limit. I'd be very interested to know whether the results of Aitchison (2019) differ from the perturbative approach, and whether that tells us anything about "how far" these low-order perturbations can take you from the underlying infinite limit.*
>
> This is a great observation! We take the reviewer's comment as a question about the limits of the explanatory power of perturbation theory. In our experiments, we tested modest widths of $n=100$, and the match between experiment and theory was still reasonable. However, with regards to the link to Aitchison (2019), we note that his empirical results were obtained using a deep ResNet with 34 layers, hence the depth-to-width ratio considered there was far larger than that of the networks we tested. Roughly speaking, Aitchison obtained correlation coefficients of around 0.1-0.5 between the finite and GP kernels for those networks; we obtain correlation coefficients of around 0.8 for the fully-connected linear networks we consider. Accurately predicting the average kernels of very deep networks would thus require going to higher order in perturbation theory. In our revised manuscript, we will pose detailed study of this limit as an interesting objective for future work.
>
> *[...]*
>
> *While Zavatone-Veth and Pehlevan (2021) appears in the Related Work, I wasn't at all clear about the relationship between the papers (at least from the discussion there). Would being really clear about the high-level and technical relationship. At a high-level, my guess would be that this papers looks at representation learning, while that paper focuses on the distribution of a single output variable? But it would also be useful to sketch the technical relationship, if any!*
>
> Thank you for this question. In their work (https://arxiv.org/abs/2104.11734), Zavatone-Veth and Pehlevan noted that the heavy-tailed nature of the function space prior for a single training example is not captured by the perturbative Edgeworth approximation to that density. Though the perturbative derivation of Conjecture 1 given in Appendix B does not rely on the Edgeworth approximation, the perturbative study of deep linear networks in Appendices C, D, F, G, and H as well as the previous work of Yaida [11] rely upon closely related techniques. Our intention was thus to to give a concrete illustration of the caveat that, even if one were to continue the perturbative approach of our paper to higher orders, one might not be able to capture all features of the true finite-width behavior. We will revise that section of the discussion to make this connection more clear.

---

### Official Review · Reviewer_8cN5 · 2021-07-14

**Rating:** 7
**Confidence:** 3

**Summary:**

This paper shows analytically and numerically that the leading finite-width corrections to the average hidden layer kernels of any fully-connected feedforward networks with linear readout and least-squares loss have a largely prescribed form :  these assumptions fix the dependency of the correction on the target outputs, the size of the correction scale with the ratio of the number of outputs to the typical hidden layer width. Authors explicitly compute the leading finite-width corrections for deep linear networks 	and networks with a single nonlinear hidden layer . Theyshow how certain directions in the output kernel matrix can be enhanced or suppressed in the hidden layer representations depending on the structure of the input similarities

This is equivalent to considering the equilibrium distribution of Langevin sampling of the parameters at inverse temperature β. Moreover, by tuning β, one can then adjust whether the posterior is dominated by the prior (β ≪ 1) or the likelihood (β ≫ 1).

Authors conjecture, non-rigorously, that these results should extend to Bayesian neural network with different architectures


**Limitations And Societal Impact:**

yes

**Main Review:**

This is a very interesting paper, of particular interest as, despite the numerous critics of infinite-width neural networks low representational power, their learned features are not well characterized.

Authors derive the leading finite-width corrections to the average hidden layer kernels of any fully-connected feedforward networks with linear readout and least-squares loss rigorously. I find the high- and low-temperature limits of the leading correction as well as the recovery of Aitchison implicit recurrence very compelling.

This is a first step towards understanding the features learned by Infinite width BNNs, but conjecturing that results  should extend to other architectures is an overstatement to me. Neural Networks behave in a very complex way and most related papers rely on strong assumptions on the shape of the networks to make any conclusions. The analysis of the wide “narrow bottleneck” is interesting and aims at generalizing the results but is not enough.

For better readability, Figure 1 should be updated. It is not clear that authors obtain the predicted 1/n decay with increasing width and the linear scaling with the depth, as the axis are not correctly labeled. What do the markers represent? Why do error bars don’t appear? Moreover, both figures can be made bigger.


**Time Spent Reviewing:**

4

---

> ### Author Response · Authors · 2021-08-10
> **Response to Reviewer 8cN5**
>
> *This is a very interesting paper, of particular interest as, despite the numerous critics of infinite-width neural networks low representational power, their learned features are not well characterized.*
>
> *Authors derive the leading finite-width corrections to the average hidden layer kernels of any fully-connected feedforward networks with linear readout and least-squares loss rigorously. I find the high- and low-temperature limits of the leading correction as well as the recovery of Aitchison implicit recurrence very compelling.*
>
> *This is a first step towards understanding the features learned by Infinite width BNNs, but conjecturing that results should extend to other architectures is an overstatement to me. Neural Networks behave in a very complex way and most related papers rely on strong assumptions on the shape of the networks to make any conclusions. The analysis of the wide “narrow bottleneck” is interesting and aims at generalizing the results but is not enough.*
>
> Thank you for your comments. As we describe in detail in our discussion of the generality of Conjecture 1 under "Common concerns," our non-rigorous perturbative argument for its general form does not rely on specific assumptions on the network architecture beyond fully-connected linear readout and the existence of the GP limit. Moreover, we note that Appendix H presents a detailed argument for the case of representation learning in deep linear networks with arbitrary feedforward skip connections. We therefore believe that conjecturing that our results extend to other architectures is plausible.
>
> *For better readability, Figure 1 should be updated. It is not clear that authors obtain the predicted 1/n decay with increasing width and the linear scaling with the depth, as the axis are not correctly labeled. What do the markers represent? Why do error bars don’t appear? Moreover, both figures can be made bigger.*
>
> Thank you for this suggestion. We will update the axes of Figure 1a. The markers in 1a represent the Frobenius norm of the difference between the experimental kernel and the NNGP over realizations; as noted in the caption, the error bars do not appear as they are smaller than the markers. While the slope of the log-log plot and match with the theory curve (solid lines) confirms the 1/n scaling, the linear scaling with depth can be seen from the ratios of the curves between each other. We agree that this could have been made clearer, and will also plot a line where each curve is divided by the layer index to show that they overlap. An updated version of this figure can be viewed anonymously at the following https URL: https://ibb.co/GkjjLwS.
>
> We will increase the size of both figures in our updated manuscript to the extent possible given length constraints.

---

> > ### Comment · Reviewer_8cN5 · 2021-08-31
> > **Re: Response to Reviewer 8cN5**
> >
> > I thank the authors for their thoughtful response, especially to my concerns about Conjecture 1 under "Common concerns".
> > The statement of the revised version of Conjecture 1 outlines much more clearly the assumptions and heuristic argument.
> > This was my main concern and I am thus raising my rating to 7 (I also think the additional experiments described under "Common concerns" will make the paper stronger).
> >
> > I thank the authors for sharing an anonymous version of the figure, which looks better to me. I trust the authors to make bigger figures and remove "blank spaces".

---

### Official Review · Reviewer_QD2J · 2021-07-26

**Rating:** 7
**Confidence:** 3

**Summary:**

Layer outputs of neural networks with Gaussian priors are proved to be Gaussian processes at infinite width. The paper is focused on obtaining a characterization of layer outputs at finite width. Based on the NNGP kernel resulting in the wide regime, the main result is the introduction of perturbative corrections to the average of this kernel. These corrections appear in finite-width neural networks and don’t exist in the limit.

**Contributions**:
* Updated description of Yaida [11] on the difference between finite and infinite wide neural networks
* Feature learning interpretation

**Limitations**:
* Not rigorous
* Works on very simple setting but no additional experiments
* Not evident how to use the results

**Limitations And Societal Impact:**

The authors adequately addressed the limitations and potential negative societal impact of their work.

**Main Review:**

**Questions**:

* What is the difference with the kernel used in the paper and introduced in Lee et al. “Deep neural networks as Gaussian processes”? I guess the regime when widths of layers tend to infinity sequentially can be easily addressed in the paper?
* “As a possible complication, one could imagine a situation in which the activation distributions are ‘rotated’ during inference without changing the kernels. However, Yang and Hu [8] have shown that this cannot happen in the infinite-width limit…” Could you please precise the place in the manuscript?
* The authors cite paper [16] in related works about the heavy-tailed nature of the prior densities over the outputs. I wonder why not cite other works that stated heavy tails like “Precise characteristics..” (the concurrent work of [16]) and Vladimirova et al. “Understanding priors…”

**Quality**:

I find the paper and the appendix well-written and I believe that mathematics is correct.

**Significance**:

The results follow recent results of Yaida [11] and can be well-placed in the literature. However, I would like to see applications of these corrections (at least possible).

**Conclusion**:

The paper has a solid theory. I recommend publication even if it misses the application part.

**Minor**:
* “Pertubative”
* “The second class of corrections are”
* missed comma in (A.2)
* I would reintroduce the notations in the appendix as well, like $E$ for error function, $n$, etc


**Time Spent Reviewing:**

30h

---

> ### Author Response · Authors · 2021-08-10
> **Response to Reviewer QD2J**
>
> *[...]*
>
> ***Questions:***
>
> - *What is the difference with the kernel used in the paper and introduced in Lee et al. “Deep neural networks as Gaussian processes”? I guess the regime when widths of layers tend to infinity sequentially can be easily addressed in the paper?*
>
>     Thank you for this question; we agree that it is important to clarify the relationship between the kernels considered here and in Lee et al. In our notation, for a fully-connected feedforward network with preactivations recursively defined as $\mathbf{h}\_{\mu}^{(\ell)} = W^{(\ell)} \phi(\mathbf{h}\_{\mu}^{(\ell-1)}) + \mathbf{b}^{(\ell)}$, we consider the postactivation kernels $K^{(\ell)}\_{\mu\nu} \equiv n\_{\ell}^{-1} \phi(\mathbf{h}\_{\mu}^{(\ell)}) \cdot \phi(\mathbf{h}\_{\nu}^{(\ell)})$, while Lee et al. considered the corresponding preactivation kernels $H^{(\ell)}\_{\mu\nu} \equiv n\_{\ell}^{-1} \mathbf{h}\_{\mu}^{(\ell)} \cdot \mathbf{h}\_{\nu}^{(\ell)}$. For the deep linear networks considered in Section 4.1 and Appendices C, D, F, G, and H, these definitions coincide, i.e., $K^{(\ell)} = H^{(\ell)}$. We will note this point explicitly in our revised manuscript.
>
>     More generally, your comment led us to carefully reconsider our perturbative argument for Conjecture 1. As detailed in our remarks under "Common concerns," we realized that this heuristic argument should apply to objects other than just the postactivation kernels.
>
>     Moreover, we agree that the sequential infinite-width limit can be easily addressed. Concretely, one could compute the covariances in (8) in this limit; in the fully-connected case the desired results would again follow from Yaida [11]'s approximations for all layers large but finite. In the case of deep linear networks, the calculation described in Appendix D would be particularly simple in the sequential infinite-width limit, since the hidden layers are integrated out one by one in the same order as their widths are taken to infinity. We will note this point explicitly in our revised manuscript.
>
> - *“As a possible complication, one could imagine a situation in which the activation distributions are ‘rotated’ during inference without changing the kernels. However, Yang and Hu [8] have shown that this cannot happen in the infinite-width limit…” Could you please precise the place in the manuscript?*
>
>     Thank you for this comment. This note was a regrettably imprecise reference to Theorem 3.5 of Yang and Hu [8], which shows that such rotation cannot occur for infinitely-wide networks of the architectures considered here when trained under SGD. However, as we are considering Bayesian networks with linear readout and Gaussian likelihood, such rotation would not affect inference, as the function-space posterior depends only on the kernel. We will therefore remove this comment from our revised manuscript.
>
> - *The authors cite paper [16] in related works about the heavy-tailed nature of the prior densities over the outputs. I wonder why not cite other works that stated heavy tails like “Precise characteristics..” (the concurrent work of [16]) and Vladimirova et al. “Understanding priors…”*
>
>     Thank you for this suggestion; we will cite both of these papers in our updated manuscript. We would like to emphasize that we did not cite Noci et al., "Precise characterization of the prior predictive distribution of deep ReLU networks" (https://arxiv.org/abs/2106.06615) because it was not posted on the arXiv until after the submission deadline.
>
> *[...]*
>
> ***Significance:*** *The results follow recent results of Yaida [11] and can be well-placed in the literature. However, I would like to see applications of these corrections (at least possible).*
>
> As we describe in our response under "Common concerns," we intend to add new versions of experiments done in Figure 1 and 2 where the sample sizes are larger (~10000).
>
> A major application of our work, which we will pursue next, will be to explore the relation between representation learning and generalization. We already calculated predictor statistics and the effect of width on generalization in linear networks in Appendix F.3. We also derive an explicit formula for test error. We will include a discussion of these point in the main text in the revised version and perform experiments to test these predictions, as also mentioned in our response under "Common concerns".
>
> *[...]*
>
> ***Minor:***
>
> - *“Pertubative”*
> - *“The second class of corrections are”*
> - *missed comma in (A.2)*
>
>     Thank you for noting these typos; all have been fixed.
>
> - I would reintroduce the notations in the appendix as well, like $E$ for error function, $n$, etc.
>
>     Thank you for this suggestion. We agree that reintroducing our notational conventions would help make the appendix more readable, and will do so in our revised manuscript.

---

> > ### Comment · Reviewer_QD2J · 2021-08-30
> > **Response to authors**
> >
> > I thank the authors for the detailed response and for the provided additional experiments. I think that the paper is worth publishing, therefore, I am raising my rating to 7.

---

### Author Response · Authors · 2021-08-10
**Common concerns for "Asymptotics of representation learning in finite Bayesian neural networks"**

We thank all referees for their thoughtful reviews of our paper. In this comment, we reply to common concerns, and describe major proposed changes to our manuscript. We reply in detail to specific referee comments individually.

***The generality of Conjecture 1***

We would first like to address questions regarding the generality of Conjecture 1. To do so, we will restate the conjecture in a more general form, and then sketch the perturbative argument upon which it is based (following Appendix B). Fundamentally, Conjecture 1 relates the large-width asymptotics of the posterior mean of a 'hidden layer observable' $O$ to its joint prior cumulants with the postactivation kernel $K^{(d-1)}$ of the final hidden layer. In our submitted manuscript, we took $O$ to be the postactivation kernel $K^{(\ell)}$ of the $\ell$-th hidden layer. However, in considering the reviewers' questions, we (belatedly) realized that our argument can be applied more broadly to other functions of the hidden layer activities that have deterministic infinite-width limits. Concretely, the statement of the revised version of Conjecture 1 would read as follows:

**Conjecture 1 (generalized)**: Consider a Bayesian neural network with linear readout of the form (1), with weight-space posterior (4). Assume that this network admits a well-defined Gaussian process limit, with the postactivation kernel of the final hidden layer $K^{(d-1)}$ tending in probability to a deterministic value $K^{(d-1)}\_{\infty}$. Let $O$ be a function of the hidden layer activities that tends to a finite deterministic infinite-width limit $O\_{\infty}$. Then, the posterior cumulants of this observable admit well-behaved asymptotic series expansions at large widths in terms of its joint prior cumulants with the postactivation kernel $K^{(d-1)}$. In particular, the asymptotic expansion of the posterior mean $\langle O \rangle$ has leading terms
$$
\langle O \rangle = \mathbb{E}\_{\mathcal{W}} O + \frac{1}{2} n\_{d} \sigma\_{d}^{2} \sum_{\rho,\lambda=1}^{p} [\beta^{2} \Gamma^{-1} G_{yy} \Gamma^{-1} - \beta \Gamma^{-1}]\_{\rho\lambda} \mathrm{cov}\_{\mathcal{W}}(O, K^{(d-1)}\_{\rho\lambda}) + \ldots,
$$
where $[G_{yy}]\_{\mu\nu} \equiv n_{d}^{-1} \mathbf{y}\_{\mu} \cdot \mathbf{y}\_{\nu}$ is the Gram matrix of the outputs and $\Gamma = I\_{p} + \beta \sigma_{d}^{2} K^{(d-1)}\_{\infty}$. Here, the cumulants of the kernels are computed with respect to the prior distribution of the hidden layer parameters $\mathcal{W}$, and are themselves given by asymptotic series at large widths. The ellipsis denotes terms that are of subleading order in the inverse hidden layer widths.

Our perturbative argument for this conjecture is based on a perturbative approximation for the posterior cumulant generating function $\log Z$ of the observable of interest. This argument proceeds as follows:

1. First, we use the fact that $O$ is independent of the readout weights $W^{(d)}$ to integrate out the readout layer. This yields an expression for $\log Z$ as an expectation over the prior distribution of the hidden layer weights alone. Thanks to the assumption that the readout is linear and the likelihood is Gaussian, the quantity we have to average depends on the hidden layer weights only through $O$ and the postactivation kernel matrix of the last hidden layer,  $K^{(d-1)}\_{\mu\nu} \equiv n\_{d-1}^{-1} \psi(\mathbf{x}\_{\mu},\mathcal{W}) \cdot \psi(\mathbf{x}\_{\nu},\mathcal{W})$. This yields (B.16) in the submitted manuscript.

2. Then, we assume that $K^{(d-1)}$ and $O$ tend to deterministic values at large widths. Concretely, we assume that $K^{(d-1)} = K_{\infty}^{(d-1)} + \delta K^{(d-1)}$ and $O = O_{\infty} + \delta O$, where $K_{\infty}^{(d-1)}$ and $O_{\infty}$ are deterministic and the deviations $\delta K^{(d-1)}$ and $\delta O$ are small with high probability. Under this assumption, we expand $\log Z$ as a formal series in moments of $\delta K^{(d-1)}$ and $\delta O$ with respect to the prior. This leads to (B.24) in the submitted manuscript.

3. We finally differentiate $\log Z$ with respect to the source variable to obtain $\langle O \rangle$ as a series in moments of $\delta K^{(d-1)}$ and $\delta O$. Assuming that successively higher moments are suppressed at large widths, we truncate this series at a relatively low order. The result can be re-expressed in terms of $\mathbb{E}\_{\mathcal{W}}$ and $\mathrm{cov}\_{\mathcal{W}}(O, K^{(d-1)}\_{\rho\lambda})$; higher-order terms can similarly be expressed in terms of higher joint prior cumulants of $O$ and $K^{(d-1)}$. This is (B.29) in the submitted manuscript, which yields Conjecture 1.

We emphasize that this heuristic argument does not assume a particular architecture for the feature map $\psi$ beyond the requirement that $K^{(d-1)}$ and $O$ have deterministic limits, and that their joint cumulants decay appropriately at large widths. Nor does this argument require $O$ to be one of the hidden layer postactivation kernels. Moreover, it could be extended to compute high-order posterior cumulants, though doing so would require truncating at higher orders (for instance, $\mathrm{var}(O) = \partial^2 \log Z/\partial J^2 |\_{J=0}$). For fully-connected networks, bounds on the required prior cumulants of various observables could be obtained from recent work by Gur-Ari and colleagues (https://arxiv.org/abs/1909.11304 and https://arxiv.org/abs/2006.06687).

Moreover, the zero-temperature limit of this generalized version of Conjecture 1 for fully-connected networks is derived perturbatively in Yaida's recent monograph with Roberts and Hanin (https://arxiv.org/abs/2106.10165), posted since the submission deadline. In particular, it is equation (6.99) of the version of their work posted to the arXiv on 18 June. The original version of Conjecture 1 for the postactivation kernels of deep MLPs (at zero temperature) is given by (6.104) of their work. We note that we arrive at Conjecture 1 via a very different path than those authors - we perform elementary perturbation theory under certain assumptions on the large-width behavior of the kernels, while they perform an ab initio calculation similar to that in Yaida's earlier work [11]. Of course, while our heuristic approach is simpler and does not assume a specific network architecture, it seems more likely that their approach could be made fully rigorous (that is, with rigorous bounds on the size of the errors) in the spirit of recent related work by Hanin (https://arxiv.org/abs/2107.01562). We will discuss this relationship in our revised manuscript.

With this in mind, we propose to revise our paper to introduce this generalized version of Conjecture 1, and then specialize to the case of hidden layer kernels as a set of observables that is of particular interest. This change would include thorough revision of Section 2.2 to more clearly set up the background for the new version of Conjecture 1, in keeping with Reviewer **QD2J**'s questions regarding the kernels we consider. We welcome the referees' feedback on this proposed change.

***Simulations in practically relevant settings***

All referees raised the question of the applicability of our theory to realistic networks, given that we are restricted by analytical tractability to deep linear networks and networks with a single non-linear hidden layer. We first would like to ensure that it is clear why this restriction is present: computation of the required prior cumulants is analytically intractable in deep nonlinear networks. This issue has been noted in many previous works, including Yaida [11], Naveh et al. [14], Dyer and Gur-Ari [17], Novak et al [6], Yang and Hu [8], and others. Furthermore, there do not as yet exist efficient numerical algorithms to accurately evaluate these cumulants.

Despite these issues, we performed experiments on deep linear and (nonlinear) erf networks and tested our predictions for finite width corrections. For finite-width deep linear networks, our theory explains analytically how feature learning happens. For erf networks, our theory correctly predicts the linear scaling with depth and inverse scaling with width.

To strengthen our experimental results and to address practical settings, we will attempt two new experiments:

1. We will explore realistic dataset sizes, as requested by **QD2J** and **jCKB**. The computational bottleneck here is the slowness of Langevin sampling, as noted by previous numerical studies of BNN posteriors; see, for instance, Izmailov et al. (http://proceedings.mlr.press/v139/izmailov21a.html), Fortuin et al. (https://arxiv.org/abs/2102.06571) or Wenzel et al. (http://proceedings.mlr.press/v119/wenzel20a.html). To mitigate this issue, we will use stochastic sampling techniques such as stochastic gradient Langevin dynamics (SGLD) (Welling and Teh, 2011), and we will attempt computing posterior weights and hence kernels at each layer for realistic dataset sizes. Based on our preliminary simulations, within our computational budget, we expect that we will be able to  simulate feedforward neural networks for sample sizes around 10000. We will test our predictions about inverse width scaling and linear depth scaling in this setting. We will repeat these experiments for CIFAR-10.

2. In Appendix Section F.3, we already derived the training and test errors for deep linear networks. We had not previously numerically tested these results. We will add tests to our revised manuscript, which we think will enhance the significance of our work.

---

> ### Author Response · Authors · 2021-08-25
> **Update on Experiments**
>
> As the reviewers suggested, we are testing our theory with real datasets of realistic size. As mentioned before, Langevin sampling for large datasets takes a very long time, hence we resorted to Stochastic Gradient Langevin Dynamics (SGLD - Welling and Teh, 2011) to train our models using mini-batches. As a preliminary simulation, here we present an experiment on a linear 3-layer Bayesian feedforward neural network with varying widths trained via SGLD with mini-batch size of 100. Our training set is composed of 1000 MNIST images. This is in contrast to our previously performed experiment (Figure 1) for low dimensional synthetic data and training set of size 100.
>
> We obtain excellent agreement with experiment, as shown in the linked figure: https://ibb.co/n614z52 For the figure showing 1/n scaling (left), each datapoint is divided by their layer number to show the linear scaling with depth. Currently we are working to test our theory on larger datasets of size ~10000.

---

> ### Author Response · Authors · 2021-08-30
> **Update on applicability to CNNs with many channels**
>
> Following the referees' comments, we have derived the kernel prior covariances required to predict learned representations in linear convolutional networks with many channels, following the setup of Novak et al. (https://arxiv.org/abs/1810.05148). In this setup, the kernels have two additional spatial indices (as they measure similarity across channel activations at two locations in space for two input examples). We take the filters to have size $2k+1$ and prior distribution $w\_{ij,\chi}^{(\ell)} \sim \mathcal{N}(0, \nu\_{\chi}\sigma\_{\ell}^{2}/n\_{\ell-1})$,  where $\chi$ indexes spatial locations and the spatial weights $\nu\_{\chi}$ satisfy $\sum\_{\chi = -k}^{k} \nu\_{\chi} = 1$. The infinite-channel kernels then obey the recurrence
> $$
>     [K^{(\ell)}\_{\infty}]\_{\mu\nu,\alpha\alpha'} = (\sigma\_w^\ell)^2 \sum\_{\chi = -k}^k \nu\_\chi [K^{(\ell-1)}\_{\infty}]\_{\mu\nu,(\alpha+\chi)(\alpha'+\chi)},
> $$
> and the required prior covariance is given as
> $$
> \mathrm{cov}\_{\mathcal{W}}(K^{(\ell)}\_{\mu\nu,\alpha \alpha'},K^{(\ell+\tau)}\_{\rho\lambda,\gamma \gamma'}) = (\sigma\_w^{\ell+1})^2 \cdots (\sigma\_w^{\tau})^2 \sum\_{\{\chi\_{\ell+1},...,\chi\_\tau\}  = -k}^k \nu\_{\chi\_{\ell+1}} \cdots \nu\_{\chi\_\tau} \mathrm{cov}\_{\mathcal{W}}(K\_{\mu\nu, \alpha \alpha'}^{(\ell)}, K\_{\rho\lambda,(\gamma+\sum\_{i=\ell+1}^\tau \chi\_i)(\gamma'+\sum\_{i=\ell+1}^\tau \chi\_i)}^{(\ell)})
> $$
> for
> $$
> \mathrm{cov}\_{\mathcal{W}}(K_{\mu\nu, \alpha \alpha'}^{(\ell)}, K\_{\rho\lambda,\gamma\gamma'}^{(\ell)}) = \left(\sum\_{\ell'=1}^{\ell} \frac{1}{n\_{\ell'}}\right) \left([K\_{\infty}^{(\ell)}]\_{\mu\rho, \alpha \gamma} [K\_{\infty}^{(\ell)}]\_{\nu\lambda, \alpha'\gamma'} + [K\_{\infty}^{(\ell)}]\_{\mu\lambda,\alpha\gamma'} [K\_{\infty}^{(\ell)}]\_{\nu\rho,\alpha'\gamma}\right) + \mathcal{O}(n^{-2}).
> $$
> Depending on readout strategy, the feature map kernel appearing in our conjecture 1 is given by a linear transformation of these results (see Section 3 of Novak et al.). Thus, like for a linear MLP, the required kernel covariances for a linear CNN can be expressed in terms of the corresponding infinite-width/channel kernels, allowing the corrections to be numerically computed relatively efficiently. We are currently working to experimentally test the resulting predictions, and will include these tests in our revised manuscript.

---

### Decision · Program_Chairs · 2021-09-27

**Decision:**

Accept (Poster)

**Comment:**

This paper analyzes the leading-order corrections to the average feature kernels in Bayesian neural networks, and conjectures a universal form for the result. The analysis and conclusions should be of interest to theorists in the NeurIPS community, and I recommend acceptance.